# Nanoparticles destabilizing the cell membranes triggered by NIR light for cancer imaging and photo-immunotherapy

Dongsheng Tang[1,2], Minhui Cui[1,2], Bin Wang[1,2], Ganghao Liang[1,2], Hanchen Zhang[1,2] & Haihua Xiao [1,2] ✉

Cationic polymers have great potential for cancer therapy due to their unique interactions with cancer cells. However, their clinical application remains limited by their high toxicity. Here we show a cell membrane-targeting cationic polymer with antineoplastic activity ($P^{mt}$) and a second near-infrared (NIR-II) fluorescent biodegradable polymer with photosensitizer Bodipy units and reactive oxygen species (ROS) responsive thioketal bonds ($P^{Bodipy}$). Subsequently, these two polymers can self-assemble into antineoplastic nanoparticles (denoted mt-NP$^{Bodipy}$) which could further accumulate at the tumor and destroy cell membranes through electrostatic interactions, resulting in cell membrane destabilization. Meanwhile, the photosensitizer Bodipy produces ROS to induce damage to cell membranes, proteins, and DNAs to kill cancer cells concertedly, finally resulting in cell membrane lysis and cancer cell death. This work highlights the use of near-infrared light to spatially and temporally control cationic polymers for photodynamic therapy, photo-immunotherapy, and NIR-II fluorescence for bio-imaging.

Chemotherapy and molecular targeted therapy remain the primary clinical cancer treatment modalities[1–3]. Despite their exceptional anti-tumor efficacy through DNA damage, inhibition of metabolism, anti-angiogenesis, etc., these treatment modalities are often associated with limitations such as severe toxicities, unsatisfactory therapeutic efficacy, and the development of multidrug resistance in tumors. This has resulted in a significant need for the development of alternative treatment modalities with better benefits[4–8]. To overcome the limitations of conventional anticancer drugs, polymer-drug conjugates, and polymer-based drug delivery systems have been extensively developed in recent years[9–14]. Besides acting as drug delivery vehicles or cancer vaccine adjuvants, some synthetic cationic polymers also exhibited antineoplastic activity by disrupting the cell membranes through their specific and unique interactions with cell membrane structures[15,16]. However, these cationic polymers can also rapidly interact with negatively charged cell membranes of normal cells in healthy tissues and organs in vivo, resulting in significant toxic side

effects and hindering their application in vivo. Therefore, the positive charge of these cationic polymers should be ideally shielded before reaching the tumor sites. Then, they can be rapidly transformed into cationic polymers triggered by some stimuli inside or ideally outside the tumor microenvironment, enabling a specific attack on the tumor cell membrane.

Recently, photodynamic therapy (PDT) has attracted a considerable amount of interest particularly in oncology, due to its minimally invasive tumor-therapeutic modalities[17]. PDT is predicated upon the light-induced excitation of sensitizers, propelling them from the ground state ($S_0$) to the first excited singlet state ($S_1$). Subsequently, these sensitizers undergo a transition to the triplet state ($T_1$) through the process of intersystem crossing (ISC). The T1 state has the capability to engage in interactions with proximate oxygen and water molecules, resulting in the production of cytotoxic reactive oxygen species (ROS), such as singlet oxygen ($^1O_2$) and hydroxyl radicals (·OH)[13]. ROS can induce irreversible damage to proteins and DNA,

[1]Beijing National Laboratory for Molecular Sciences, Laboratory of Polymer Physics and Chemistry, Institute of Chemistry, Chinese Academy of Sciences, Beijing 100190, PR China. [2]University of Chinese Academy of Sciences, Beijing 100049, PR China. ✉e-mail: hhxiao@iccas.ac.cn

subsequently leading to apoptosis and necrosis in tumor cells[18]. Furthermore, increasing empirical evidence has elucidated the immense capacity of PDT to elicit immunogenic cell death (ICD) in tumor cells, thereby fostering the potential for enhanced photo-immunotherapy[19]. With abundant tumor-associated antigens (TAAs) and damage-associated molecular patterns (DAMPs) being released from dying tumor cells, PDT induced ICD has been shown to facilitate antigen presentation by dendritic cells (DCs), thereby enabling the activation and infiltration of cytotoxic T lymphocytes (CTLs), ultimately eliciting systemic antitumor immune responses[20]. Nevertheless, in the majority of instances, the ICD effects induced by PDT are found to be insufficient for evoking an effective immune response for therapeutic objectives[21]. Furthermore, it has been reported that positively charged polymers, which contain primary, secondary, or tertiary amines, may function as adjuvants. This potential is attributed to the fact that the protonated amines are capable of causing disruption to the endo/lysosomal membranes and inducing the release of pro-inflammatory factors[22]. Such a process is primarily responsible for the destruction of endosome membranes, the release of cathepsin B, and $K^+$ efflux, all of which are crucial in the activation of inflammation within the host. Hence, the simultaneous delivery of photosensitizers and cationic polymers holds immense potential for enhancing immune responses.

In this work, we develop cationic antineoplastic anticancer nanoparticles (denoted as mt-NP$^{Bodipy}$) that can specifically anchor and destabilize cell membranes upon NIR light irradiation for PDT, photo-immunotherapy, and NIR-II fluorescence bio-imaging (Fig. 1). This unique design ensures that the quaternary ammonium cations can directly interact with the cell membranes, causing their destabilization. In addition, mt-NP$^{Bodipy}$ can produce large amounts of ROS upon continuous light irradiation, resulting in lipid peroxidation and further destabilization of cell membranes, and inducing photodynamic immunotherapy on the other hand. Subsequently, a mouse tumor model is constructed to evaluate the biodistribution, tumor targeting as well as performance of mt-NP$^{Bodipy}$ in inhibiting tumor growth via photodynamic immunotherapy in vivo. This work provides a paradigm for spatiotemporal controlling cationic charge-shielded nanoparticles using external stimuli. The cholesterols in the nanoparticles specifically target and anchor cell membranes, followed by generating large amounts of ROS with NIR irradiation, destabilizing cancer cell membrane and against tumor by activating a strong immune response.

## Results

### Synthesis and characterization of Aza-TPA-Bodipy

As shown in Fig. 2a, a Aza-Bodipy-based NIR-II fluorophore (Aza-TPA-Bodipy) was elaborately designed and synthesized, in which the triphenylamine (TPA) unit bears two hydroxyl groups at the ends[23,24]. The Bodipy unit serves as an acceptor, whose quinoidal character allows a greater electron delocalization and lowers the bandgap[25,26]. Meanwhile, TPA serves as the donor, resulting in a D-A-D′ type of Aza-TPA-Bodipy with redshift absorption[27]. Subsequently, the Aza-TPA-Bodipy was characterized by nuclear magnetic resonance spectroscopy ($^1H$ NMR) and matrix assisted laser desorption/ionization time-of-flight mass spectrometry (MALDI-TOF-MS) (Supplementary Fig. 1-9). The photophysical properties of the Aza-TPA-Bodipy were demonstrated by UV-Vis-NIR absorbance and photoluminescence (PL) spectra. As shown in Fig. 2b, the Aza-TPA-Bodipy in tetrahydrofuran (THF) showed an absorption maximum of 833 nm, indicating the ability to be excited by NIR light[28,29]. Upon excitation with an 808 nm laser, the Aza-TPA-Bodipy showed a fluorescence emission peak at 1017 nm and weak emission tails beyond 1400 nm, demonstrating that Aza-TPA-Bodipy could emit NIR-II fluorescence[30]. Meanwhile, the fluorescence quantum yield of Aza-TPA-Bodipy was even more efficient (1.1% quantum yield) than IR26 (0.5% quantum yield, commercial fluorescent dyes). Subsequently, by density functional theory (DFT) calculation, the optimized geometry results suggested that Aza-TPA-Bodipy tended to

form strongly twisted conformations along the backbone (Supplementary Fig. 10, Cartesian coordinates of optimized ground state was shown in Table 1). For the electronic structures (Fig. 2c), the HOMO wave functions are well delocalized among the whole molecular skeleton, while the LUMO wave functions distribute more on the electron-deficient Bodipy acceptor, indicating there is effective intramolecular charge transfer within Aza-TPA-Bodipy backbone, thereby resulting in long wavelength absorption. It has been reported that effective separation of HOMO-LUMO distribution can reduce $\Delta E_{ST}$ (the energy gap between the lowest S1 and T1)[31], which is in good agreement with the calculation result for Aza-TPA-Bodipy at a low $\Delta E_{ST}$ value of 0.51 eV (Fig. 2d, Cartesian coordinates of optimized S1 excited state and T1 excited state were shown in Table 2 and Table 3, respectively). Due to the reduced $\Delta E_{ST}$, the ISC efficiency is remarkably improved, indicating the superior ROS generation capability of Aza-TPA-Bodipy. Next, the ROS generation properties of Aza-TPA-Bodipy and ICG (an FDA-approved dye) were investigated and compared in an aqueous solution under 808 nm light irradiation (1.0 W cm$^{-2}$) with 1,3-diphenylisobenzofuran (DPBF) as a ROS probe[32]. From the slopes of the plots depicted in Fig. 2e and Supplementary Fig. 11, the DPBF decomposition rate constants in the presence of Aza-TPA-Bodipy and ICG are calculated to be 0.00771 and 0.00193 s$^{-1}$, respectively, indicating the ROS generation of Aza-TPA-Bodipy was even more efficient (16.4% quantum yield) than ICG (7.7% quantum yield)[33].

### Preparation and characterization of mt-NP$^{Bodipy}$

Due to the hydrophobic nature of organic photosensitizers, these compounds are typically loaded into hydrophobic polymer matrices or chemically bonded to polymers for water dispersibility and biostability in vivo[34-39]. Here, an amphiphilic polymer with Aza-TPA-Bodipy units and ROS-responsive thioketal bonds (2,2'-(propane-2,2-diylbis(sulfanediyl))bis(ethan-1-ol)[40], defined as PSDE) on its main chains (P$^{Bodipy}$) was designed and synthesized (Fig. 3a and Supplementary Fig. 12). P$^{Bodipy}$ with Aza-TPA-Bodipy as the photosensitizer was easier to disperse in water. Subsequently, the cell membrane targeting cationic amphiphilic polymer with cholesterol molecules on its side chains (P$^{mt}$) was synthesized (Fig. 3b and Supplementary Figs. 13–16)[41]. In doing so, a precursor polymer (P$^{Br}$) was designed and synthesized by the condensation polymerization[42] of monomers including 2,2-bis(bromomethyl)−1,3-propanediol (defined as BBrP) and (S)-ethyl 2,6-diisocyanatohexanoate (defined as EDH). Notably, P$^{Br}$ was unique as there were BBrP units with bromide groups providing a possibility for quaternization with cholesterol derivative (N, N-dimethyl-chol)[43]. In this way, a membrane-targeting cationic amphiphilic polymer P$^{mt}$ was obtained. P$^{Bodipy}$ and P$^{mt}$ were further characterized by $^1H$ NMR.

Next, P$^{mt}$ was self-assembled to NP$^{mt}$, which was subsequently shielded by P$^{Bodipy}$ to get mt-NP$^{Bodipy}$. For comparison, P$^{Bodipy}$ was co-assembled into the nanoparticles (termed as NP$^{Bodipy}$). The morphology of NP$^{Bodipy}$ and mt-NP$^{Bodipy}$ was further observed by transmission electron microscopy (TEM). The results indicated that NP$^{Bodipy}$ and mt-NP$^{Bodipy}$ had spherical morphology and a uniform size distribution (Fig. 3c and Supplementary Fig. 17a). The average particle sizes of NP$^{Bodipy}$ and mt-NP$^{Bodipy}$ measured by dynamic light scattering (DLS) were around 84 nm and 92 nm (Fig. 3d, e and Supplementary Fig. 17b−d) with polydispersity indexes (PDI) at 0.20 and 0.24, and Zeta potentials (mV) at −20 mV and −11 mV, respectively.

To further study the optical properties of NP$^{Bodipy}$ and mt-NP$^{Bodipy}$, the absorption spectra of NP$^{Bodipy}$ and mt-NP$^{Bodipy}$ were measured (Fig. 3f and Supplementary Fig. 17e). Compared with Aza-TPA-Bodipy, the maximum absorption peak ($\lambda_{max}$) of NP$^{Bodipy}$ and mt-NP$^{Bodipy}$ decreased. The $\lambda_{max}$ of NP$^{Bodipy}$ and mt-NP$^{Bodipy}$ decreased from 833 nm to 789 nm and 792 nm, respectively. Furthermore, the fluorescence (FL) emission spectra of NP$^{Bodipy}$ and mt-NP$^{Bodipy}$ demonstrated that they had maximum emission peaks at 926 and 914 nm, respectively, with a long emission tail extended to 1000-1200 nm (Fig. 3g and

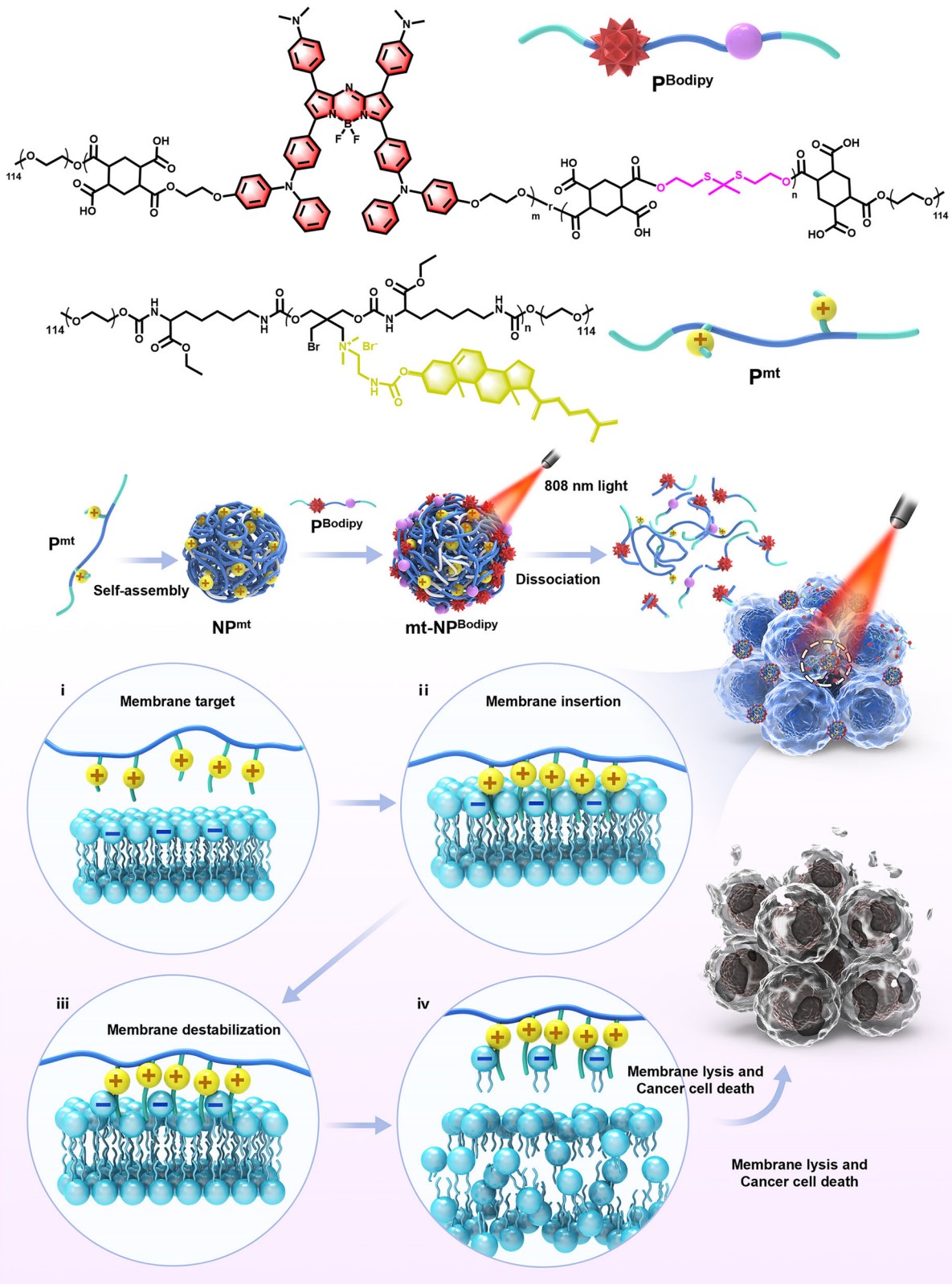

**Fig. 1 | Schematic illustration of nanoparticles destabilizing the cell membranes triggered by NIR light.** P[Bodipy] with NIR-II fluorescent Bodipy units and ROS-responsive thioketal linkages can be excited by NIR light to induce ROS, resulting in subsequent breakage of the thioketal bonds and degradation of the polymer. P[mt] contains numerous cholesterols for cell membrane targeting and quaternary ammonium salts for cell membrane destabilization. P[mt] was self-assembled to NP[mt], which was subsequently shielded by P[Bodipy] into mt-NP[Bodipy]. Under excitation with a laser at 808 nm, mt-NP[Bodipy] was dissociated and the cationic polymers P[mt] were released. Through electrostatic interactions, P[mt] can target and insert cell membranes (i, ii) and then destabilize and lyse the cell membranes (iii, iv), causing tumor cell death. Source data are provided as a Source Data file.

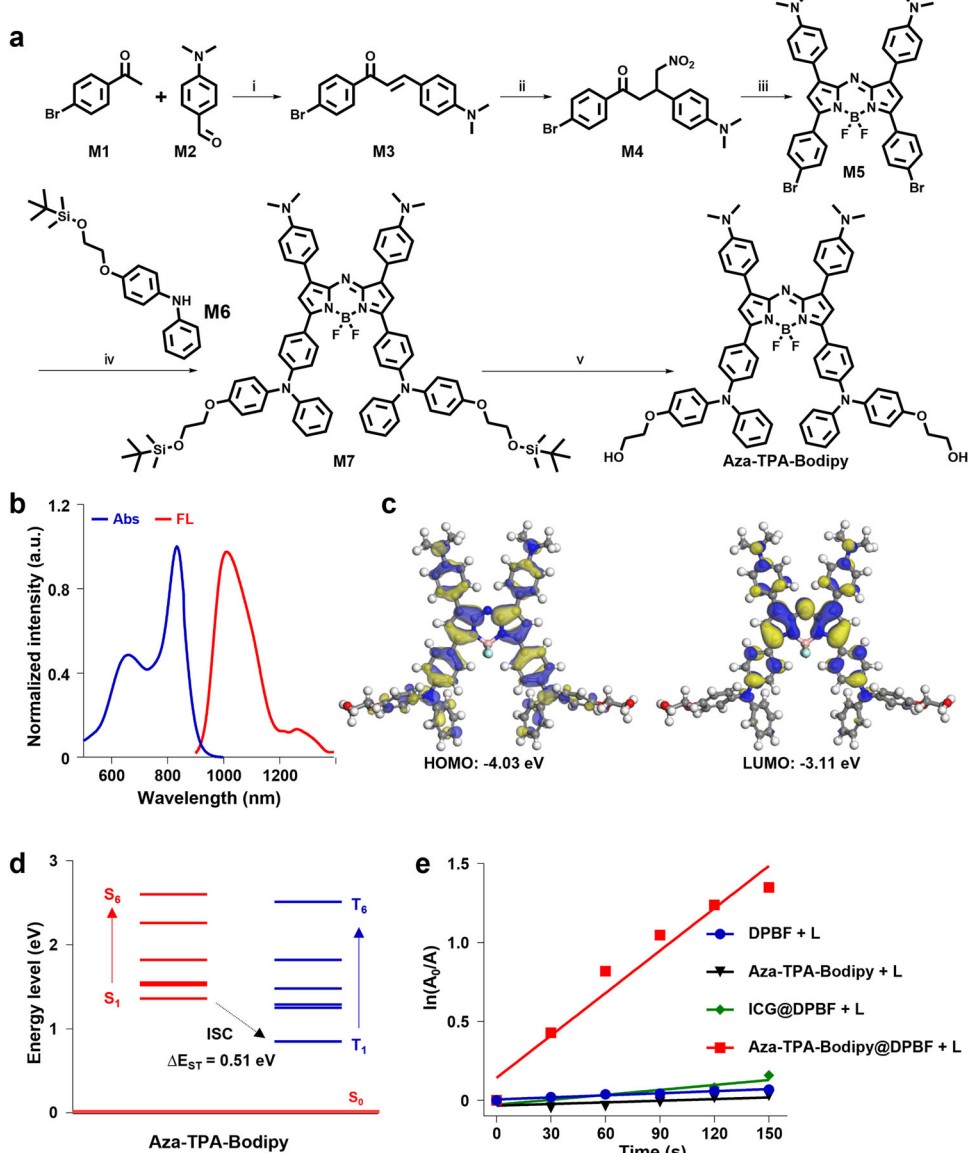

**Fig. 2 | Synthesis and characterization of Aza-TPA-Bodipy. a** Synthetic route of Aza-TPA-Bodipy. Reagents and conditions: i) CH₃COOH, piperidine, toluene, 80 °C, 24 h. ii) MeNO₂, Et₃N, EtOH, 95 °C, 48 h. iii) NH₄OAc, n-BuOH, 120 °C, 24 h; BF₃OEt₂, DIPEA, DCM, 25 °C, 24 h. iv) Pd₂(dba)₃, P(t-Bu)₃, sodium tert-butoxide, toluene, 85 °C, 24 h. v) Hydrogen fluoride pyridine, DCM, rt, 2 h. **b** Absorption and PL spectra of Aza-TPA-Bodipy in THF. **c** Optimized geometry and HOMO-LUMO distributions by DFT calculations of Aza-TPA-Bodipy. **d** Energy levels of S1-S6 and T1-T6 calculated by the vertical excitation of the optimized structures in (**c**). **e** Plot of ln($A_O/A$) against light exposure time, where $A_O$ and $A$ are the DPBF absorbance (415 nm) before and after irradiation, respectively. Source data are provided as a Source Data file.

Supplementary Fig. 17f). Notably, NP^Bodipy and mt-NP^Bodipy had a large portion of the emission spectrum in the NIR-II region (>1000 nm), suggesting that the emitted photons could be used for in vivo NIR-II fluorescence bioimaging[44]. Meanwhile, compared with Aza-TPA-Bodipy, the maximum emission peak of NP^Bodipy and mt-NP^Bodipy also decreased, implying that the physical and optical properties of NP^Bodipy and mt-NP^Bodipy displayed notable change compared to Aza-TPA-Bodipy.

Subsequently, the generation of ¹O₂ of mt-NP^Bodipy was detected by electron spin resonance (ESR) using 2,2,6,6-tetramethylpiperide (TEMP) as a trapping agent (Fig. 3h). Compared with control groups and mt-NP^Bodipy, the peak intensity of mt-NP^Bodipy + L increased, indicating mt-NP^Bodipy + L could efficiently generate ¹O₂. Further, the morphological and size changes of the mt-NP^Bodipy + L were studied. After 808 nm light irradiation for 10 min, the particle sizes were enlarged and a small portion of mt-NP^Bodipy was disassociated possibly owing to the breakdown of thioketal bonds (Fig. 3i). Notably, the TEM results

displayed that the spheric nanoparticles almost disappeared after light irradiation for 10 min, indicating the complete disassociation of mt-NP^Bodipy (Fig. 3j).

### In vitro evaluation of mt-NP^Bodipy

The intracellular uptake of mt-NP^Bodipy was studied by labeling the nanoparticles with Cy5.5 (mt-NP^Bodipy@Cy5.5, red fluorescence, Supplementary Fig. 18). mt-NP^Bodipy@Cy5.5 were then treated with CT26 cells and observed by a confocal laser scanning microscopy (CLSM). Moreover, mt-NP^Bodipy@Cy5.5 was firstly irradiated by an 808 nm laser at different time points from 0 to 10 min (Fig. 4a), resulting in partially dissociated nanoparticles which were then used to incubate with CT26 cells for 1 h. As shown in Fig. 4b, mt-NP^Bodipy@Cy5.5 irradiated for 10 min demonstrated strong red fluorescence on the cell membranes (labeled with DiO, green fluorescence). However, mt-NP^Bodipy@Cy5.5 irradiated for 0, 1, and 5 min exhibited negligible red fluorescence. The quantification of the fluorescence signal indicated that mt-

**Table 1 | Cartesian coordinates of optimized ground state calculated by the DFT, B3LYP/6-31 G(d), Gaussian 16 program**

| Atom | x | y | z |
|---|---|---|---|
| C | −4.10633 | 7.77056 | 0.1619 |
| N | −4.02095 | 6.38343 | 0.1711 |
| B | −2.82209 | 5.57168 | −0.39307 |
| N | −1.61888 | 6.55118 | −0.41886 |
| C | −1.74762 | 7.93703 | −0.48838 |
| C | −3.00965 | 8.52266 | −0.29444 |
| C | −5.3758 | 8.1332 | 0.74783 |
| C | −6.01248 | 6.94753 | 1.04981 |
| C | −5.17941 | 5.86673 | 0.66278 |
| C | −0.30797 | 6.2145 | −0.55714 |
| C | 0.43966 | 7.41617 | −0.67644 |
| C | −0.42183 | 8.49046 | −0.6381 |
| C | −5.92848 | 9.47965 | 1.1081 |
| C | −5.44224 | 4.46292 | 0.68855 |
| C | 0.04813 | 9.91416 | −0.62322 |
| C | 0.18504 | 4.86821 | −0.54215 |
| F | −2.53732 | 4.49533 | 0.46471 |
| F | −3.10484 | 5.0952 | −1.67157 |
| C | −6.58197 | 3.89111 | 1.147 |
| C | −6.8552 | 2.46923 | 1.10429 |
| C | 1.46913 | 4.56746 | −0.85809 |
| C | 2.13266 | 3.28348 | −0.80216 |
| C | −7.86679 | 1.90298 | 1.90103 |
| C | −8.09567 | 0.53634 | 1.92892 |
| C | −7.32781 | −0.32584 | 1.13103 |
| C | −6.35858 | 0.23739 | 0.27885 |
| C | −6.12441 | 1.59503 | 0.27532 |
| C | 3.46662 | 3.19857 | −1.24356 |
| C | 4.1888 | 2.02421 | −1.17255 |
| C | 3.61282 | 0.863 | −0.62849 |
| C | 2.27037 | 0.92345 | −0.20511 |
| C | 1.55244 | 2.10293 | −0.29494 |
| C | 0.74241 | −6.43475 | −3.76498 |
| C | 0.46781 | −5.11178 | −4.10301 |
| C | −0.5752 | −4.43274 | −3.48149 |
| C | −1.35273 | −5.05512 | −2.49393 |
| C | −1.07034 | −6.3898 | −2.17157 |
| C | −0.03769 | −7.0737 | −2.80389 |
| C | −2.49435 | −4.33783 | −1.87448 |
| C | −3.3159 | −3.52016 | −2.81057 |
| C | −2.78084 | −4.40277 | −0.54732 |
| C | −4.04571 | −3.85174 | −0.0019 |
| C | −1.82795 | −4.9127 | 0.47361 |
| C | −3.59675 | −2.1768 | −2.52867 |
| C | −4.36152 | −1.41064 | −3.40065 |
| C | −4.86138 | −1.97492 | −4.57321 |
| C | −4.57365 | −3.30463 | −4.87406 |
| C | −3.7974 | −4.0664 | −4.00609 |
| C | −2.24121 | −5.84523 | 1.43402 |
| C | −1.37994 | −6.25136 | 2.4467 |
| C | −0.1024 | −5.7028 | 2.53946 |
| C | 0.31268 | −4.7598 | 1.60338 |
| C | −0.53712 | −4.37903 | 0.56961 |
| C | −5.2857 | −4.12557 | −0.59415 |

**Table 1 (continued) | Cartesian coordinates of optimized ground state calculated by the DFT, B3LYP/6-31 G(d), Gaussian 16 program**

| Atom | x | y | z |
|---|---|---|---|
| C | −6.4392 | −3.47613 | −0.17854 |
| C | −6.3744 | −2.53074 | 0.84786 |
| C | −5.16246 | −2.32799 | 1.51632 |
| C | −4.01645 | −2.98568 | 1.10005 |
| N | −7.49437 | −1.71924 | 1.17262 |
| C | −8.76776 | −2.31976 | 1.34154 |
| C | −8.89043 | −3.49129 | 2.10097 |
| C | −10.12105 | −4.10502 | 2.2563 |
| C | −11.26742 | −3.55397 | 1.67295 |
| C | −11.15575 | −2.38397 | 0.91934 |
| C | −9.91015 | −1.78364 | 0.74792 |
| O | −12.43365 | −4.22474 | 1.89476 |
| C | −13.62783 | −3.7126 | 1.32046 |
| C | −14.74841 | −4.64551 | 1.72993 |
| O | −15.93415 | −4.11984 | 1.15038 |
| O | 0.9291 | −7.81311 | 0.13409 |
| C | 2.27973 | −7.84228 | −0.27868 |
| C | 3.02316 | −6.57355 | 0.08332 |
| O | 2.43722 | −5.52955 | −0.69167 |
| C | 2.82453 | −1.99195 | −1.38716 |
| C | 2.36736 | −3.30014 | −1.41435 |
| C | 2.91815 | −4.26234 | −0.55977 |
| C | 3.89991 | −3.88221 | 0.35671 |
| C | 4.33493 | −2.56036 | 0.39683 |
| C | 3.82463 | −1.60574 | −0.48392 |
| N | 4.38216 | −0.29862 | −0.49659 |
| C | 8.01638 | −0.86761 | −1.02396 |
| C | 6.64032 | −0.93188 | −1.19036 |
| C | 5.79588 | −0.19667 | −0.35414 |
| C | 6.35363 | 0.61645 | 0.63606 |
| C | 7.72953 | 0.67105 | 0.79977 |
| C | 10.5034 | 1.1104 | −1.97831 |
| C | 11.24892 | 1.30093 | −3.13697 |
| C | 12.35708 | 0.49706 | −3.39587 |
| C | 12.71047 | −0.50015 | −2.4885 |
| C | 11.97154 | −0.68215 | −1.32555 |
| C | 8.96923 | −1.76702 | 2.52253 |
| C | 8.22712 | −2.169 | 3.62693 |
| C | 8.32361 | −1.47022 | 4.82916 |
| C | 9.18301 | −0.37746 | 4.92145 |
| C | 9.9346 | 0.01631 | 3.81901 |
| C | 10.86387 | 0.1288 | −1.04556 |
| C | 8.58654 | −0.08626 | −0.00988 |
| C | 10.05831 | −0.06345 | 0.18889 |
| C | 9.82344 | −0.65879 | 2.5967 |
| C | 10.61876 | −0.21972 | 1.41839 |
| C | 14.15773 | −0.64869 | 2.69528 |
| C | 12.80782 | −0.87221 | 2.44593 |
| C | 12.05926 | 0.03339 | 1.68202 |
| C | 12.69363 | 1.18904 | 1.20737 |
| C | 14.03971 | 1.41887 | 1.46579 |
| C | 14.77987 | 0.49734 | 2.20436 |
| C | −3.21433 | 9.97432 | −0.65592 |

**Table 1 (continued) | Cartesian coordinates of optimized ground state calculated by the DFT, B3LYP/6-31 G(d), Gaussian 16 program**

| Atom | x | y | z |
|------|-----|-----|-----|
| F | −4.47172 | 10.22441 | −1.0678 |
| F | −2.42149 | 10.36068 | −1.6739 |
| F | −2.96481 | 10.8178 | 0.3816 |
| H | −6.97979 | 6.85704 | 1.52179 |
| H | 1.51634 | 7.48286 | −0.71928 |
| H | −6.67096 | 9.36559 | 1.90018 |
| H | −5.15899 | 10.16273 | 1.46826 |
| H | −6.41713 | 9.96276 | 0.26026 |
| H | −4.64827 | 3.83022 | 0.3187 |
| H | −0.49534 | 10.52543 | 0.09738 |
| H | 1.10579 | 9.94199 | −0.3546 |
| H | −0.06108 | 10.39194 | −1.59814 |
| H | −0.51327 | 4.09678 | −0.24719 |
| H | −7.35122 | 4.51522 | 1.59648 |
| H | 2.10917 | 5.38072 | −1.19219 |
| H | −8.45849 | 2.54916 | 2.54264 |
| H | −8.85694 | 0.12537 | 2.58084 |
| H | −5.79501 | −0.40714 | −0.38338 |
| H | −5.38329 | 1.995 | −0.40673 |
| H | 3.94109 | 4.08293 | −1.65811 |
| H | 5.21103 | 1.99655 | −1.52726 |
| H | 1.80626 | 0.04274 | 0.22173 |
| H | 0.5303 | 2.11856 | 0.06702 |
| H | 1.55119 | −6.96703 | −4.25399 |
| H | 1.06531 | −4.60685 | −4.85505 |
| H | −0.79688 | −3.40903 | −3.76187 |
| H | −1.68142 | −6.89863 | −1.4355 |
| H | 0.15505 | −8.10626 | −2.53725 |
| H | −3.21484 | −1.7379 | −1.61433 |
| H | −4.56425 | −0.37039 | −3.16774 |
| H | −5.4623 | −1.37906 | −5.25193 |
| H | −4.95222 | −3.74913 | −5.78874 |
| H | −3.56732 | −5.09863 | −4.24746 |
| H | −3.24668 | −6.24899 | 1.38027 |
| H | −1.70673 | −6.99286 | 3.16784 |
| H | 0.56609 | −6.01252 | 3.33545 |
| H | 1.2983 | −4.3138 | 1.67329 |
| H | −0.2081 | −3.64762 | −0.15881 |
| H | −5.33358 | −4.81484 | −1.4284 |
| H | −7.38377 | −3.6608 | −0.67689 |
| H | −5.12219 | −1.61515 | 2.33178 |
| H | −3.07573 | −2.79943 | 1.60578 |
| H | −8.00867 | −3.91784 | 2.56482 |
| H | −10.22335 | −5.01184 | 2.84108 |
| H | −12.02176 | −1.93879 | 0.44746 |
| H | −9.82871 | −0.88607 | 0.14652 |
| H | −13.83463 | −2.70049 | 1.68589 |
| H | −13.55298 | −3.67851 | 0.22786 |
| H | −14.52753 | −5.65749 | 1.36628 |
| H | −14.80757 | −4.68033 | 2.82557 |
| H | −16.67261 | −4.68898 | 1.3879 |
| H | 0.53998 | −6.98683 | −0.18334 |
| H | 2.7444 | −8.69237 | 0.22648 |

**Table 1 (continued) | Cartesian coordinates of optimized ground state calculated by the DFT, B3LYP/6-31 G(d), Gaussian 16 program**

| Atom | x | y | z |
|------|-----|-----|-----|
| H | 2.36591 | −7.99973 | −1.3629 |
| H | 2.90516 | −6.36237 | 1.15169 |
| H | 4.09168 | −6.66177 | −0.15091 |
| H | 2.41855 | −1.26489 | −2.08086 |
| H | 1.60668 | −3.61526 | −2.11754 |
| H | 4.33395 | −4.59908 | 1.04116 |
| H | 5.10347 | −2.27255 | 1.10432 |
| H | 8.66361 | −1.43688 | −1.68196 |
| H | 6.20764 | −1.55779 | −1.96233 |
| H | 5.70114 | 1.19698 | 1.27787 |
| H | 8.1499 | 1.2885 | 1.58423 |
| H | 9.63576 | 1.7315 | −1.78406 |
| H | 10.96248 | 2.07585 | −3.84052 |
| H | 12.93618 | 0.64098 | −4.30197 |
| H | 13.56415 | −1.13948 | −2.68794 |
| H | 12.25296 | −1.4551 | −0.62049 |
| H | 8.88729 | −2.30917 | 1.58799 |
| H | 7.57451 | −3.03286 | 3.5509 |
| H | 7.74072 | −1.78062 | 5.68992 |
| H | 9.27047 | 0.16818 | 5.85526 |
| H | 10.60987 | 0.86136 | 3.89698 |
| H | 14.72414 | −1.36855 | 3.2771 |
| H | 12.32417 | −1.75991 | 2.83855 |
| H | 12.12344 | 1.906 | 0.62891 |
| H | 14.51214 | 2.32088 | 1.09099 |
| H | 15.83145 | 0.67551 | 2.40322 |

NP$^{Bodipy}$@Cy5.5 irradiated for 10 min demonstrated strong red fluorescence (Fig. 4c, d). These results clearly demonstrated that NIR light could induce ROS generation and trigger the dissociation of mt-NP$^{Bodipy}$ to release P$^{mt}$ to target and destroy the cell membranes.

The cytotoxicity of the mt-NP$^{Bodipy}$ + L was evaluated against mouse colon cancer cell line CT26 by a 3-(4,5-dimethylthiazol-2-yl)−2,5-diphenyltetra-zolium bromide (MTT) assay (Fig. 4e). The results showed that NP$^{Bodipy}$ + L and mt-NP$^{Bodipy}$ + L demonstrated strong anticancer activity with low IC$_{50}$ values. mt-NP$^{Bodipy}$ formed from P$^{Bodipy}$ and P$^{mt}$ with more cationic charges than NP$^{Bodipy}$, had lower IC$_{50}$ values (NP$^{Bodipy}$ + L, IC$_{50}$ = 0.62 ± 0.30 μg mL$^{-1}$; mt-NP$^{Bodipy}$ + L, IC$_{50}$ = 0.50 ± 0.20 μg mL$^{-1}$). Particularly, the anticancer activity of NP$^{Bodipy}$ and mt-NP$^{Bodipy}$ was low as evidenced by high IC$_{50}$ values (both IC$_{50}$ > 10 μg mL$^{-1}$). This was possibly due to the excellent biocompatibility of NP$^{Bodipy}$ and mt-NP$^{Bodipy}$. The anticancer activity of mt-NP$^{Bodipy}$ + L was further visualized by 3D tumor cell spheres by CLSM (Fig. 4f). The results showed that 3D tumor cell spheres treated with PBS and NP$^{Bodipy}$ had strong green fluorescence (Calcein-AM, cells alive) and little red fluorescence (propidium iodide, cells dead), indicating NP$^{Bodipy}$ had no apparent toxicity. Particularly, the 3D tumor cell spheres treated with mt-NP$^{Bodipy}$ demonstrated more red fluorescence than NP$^{Bodipy}$, which was possibly due to the cationic nature of mt-NP$^{Bodipy}$. In contrast, the anticancer efficacy of NP$^{Bodipy}$ + L and mt-NP$^{Bodipy}$ + L was the highest as evidenced by the strong red fluorescence signals. We hypothesized that the mt-NP$^{Bodipy}$ were attached to the cell membranes via electrostatic interaction, resulting in damage to the membranes and thus leading to cell necrosis. Furthermore, the morphological changes in CT26 cells were observed by scanning electron microscopy (SEM). As shown in Fig. 4g, compared to PBS, NP$^{Bodipy}$, and mt-NP$^{Bodipy}$ treated cells with intact membranes, the cells shrank and the

**Table 2 | Cartesian coordinates of optimized S1 excited state calculated by the DFT, B3LYP/6-31 G(d), Gaussian 16 program**

| Atom | x | y | z |
|---|---|---|---|
| C | 3.73724 | 7.91101 | −0.10824 |
| N | 3.72322 | 6.53768 | −0.10384 |
| B | 2.58921 | 5.6727 | 0.50922 |
| N | 1.33447 | 6.58271 | 0.52584 |
| C | 1.3866 | 7.96072 | 0.58873 |
| C | 2.61771 | 8.61794 | 0.39582 |
| C | 4.95203 | 8.34822 | −0.736 |
| C | 5.6493 | 7.19964 | −1.0552 |
| C | 4.89446 | 6.08128 | −0.64443 |
| C | 0.03423 | 6.18049 | 0.66998 |
| C | −0.76535 | 7.34203 | 0.7965 |
| C | 0.04188 | 8.45434 | 0.74543 |
| C | 5.39522 | 9.72708 | −1.10846 |
| C | 5.2286 | 4.69655 | −0.6818 |
| C | −0.48088 | 9.85493 | 0.7265 |
| C | −0.4024 | 4.82193 | 0.63713 |
| F | 2.35535 | 4.56367 | −0.3096 |
| F | 2.92209 | 5.25371 | 1.78732 |
| C | 6.38476 | 4.17105 | −1.18071 |
| C | 6.72648 | 2.77169 | −1.12819 |
| C | −1.68098 | 4.46249 | 0.95762 |
| C | −2.30435 | 3.16717 | 0.86771 |
| C | 7.77553 | 2.25365 | −1.91936 |
| C | 8.09953 | 0.91039 | −1.91451 |
| C | 7.39456 | 0.0153 | −1.0884 |
| C | 6.37704 | 0.52558 | −0.2592 |
| C | 6.05171 | 1.86185 | −0.2848 |
| C | −3.63212 | 3.03689 | 1.32519 |
| C | −4.33183 | 1.85502 | 1.22595 |
| C | −3.74424 | 0.72182 | 0.63539 |
| C | −2.40722 | 0.81937 | 0.19836 |
| C | −1.7087 | 2.0101 | 0.31653 |
| C | −0.08923 | −7.50311 | 3.13107 |
| C | 0.03124 | −6.23191 | 3.6875 |
| C | 1.00675 | −5.35609 | 3.22643 |
| C | 1.87265 | −5.7219 | 2.18319 |
| C | 1.74611 | −7.01032 | 1.63989 |
| C | 0.77754 | −7.89163 | 2.10983 |
| C | 2.92728 | −4.78564 | 1.72506 |
| C | 3.61488 | −4.03798 | 2.81103 |
| C | 3.24085 | −4.60218 | 0.40231 |
| C | 4.43442 | −3.84459 | −0.0304 |
| C | 2.3823 | −5.07708 | −0.71598 |
| C | 3.7385 | −2.64192 | 2.7598 |
| C | 4.35973 | −1.94325 | 3.78883 |
| C | 4.87354 | −2.62673 | 4.88931 |
| C | 4.74555 | −4.01225 | 4.96003 |
| C | 4.11119 | −4.70964 | 3.93757 |
| C | 2.94156 | −5.77445 | −1.79589 |
| C | 2.1559 | −6.17065 | −2.87313 |
| C | 0.80137 | −5.85092 | −2.90601 |
| C | 0.23682 | −5.13801 | −1.85017 |
| C | 1.01706 | −4.76107 | −0.76109 |
| C | 5.69817 | −4.02925 | 0.55952 |

**Table 2 (continued) | Cartesian coordinates of optimized S1 excited state calculated by the DFT, B3LYP/6-31 G(d), Gaussian 16 program**

| Atom | x | y | z |
|---|---|---|---|
| C | 6.78325 | −3.24899 | 0.19589 |
| C | 6.62968 | −2.24981 | −0.77584 |
| C | 5.39479 | −2.10657 | −1.42567 |
| C | 4.32247 | −2.89751 | −1.06346 |
| N | 7.6753 | −1.35518 | −1.08395 |
| C | 8.98616 | −1.8466 | −1.24284 |
| C | 9.20541 | −3.02363 | −1.97605 |
| C | 10.48233 | −3.52791 | −2.12644 |
| C | 11.5784 | −2.86355 | −1.56071 |
| C | 11.37062 | −1.6892 | −0.83207 |
| C | 10.08233 | −1.19433 | −0.66965 |
| O | 12.7865 | −3.43126 | −1.77322 |
| C | 13.92593 | −2.80244 | −1.22094 |
| C | 15.1244 | −3.63961 | −1.61364 |
| O | 16.24868 | −2.99811 | −1.05666 |
| O | −0.95892 | −7.86892 | −0.53873 |
| C | −2.28565 | −7.92065 | −0.0856 |
| C | −3.04653 | −6.6409 | −0.36038 |
| O | −2.445 | −5.62774 | 0.43049 |
| C | −2.8887 | −2.13128 | 1.26989 |
| C | −2.39944 | −3.4273 | 1.23466 |
| C | −2.95265 | −4.37188 | 0.35931 |
| C | −3.97758 | −3.98536 | −0.50591 |
| C | −4.44775 | −2.67734 | −0.48236 |
| C | −3.92992 | −1.73885 | 0.41427 |
| N | −4.49935 | −0.4461 | 0.48001 |
| C | −8.11781 | −1.07555 | 1.00399 |
| C | −6.74038 | −1.14512 | 1.14989 |
| C | −5.90761 | −0.35498 | 0.35012 |
| C | −6.48275 | 0.5092 | −0.58823 |
| C | −7.86079 | 0.566 | −0.73106 |
| C | −10.52079 | 0.79537 | 2.18674 |
| C | −11.20557 | 0.8771 | 3.3947 |
| C | −12.30856 | 0.06106 | 3.62971 |
| C | −12.71925 | −0.83745 | 2.6483 |
| C | −12.04063 | −0.91109 | 1.43718 |
| C | −9.26043 | −1.71959 | −2.62116 |
| C | −8.60192 | −2.06534 | −3.796 |
| C | −8.73927 | −1.27686 | −4.9355 |
| C | −9.55383 | −0.14852 | −4.89482 |
| C | −10.22252 | 0.18952 | −3.72303 |
| C | −10.93699 | −0.08792 | 1.1821 |
| C | −8.70702 | −0.2361 | 0.04742 |
| C | −10.18482 | −0.1778 | −0.0996 |
| C | −10.07115 | −0.57929 | −2.56183 |
| C | −10.79676 | −0.21248 | −1.31454 |
| C | −14.38685 | −0.39528 | −2.51149 |
| C | −13.0424 | −0.69414 | −2.31835 |
| C | −12.2371 | 0.11027 | −1.50106 |
| C | −12.81166 | 1.24295 | −0.91088 |
| C | −14.15265 | 1.54856 | −1.11308 |
| C | −14.9476 | 0.7273 | −1.90857 |
| C | 2.77086 | 10.05304 | 0.78773 |

**Table 2 (continued) | Cartesian coordinates of optimized S1 excited state calculated by the DFT, B3LYP/6-31 G(d), Gaussian 16 program**

| Atom | x | y | z |
|---|---|---|---|
| F | 4.02459 | 10.3356 | 1.17646 |
| F | 1.98478 | 10.37922 | 1.82678 |
| F | 2.47301 | 10.92244 | −0.20955 |
| H | 6.60782 | 7.1655 | −1.55754 |
| H | −1.84643 | 7.35902 | 0.84714 |
| H | 6.11135 | 9.66734 | −1.934 |
| H | 4.56015 | 10.35545 | −1.43134 |
| H | 5.88546 | 10.24568 | −0.27784 |
| H | 4.47754 | 4.02656 | −0.27708 |
| H | 0.04212 | 10.48162 | −0.00132 |
| H | −1.54222 | 9.84063 | 0.45977 |
| H | −0.38722 | 10.34623 | 1.70028 |
| H | 0.32824 | 4.08215 | 0.32613 |
| H | 7.10923 | 4.82539 | −1.66356 |
| H | −2.3453 | 5.24499 | 1.32282 |
| H | 8.32055 | 2.92648 | −2.57733 |
| H | 8.88585 | 0.53685 | −2.56356 |
| H | 5.85766 | −0.14255 | 0.42044 |
| H | 5.28162 | 2.22534 | 0.38851 |
| H | −4.117 | 3.89915 | 1.77738 |
| H | −5.34906 | 1.79629 | 1.59938 |
| H | −1.93221 | −0.03991 | −0.2659 |
| H | −0.69047 | 2.05873 | −0.06011 |
| H | −0.84638 | −8.191 | 3.49739 |
| H | −0.63446 | −5.92155 | 4.48876 |
| H | 1.10492 | −4.371 | 3.67529 |
| H | 2.42325 | −7.32383 | 0.85085 |
| H | 0.70121 | −8.88306 | 1.6718 |
| H | 3.33606 | −2.10641 | 1.90427 |
| H | 4.43608 | −0.86017 | 3.73581 |
| H | 5.36167 | −2.08071 | 5.69217 |
| H | 5.13649 | −4.55238 | 5.81853 |
| H | 4.00083 | −5.78915 | 4.00309 |
| H | 4.00316 | −6.00888 | −1.78198 |
| H | 2.60377 | −6.72766 | −3.69183 |
| H | 0.18942 | −6.15242 | −3.75154 |
| H | −0.81528 | −4.86391 | −1.87596 |
| H | 0.57108 | −4.20574 | 0.05954 |
| H | 5.81541 | −4.77482 | 1.33997 |
| H | 7.74507 | −3.38238 | 0.68251 |
| H | 5.28607 | −1.35072 | −2.19811 |
| H | 3.36911 | −2.77159 | −1.56892 |
| H | 8.36086 | −3.53463 | −2.42866 |
| H | 10.66211 | −4.43528 | −2.69456 |
| H | 12.1998 | −1.16158 | −0.37357 |
| H | 9.92354 | −0.29207 | −0.08683 |
| H | 14.04038 | −1.78281 | −1.6125 |
| H | 13.85287 | −2.74808 | −0.12653 |
| H | 14.99256 | −4.66109 | −1.22665 |
| H | 15.17845 | −3.69827 | −2.711 |
| H | 17.03185 | −3.51109 | −1.28986 |
| H | −0.5562 | −7.07827 | −0.14771 |
| H | −2.77136 | −8.74188 | −0.62336 |

**Table 2 (continued) | Cartesian coordinates of optimized S1 excited state calculated by the DFT, B3LYP/6-31 G(d), Gaussian 16 program**

| Atom | x | y | z |
|---|---|---|---|
| H | −2.34226 | −8.14056 | 0.99264 |
| H | −2.97137 | −6.38656 | −1.42658 |
| H | −4.1076 | −6.75017 | −0.09194 |
| H | −2.47798 | −1.4168 | 1.97746 |
| H | −1.60796 | −3.74652 | 1.9073 |
| H | −4.41804 | −4.69075 | −1.20221 |
| H | −5.24637 | −2.38384 | −1.15709 |
| H | −8.75505 | −1.68135 | 1.64283 |
| H | −6.29701 | −1.81028 | 1.88523 |
| H | −5.84041 | 1.13288 | −1.20307 |
| H | −8.29419 | 1.2328 | −1.47042 |
| H | −9.65404 | 1.4276 | 2.01134 |
| H | −10.87385 | 1.57852 | 4.15585 |
| H | −12.84102 | 0.11972 | 4.57523 |
| H | −13.57111 | −1.48823 | 2.82733 |
| H | −12.36503 | −1.61324 | 0.67462 |
| H | −9.15288 | −2.33816 | −1.73463 |
| H | −7.98425 | −2.95951 | −3.82386 |
| H | −8.22195 | −1.54532 | −5.85268 |
| H | −9.67333 | 0.46971 | −5.7807 |
| H | −10.86807 | 1.06373 | −3.69933 |
| H | −14.9971 | −1.03945 | −3.13922 |
| H | −12.60566 | −1.5645 | −2.80131 |
| H | −12.19573 | 1.88714 | −0.29018 |
| H | −14.57732 | 2.43533 | −0.64995 |
| H | −15.99638 | 0.96567 | −2.06429 |

morphology of the membranes became irregular after treatment with NP$^{Bodipy}$ + L. Upon treatment of the cells with mt-NP$^{Bodipy}$ + L, a large number of necrotic cells with completely disintegrated cell membranes were observed. This phenomenon indicated that the mt-NP$^{Bodipy}$ + L could successfully release cationic polymer P$^{mt}$ to destroy the cell membranes.

### Mechanisms of action by RNA-seq analysis

To investigate the impact of NP$^{Bodipy}$ + L, the transcriptome landscape of the CT26 cells treated by PBS, mt-NP$^{Bodipy}$, and mt-NP$^{Bodipy}$ + L was profiled. A total of 16592 genes were identified. Both the principal component analysis (PCA) and Venn diagram indicated the significant discrepancy in the transcriptome landscape between the CT26 cells treated with mt-NP$^{Bodipy}$ + L and PBS (Fig. 5a, b). Compared to the cells treated with PBS, 8521 differentially expressed genes were found under a threshold with absolute fold changes > 2 and $p$ values < 0.05 after cells treated with mt-NP$^{Bodipy}$ + L. Specifically, 4416 upregulated genes and 4105 downregulated genes were identified in CT26 cells treated with mt-NP$^{Bodipy}$ + L relative to cells treated with PBS (Fig. 5c). Gene Ontology (GO) enrichment analysis showed that the CT26 cells treated with mt-NP$^{Bodipy}$ + L mainly displayed influenced expressions of genes in the nucleus, cytoplasm, cytosol, nucleoplasm, and cytoskeleton in Cellular Component (Fig. 5d), and in the major pathways of cellular responses to DNA damage stimulus, DNA repair, DNA replication, membrane fusion, and double-strand break repair in Biological Process (Fig. 5e), protein binding, DNA binding, single-stranded DNA binding, double-stranded DNA binding, and damaged DNA binding in Molecular Function (Fig. 5f). It is well known that ROS can induce base oxidation and single strand breaks (SSBs) in DNA, which can be

**Table 3 | Cartesian coordinates of optimized T1 excited state calculated by the DFT, B3LYP/6-31 G(d), Gaussian 16 program**

| Atom | x | y | z |
|---|---|---|---|
| C | 4.07202 | 7.72591 | −0.19961 |
| N | 3.98943 | 6.36487 | −0.18438 |
| B | 2.83328 | 5.56693 | 0.47663 |
| N | 1.63012 | 6.54095 | 0.51914 |
| C | 1.75783 | 7.90346 | 0.55664 |
| C | 3.02181 | 8.51128 | 0.34678 |
| C | 5.29576 | 8.09974 | −0.8976 |
| C | 5.92071 | 6.9236 | −1.23039 |
| C | 5.12945 | 5.83097 | −0.77589 |
| C | 0.2933 | 6.20133 | 0.69331 |
| C | −0.43522 | 7.41945 | 0.81819 |
| C | 0.43083 | 8.47902 | 0.72447 |
| C | 5.76126 | 9.46569 | −1.29336 |
| C | 5.3867 | 4.45291 | −0.80516 |
| C | 0.02002 | 9.91729 | 0.67531 |
| C | −0.20038 | 4.88188 | 0.68656 |
| F | 2.51428 | 4.44698 | −0.31654 |
| F | 3.18788 | 5.14279 | 1.75678 |
| C | 6.50435 | 3.85411 | −1.33691 |
| C | 6.76721 | 2.44399 | −1.27767 |
| C | −1.5075 | 4.57597 | 0.98218 |
| C | −2.15568 | 3.29712 | 0.91107 |
| C | 7.78356 | 1.86154 | −2.06677 |
| C | 8.01741 | 0.49876 | −2.06971 |
| C | 7.25147 | −0.35502 | −1.25595 |
| C | 6.27504 | 0.22055 | −0.41649 |
| C | 6.03569 | 1.57446 | −0.43531 |
| C | −3.5059 | 3.20223 | 1.31512 |
| C | −4.21598 | 2.02466 | 1.22762 |
| C | −3.61585 | 0.86246 | 0.70733 |
| C | −2.26129 | 0.93007 | 0.32208 |
| C | −1.55361 | 2.11283 | 0.42578 |
| C | −0.66482 | −6.2942 | 4.02864 |
| C | −0.38839 | −4.95695 | 4.30315 |
| C | 0.63335 | −4.30001 | 3.62517 |
| C | 1.38729 | −4.96023 | 2.64385 |
| C | 1.10375 | −6.30868 | 2.3859 |
| C | 0.09235 | −6.9698 | 3.07429 |
| C | 2.50875 | −4.26551 | 1.96527 |
| C | 3.34321 | −3.39564 | 2.84119 |
| C | 2.76788 | −4.39186 | 0.63686 |
| C | 4.01817 | −3.85845 | 0.0422 |
| C | 1.79827 | −4.95197 | −0.34092 |
| C | 3.59839 | −2.06372 | 2.48935 |
| C | 4.375 | −1.24779 | 3.3039 |
| C | 4.91276 | −1.75032 | 4.48771 |
| C | 4.65091 | −3.06786 | 4.85801 |
| C | 3.86263 | −3.87918 | 4.04771 |
| C | 2.19675 | −5.92721 | −1.26445 |
| C | 1.31901 | −6.38033 | −2.24244 |
| C | 0.03922 | −5.8376 | −2.33837 |
| C | −0.3615 | −4.85335 | −1.4393 |
| C | 0.50485 | −4.42488 | −0.43847 |
| C | 5.27066 | −4.10064 | 0.62228 |

**Table 3 (continued) | Cartesian coordinates of optimized T1 excited state calculated by the DFT, B3LYP/6-31 G(d), Gaussian 16 program**

| Atom | x | y | z |
|---|---|---|---|
| C | 6.41133 | −3.4605 | 0.16022 |
| C | 6.31904 | −2.55693 | −0.90073 |
| C | 5.09535 | −2.39031 | −1.5567 |
| C | 3.96224 | −3.0402 | −1.09462 |
| N | 7.42587 | −1.74438 | −1.27008 |
| C | 8.69642 | −2.34454 | −1.46219 |
| C | 8.80195 | −3.52772 | −2.20587 |
| C | 10.02961 | −4.14112 | −2.38207 |
| C | 11.18871 | −3.57924 | −1.83443 |
| C | 11.09306 | −2.39834 | −1.09511 |
| C | 9.85117 | −1.79722 | −0.90352 |
| O | 12.34951 | −4.25102 | −2.07375 |
| C | 13.55735 | −3.72895 | −1.5366 |
| C | 14.66896 | −4.66648 | −1.95975 |
| O | 15.86686 | −4.13032 | −1.4165 |
| O | −0.92735 | −7.83826 | 0.19298 |
| C | −2.27313 | −7.84951 | 0.62256 |
| C | −3.01949 | −6.59595 | 0.21647 |
| O | −2.42467 | −5.5218 | 0.94252 |
| C | −2.82175 | −1.9639 | 1.51853 |
| C | −2.36294 | −3.26961 | 1.59121 |
| C | −2.90713 | −4.26075 | 0.76612 |
| C | −3.88359 | −3.91242 | −0.16859 |
| C | −4.3194 | −2.59339 | −0.25619 |
| C | −3.81662 | −1.60885 | 0.59638 |
| N | −4.37498 | −0.30355 | 0.56129 |
| C | −8.02696 | −0.85277 | 0.97912 |
| C | −6.6575 | −0.9199 | 1.19144 |
| C | −5.78336 | −0.20378 | 0.36926 |
| C | −6.30411 | 0.59291 | −0.65367 |
| C | −7.67365 | 0.64977 | −0.86342 |
| C | −10.53113 | 1.15399 | 1.81695 |
| C | −11.31168 | 1.36925 | 2.94788 |
| C | −12.43184 | 0.57568 | 3.18609 |
| C | −12.76212 | −0.436 | 2.28605 |
| C | −11.98805 | −0.64275 | 1.15042 |
| C | −8.87384 | −1.81366 | −2.58074 |
| C | −8.09903 | −2.23944 | −3.65334 |
| C | −8.15129 | −1.56067 | −4.86967 |
| C | −8.99944 | −0.46402 | −5.00854 |
| C | −9.78372 | −0.04646 | −3.93813 |
| C | −10.8675 | 0.15734 | 0.89118 |
| C | −8.56003 | −0.08842 | −0.06781 |
| C | −10.02461 | −0.06145 | −0.31357 |
| C | −9.71709 | −0.70118 | −2.70157 |
| C | −10.54672 | −0.23646 | −1.55734 |
| C | −14.04567 | −0.66704 | −2.9394 |
| C | −12.70585 | −0.89407 | −2.64317 |
| C | −11.97644 | 0.02023 | −1.87116 |
| C | −12.61821 | 1.18757 | −1.43679 |
| C | −13.95379 | 1.42078 | −1.74225 |
| C | −14.67578 | 0.49094 | −2.48833 |
| C | 3.27887 | 9.90614 | 0.79094 |

**Table 3 (continued) | Cartesian coordinates of optimized T1 excited state calculated by the DFT, B3LYP/6-31 G(d), Gaussian 16 program**

| Atom | x | y | z |
|---|---|---|---|
| F | 4.56657 | 10.09374 | 1.16521 |
| F | 2.52959 | 10.24778 | 1.86547 |
| F | 3.02669 | 10.86679 | −0.15607 |
| H | 6.85017 | 6.83605 | −1.77445 |
| H | −1.50972 | 7.49711 | 0.89543 |
| H | 6.47138 | 9.38879 | −2.11908 |
| H | 4.93295 | 10.09807 | −1.61784 |
| H | 6.25481 | 9.98791 | −0.47179 |
| H | 4.61725 | 3.82507 | −0.37881 |
| H | 0.56296 | 10.46928 | −0.09377 |
| H | −1.04642 | 9.98759 | 0.45217 |
| H | 0.20087 | 10.4312 | 1.62094 |
| H | 0.49807 | 4.10023 | 0.41856 |
| H | 7.24137 | 4.46527 | −1.85201 |
| H | −2.15322 | 5.38955 | 1.30231 |
| H | 8.37306 | 2.49894 | −2.71885 |
| H | 8.7799 | 0.07959 | −2.71478 |
| H | 5.71363 | −0.41385 | 0.25698 |
| H | 5.2911 | 1.97895 | 0.23958 |
| H | −3.99692 | 4.08493 | 1.71319 |
| H | −5.24768 | 1.99066 | 1.55337 |
| H | −1.78004 | 0.05017 | −0.08729 |
| H | −0.52203 | 2.1327 | 0.09246 |
| H | −1.45674 | −6.80899 | 4.56191 |
| H | −0.96758 | −4.42359 | 5.04987 |
| H | 0.85652 | −3.26438 | 3.85599 |
| H | 1.6982 | −6.8459 | 1.65642 |
| H | −0.1019 | −8.01351 | 2.85672 |
| H | 3.18609 | −1.67302 | 1.56633 |
| H | 4.55686 | −0.2169 | 3.01818 |
| H | 5.5225 | −1.11563 | 5.12183 |
| H | 5.0589 | −3.46388 | 5.78224 |
| H | 3.65262 | −4.90158 | 4.34323 |
| H | 3.20365 | −6.32731 | −1.20954 |
| H | 1.63464 | −7.15389 | −2.93424 |
| H | −0.64206 | −6.18376 | −3.10807 |
| H | −1.34872 | −4.41153 | −1.51278 |
| H | 0.18668 | −3.66169 | 0.26177 |
| H | 5.33803 | −4.75478 | 1.48295 |
| H | 7.36611 | −3.61649 | 0.64882 |
| H | 5.03546 | −1.71001 | −2.39834 |
| H | 3.01113 | −2.88168 | −1.59013 |
| H | 7.90987 | −3.96267 | −2.64122 |
| H | 10.11995 | −5.05649 | −2.95526 |
| H | 11.96916 | −1.94577 | −0.64978 |
| H | 9.78166 | −0.89187 | −0.31241 |
| H | 13.75307 | −2.72224 | −1.92216 |
| H | 13.50904 | −3.67904 | −0.44322 |
| H | 14.45863 | −5.67332 | −1.57617 |
| H | 14.70167 | −4.71711 | −3.05587 |
| H | 16.60104 | −4.70093 | −1.66359 |
| H | −0.53517 | −6.99944 | 0.47124 |
| H | −2.74397 | −8.71939 | 0.15857 |

**Table 3 (continued) | Cartesian coordinates of optimized T1 excited state calculated by the DFT, B3LYP/6-31 G(d), Gaussian 16 program**

| Atom | x | y | z |
|---|---|---|---|
| H | −2.34646 | −7.96173 | 1.71334 |
| H | −2.91142 | −6.42968 | −0.8607 |
| H | −4.08572 | −6.67208 | 0.46464 |
| H | −2.42192 | −1.21488 | 2.19192 |
| H | −1.60534 | −3.55965 | 2.30828 |
| H | −4.3128 | −4.65218 | −0.8314 |
| H | −5.08221 | −2.33075 | −0.9793 |
| H | −8.69814 | −1.40597 | 1.62669 |
| H | −6.25311 | −1.53213 | 1.98927 |
| H | −5.62801 | 1.15874 | −1.28403 |
| H | −8.0654 | 1.25436 | −1.67226 |
| H | −9.65466 | 1.7675 | 1.63893 |
| H | −11.04337 | 2.1555 | 3.64595 |
| H | −13.03839 | 0.73896 | 4.07067 |
| H | −13.62529 | −1.06722 | 2.46986 |
| H | −12.25163 | −1.42673 | 0.45067 |
| H | −8.82679 | −2.34053 | −1.63509 |
| H | −7.45555 | −3.10626 | −3.5416 |
| H | −7.54293 | −1.8896 | −5.70555 |
| H | −9.05251 | 0.06606 | −5.95382 |
| H | −10.44999 | 0.80159 | −4.0524 |
| H | −14.59779 | −1.39348 | −3.52674 |
| H | −12.21556 | −1.79118 | −3.00495 |
| H | −12.06225 | 1.9111 | −0.8527 |
| H | −14.43217 | 2.33191 | −1.39835 |
| H | −15.71928 | 0.67181 | −2.72387 |

repaired through base excision/single strand breaks (BER/SSBR). If left unrepaired, damage caused by ROS can impede DNA replication and transcription, ultimately resulting in genomic instability, mutations, and the promotion of tumor development. Therefore, this is consistent with the experimental results that mt-NP$^{Bodipy}$ + L exhibited predominant localization within the cell membranes and yielded a substantial quantity of ROS, capable of inducing DNA damage while also perturbing cell membrane function. What's more, the Kyoto Encyclopedia of Genes and Genomes (KEGG) and Gene set enrichment analysis (GSEA) showed that mt-NP$^{Bodipy}$ + L substantially affected the cell cycle, DNA replication, homologous recombination, as well as a series of immune and inflammatory-related signaling pathways, including mitogen-activated protein kinase (MAPK) signaling pathway, forkhead box O (FoxO) signaling pathway, NF-κB, Toll-like receptor, chemical carcinogenesis-DNA adducts, cytokine-cytokine receptor interaction, IL-17 and antigen processing and presentation signaling pathways (Fig. 5g, h, and Supplementary Fig. 19).

**In vivo evaluation of mt-NP$^{Bodipy}$ therapeutics**
Subsequently, the efficacy of mt-NP$^{Bodipy}$ + L in inhibiting tumor growth was assessed using a CT26 tumor-bearing mouse model. This involved a single therapeutic injection administered on day 0, which was followed by 5 minutes of 808 nm irradiation on the same day. (Fig. 6a). Initially, mice were administered mt-NP$^{Bodipy}$ intravenously for bioimaging purposes at various intervals from 0 to 24 hours (Fig. 6b-e). The result indicated a continuous rise in fluorescence intensity within the tumors post-injection, peaking at 7 hours. After completing the **in vivo** bioimaging, the mice were euthanized, and the tumors, along with major organs such as the heart, liver, lungs, kidneys, and spleen,

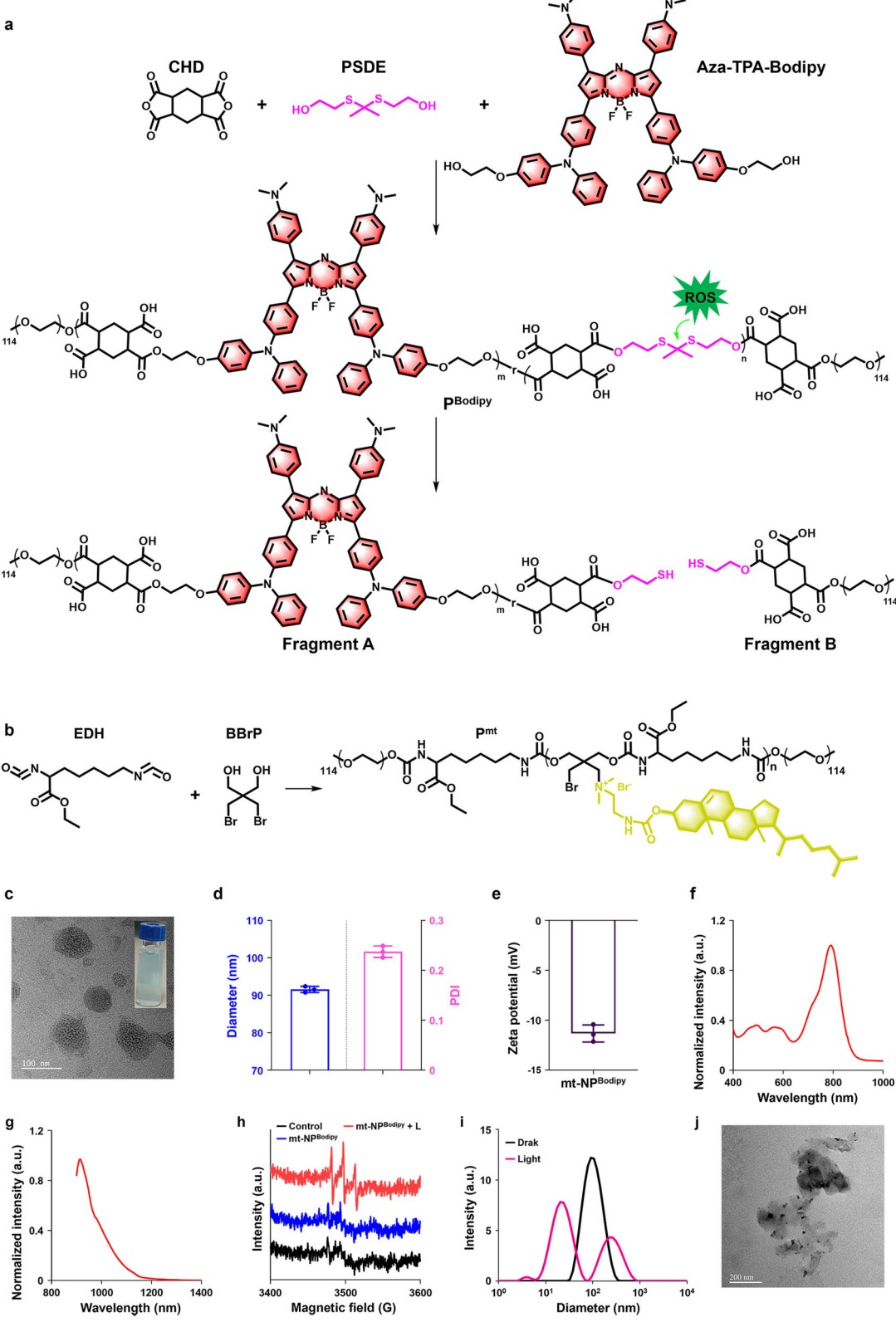

**Fig. 3 | Preparation and characterization of mt-NP$^{Bodipy}$. a** Synthetic route of P$^{Bodipy}$. Reagents and conditions: Dry DMF, rt, 48 h; mPEG$_{5000}$OH, rt, 24 h. **b** Synthetic route of P$^{mt}$. Reagents and conditions: Dry DMF, rt, 48 h; mPEG$_{5000}$OH, rt, 24 h. DMF/CH$_3$CN, N,N-dimethyl-chol, 80 °C, 24 h. **c** The representative TEM images of mt-NP$^{Bodipy}$ (n = 3 independent experiments). **d** Hydrodynamic diameters and PDI of mt-NP$^{Bodipy}$ (n = 3 independent experiments). **e** Zeta potential of mt-NP$^{Bodipy}$ (n = 3 independent experiments). **f** Absorption spectra of mt-NP$^{Bodipy}$. **g** PL spectra of mt-NP$^{Bodipy}$. **h** ESR spectra of mt-NP$^{Bodipy}$ after NIR light irradiation with TEMP as the trap. **i** Changes in hydrodynamic diameters of mt-NP$^{Bodipy}$ after NIR light irradiation. **j** The representative TEM image of mt-NP$^{Bodipy}$ after NIR light irradiation for 10 min (n = 3 independent experiments). Data are presented as mean ± SD. Source data are provided as a Source Data file.

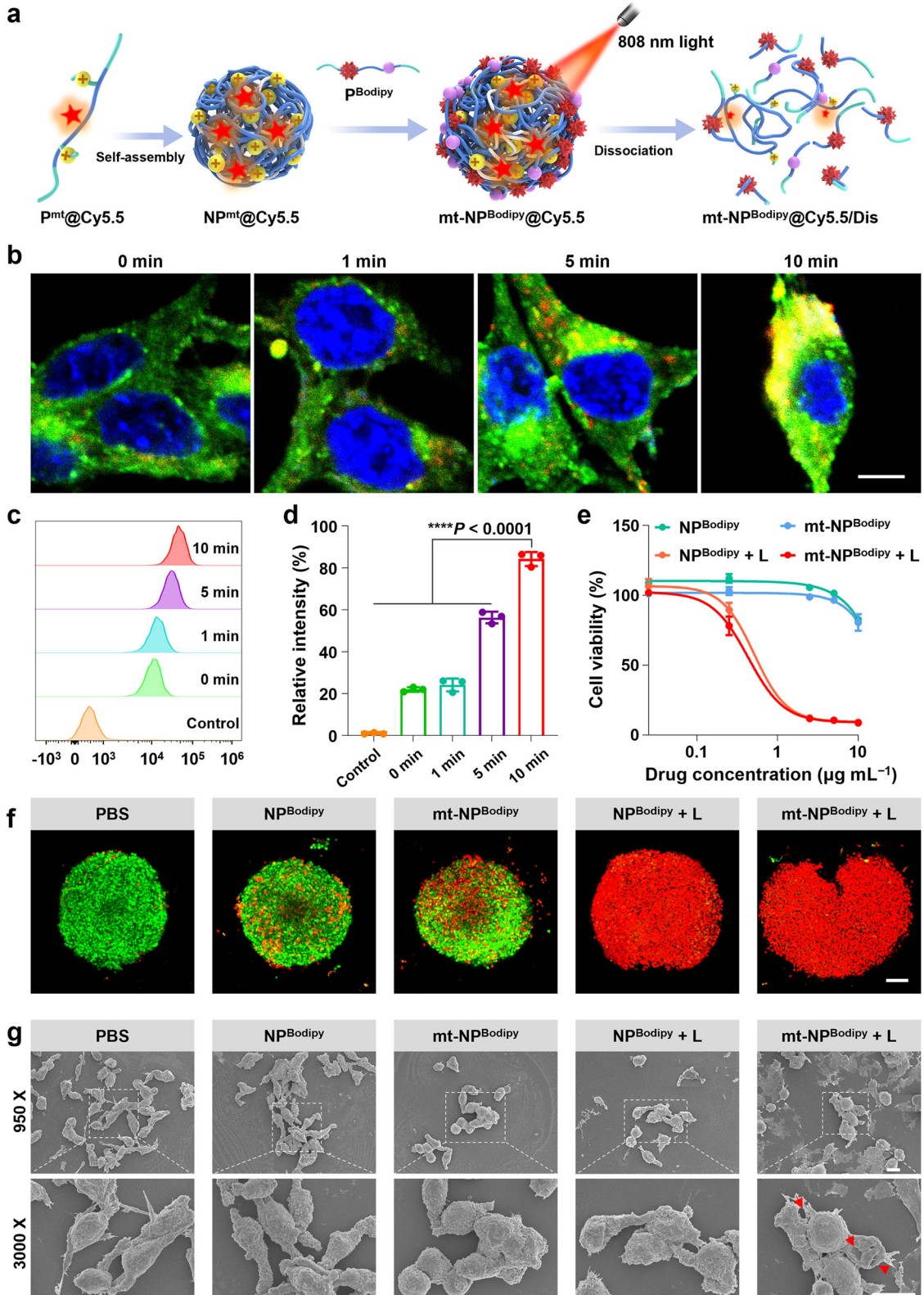

**Fig. 4 | In vitro evaluation of mt-NP^Bodipy. a** Schematic mechanism of mt-NP^Bodipy + L. **b** CLSM images of CT26 incubated with mt-NP^Bodipy@Cy5.5. Overlay images show the corresponding localization of mt-NP^Bodipy@Cy5.5 in the cells. DiO (green) is used to stain the cell membrane. Colocalization of mt-NP^Bodipy@Cy5.5 and DiO is indicated in yellow. The scale bar is 10 μm. **c** FCM assay and **d** quantification of CT26 incubated with mt-NP^Bodipy@Cy5.5 (n = 3 experimental replicates). Data were analyzed by one-way ANOVA with Bonferroni multiple comparisons post-test. *$p < 0.05$, **$p < 0.01$, ***$p < 0.001$, ****$p < 0.0001$. **e** Drug-response curves upon treatment of CT26 cells with mt-NP^Bodipy + L (n = 4 experimental replicates). **f** CLSM images of the 3D tumor spheroids of CT26 cells stained with calcein-AM (green, cells alive) and PI (red, cells dead) treated with PBS, NP^Bodipy, mt-NP^Bodipy, NP^Bodipy + L, and mt-NP^Bodipy + L (10 μg mL⁻¹) at 12 h. The scale bar is 100 μm. **g** SEM images of CT26 cells after treatment with PBS, NP^Bodipy, mt-NP^Bodipy, NP^Bodipy + L, and mt-NP^Bodipy + L (10 μg mL⁻¹) at 12 h. The scale bar is 10 μm. For (**b**, **f**, **g**), experiment was repeated three times independently with similar results. For **d** and **e**, data are presented as mean ± SD. Source data are provided as a Source Data file.

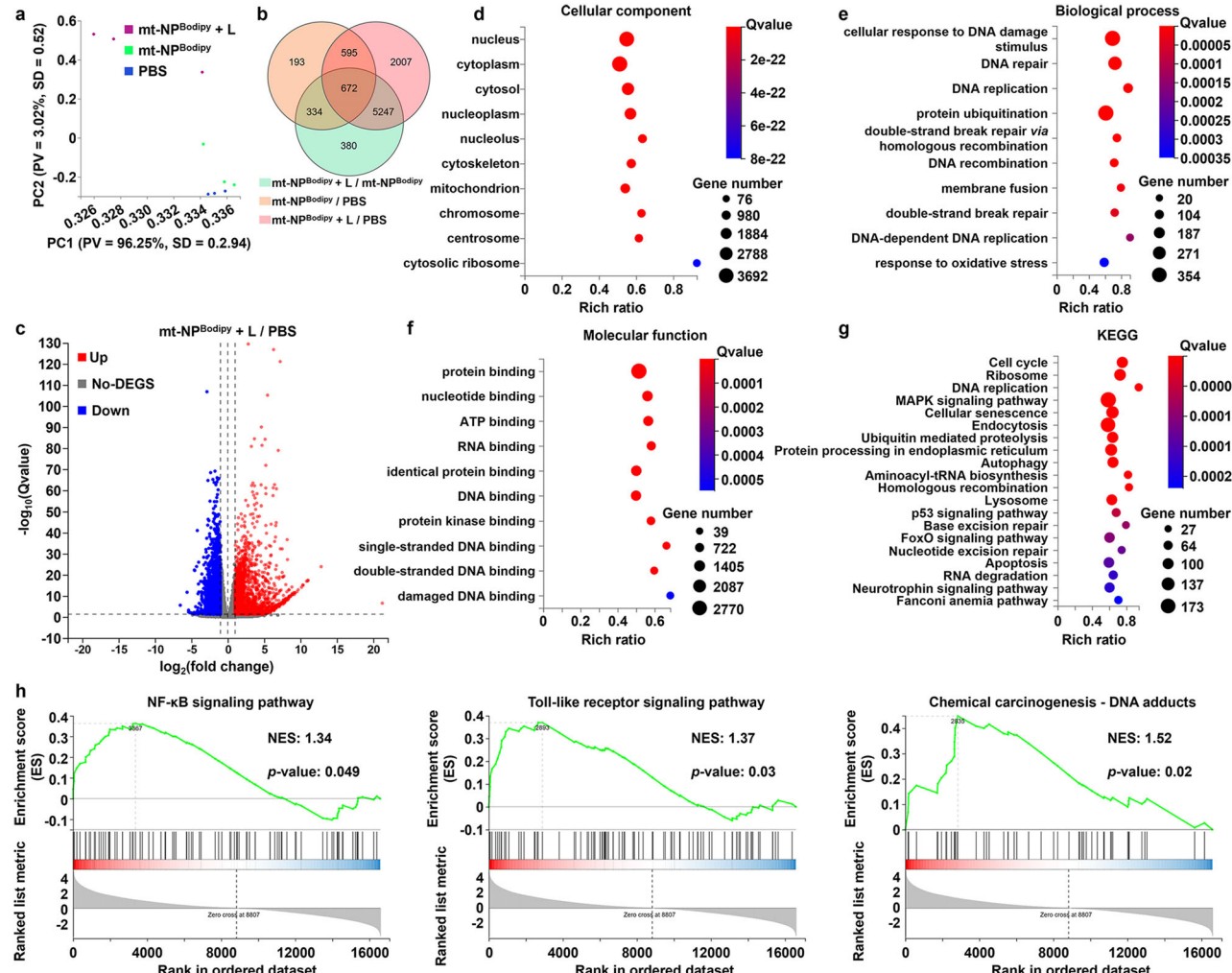

**Fig. 5 | Transcriptome analysis of CT26 cells treated with mt-NP^Bodipy + L. a** PCA score plot of the expressed genes of CT26 cells treated with PBS, mt-NP^Bodipy, and mt-NP^Bodipy + L. **b** Venn diagram of the identified differentially expressed genes. **c** Compared to PBS, Volcano plot showing 8521 differentially expressed genes (16592 total genes) from the cells treated with mt-NP^Bodipy + L. 4416 upregulated genes and 4105 downregulated genes were differentially expressed in the cells treated with mt-NP^Bodipy + L. **d–f** GO categorization of Cellular Component (**d**), Biological Process (**e**), and Molecular Function (**f**) after mt-NP^Bodipy + L treatment. **g** KEGG enrichment analysis of differentially expressed genes after mt-NP^Bodipy + L treatment. **h** GSEA reveals positive enrichment of genes altered in mt-NP^Bodipy + L treatment cells (data were analyzed by utilizing the GSEA software package without any modifications).

were analyzed **ex vivo** to determine biodistribution. The results showed that the fluorescence intensity in the tumors was 2.3, 1.6, 1.8, and 1.6 times greater than in the heart, spleen, lungs, and kidneys, respectively, demonstrating a pronounced tumor-targeting capability of mt-NP^Bodipy.

The therapeutic efficacy of mt-NP^Bodipy + L was then investigated using the same model. The tumor volume of mice receiving different treatments was measured over a span of 12 days (Fig. 6f and Supplementary Fig. 20). The tumor growth rate of mice treated with NP^Bodipy and mt-NP^Bodipy showed no difference compared to mice treated with PBS, suggesting that NP^Bodipy and mt-NP^Bodipy did not exhibit any significant antitumor effects. Moreover, mice treated with NP^Bodipy + L exhibited a partial inhibition of tumor growth with a 68% tumor inhibition rate. In contrast, the tumor inhibition rate for mt-NP^Bodipy + L exceeded 90%, indicating that mt-NP^Bodipy + L substantially suppressed tumor progression. These findings suggest that cationic polymers also demonstrate antineoplastic activity by disrupting cell membranes in vivo. Furthermore, all the animals did not show any indications of pain or stress, and their body weight remained stable (Fig. 6g). Finally, the tumorous tissue underwent further histological analysis using hematoxylin and eosin (H&E) staining and Ki67 staining

(Supplementary Fig. 21). Compared to animals treated with PBS, NP^Bodipy, or mt-NP^Bodipy, tumor tissues from mice receiving mt-NP^Bodipy + L showed significant cellular damage, characterized by nuclear fragmentation and nucleolysis. These results demonstrate the superior ability of mt-NP^Bodipy + L to induce tumor remission.

The therapeutic efficacy of mt-NP^Bodipy + L for photo-immunotherapy was further evaluated using FCM. As shown in Fig. 6h, i, both NP^Bodipy + L and mt-NP^Bodipy + L effectively increased the proportion of mature DCs in tumors, thereby enhancing antigen presentation and DCs maturation[45]. Specifically, mt-NP^Bodipy + L led to a 2.2-fold and 2.0-fold increase in the proportion of mature DCs, compared to NP^Bodipy and mt-NP^Bodipy, respectively. FCM was also employed to analyze the maturation of DCs in the LNDs of mice administered mt-NP^Bodipy + L (Fig. 6j, k). The findings revealed that the percentage of mature DCs in the LNDs of mice treated with mt-NP^Bodipy + L (50.6%) was almost twice as high as that in mice treated with mt-NP^Bodipy (24.9%). Consequently, we analyzed the relative population of CD8^+ T cells within the tumor tissues (Fig. 6l, m). The percentage of CD8^+ T cells (41.0%) in tumors from mice receiving mt-NP^Bodipy + L treatment was 1.5 times greater compared to those treated with NP^Bodipy (26.8%) and mt-NP^Bodipy (27.3%), indicating that mt-NP^Bodipy + L not only activated T cells

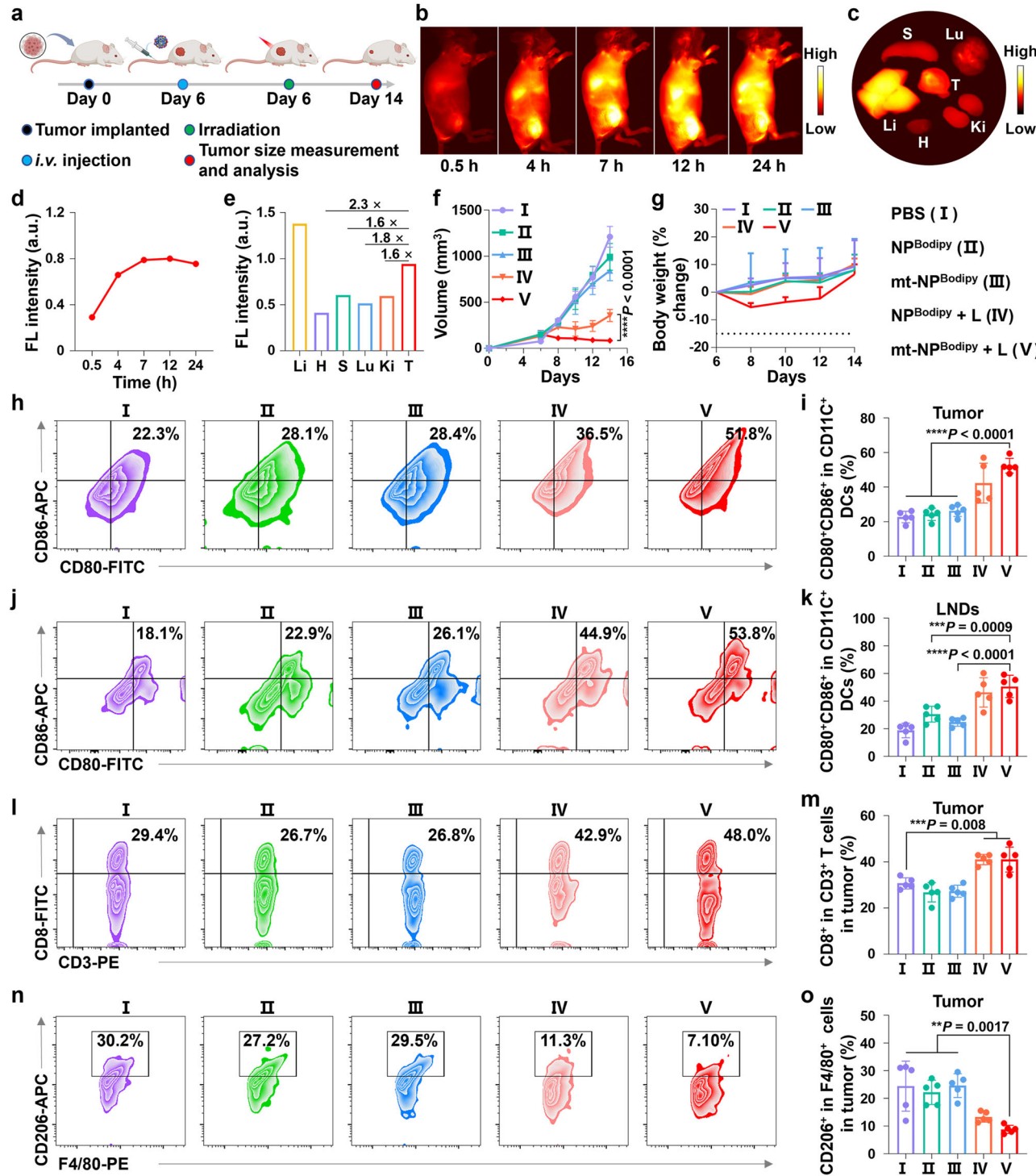

**Fig. 6 | Imaging and photo-immunotherapy properties of mt-NP^Bodipy + L were evaluated in a CT26 tumor-bearing BALB/c mice model. a** Schedule diagram in a CT26 tumor-bearing BALB/c mice model treated with mt-NP^Bodipy + L. Figure created with Biorender.com. **b** NIR-II fluorescence bioimaging of mice injected with mt-NP^Bodipy after various time points in vivo. **c** NIR-II fluorescence imaging of major tissues and organs (S, spleen; H, heart; Lu, lung; Ki, kidney; L, liver; T, tumor) after sacrificing the mice at 24 h. **d** Semi-quantitative NIR-II fluorescence analysis in the tumor sites at different times. **e** Semi-quantitative NIR-II fluorescence analysis of organs after 24 h (S, spleen; H, heart; Lu, lung; Ki, kidney; L, liver; T, tumor). **f** Comparison of tumor growth inhibition curves. **g** Monitoring of the weight change of the animal model. **h** FCM plot of CD80+CD86+ dendritic cells gated on

CD11c+ cells in the tumorous tissues. **i** Quantification of (**h**). **j** FCM plot of CD80+CD86+ dendritic cells gated on CD11c+ cells in the lymph nodes. **k** Quantification of (**j**). **l** FCM plot of CD8+ gated on CD3+ cells in the tumorous tissues. **m** Quantification of (**l**). **n** FCM plot of M2 macrophages (CD206+ gated on F4/80+ cells) in the tumorous tissues. **o** Quantification of (**n**). Data were expressed as means ± SD. n = 5 biologically independent mice for each group. Data were analyzed by one-way ANOVA with Bonferroni multiple comparisons post-test (**i, k, m, o**), and two-way ANOVA with Bonferroni multiple comparisons post-test (**f**). *p < 0.05, **p < 0.01, ***p < 0.001, ****p < 0.0001. Source data are provided as a Source Data file.

but also increased the infiltration of CD8$^+$ T cells within the tumor. Moreover, photo-immunotherapy demonstrated the ability to reprogram the immunosuppressive tumor microenvironment (TME) by transforming macrophages with an immunosuppressive phenotype (M2) into those with an immunostimulatory phenotype (M1), which resulted in the release of antitumor cytokines[46]. To verify the changes in TME, we evaluated the M2 phenotype macrophages in tumor-associated macrophages (TAMs) within tumor tissues (Fig. 6n, o). The results demonstrated that the percentage of M2 polarized macrophages (CD206$^+$ gated on F4/80$^+$ cells) in mice treated with mt-NP$^{Bodipy}$ + L (8.8%) was lower than in those treated with NP$^{Bodipy}$ (22.2%) and mt-NP$^{Bodipy}$ (24.6%). These findings indicate that mt-NP$^{Bodipy}$ + L effectively reprogrammed the immunosuppressive TME. In conclusion, mt-NP$^{Bodipy}$ + L has the potential to activate systemic antitumor immunity and reverse the immunosuppressive TEM of the tumor.

To further investigate the immunostimulatory effect of mt-NP$^{Bodipy}$ + L, a bilateral tumor model was established (Fig. 7a). The primary tumors were treated as described previously. The results indicated that PBS and mt-NP$^{Bodipy}$ alone had no significant impact on inhibiting either the primary or distant tumors (Fig. 7b, c, and Supplementary Fig. 22). Notably, mt-NP$^{Bodipy}$ + L not only eradicated the primary tumors but also significantly suppressed the growth of distant tumors. To understand the therapeutic mechanism of mt-NP$^{Bodipy}$ + L, immune cells in distant tumors were assessed 18 days after the primary tumor inoculation. FCM results showed a significant increase in the proportion of mature DCs in the lymph nodes of mice, which could promote antigen presentation and DC maturation. As shown in Fig. 7d, e, mt-NP$^{Bodipy}$ + L led to a 1.3-fold increase in the proportion of mature DCs compared to both mt-NP$^{Bodipy}$ and PBS. The FCM results also demonstrated a significant increase in the number of CD8$^+$ T cells in the distant tumors of mice treated with mt-NP$^{Bodipy}$ + L (Fig. 7f, g), compared to mice treated with mt-NP$^{Bodipy}$ and PBS. Additionally, immunofluorescent staining revealed that the population of CD8$^+$ T cells increased in both primary and distant tumors of mice treated with mt-NP$^{Bodipy}$ + L (Supplementary Fig. 23).

In addition to immune activation, the efficacy of mt-NP$^{Bodipy}$ + L in inducing immune memory was examined. Specifically, a secondary tumor was inoculated 60 days after the initial treatment (Fig. 7h). Predictably, all naïve mice exhibited rapid tumor progression following the challenge with CT26 tumor cells (Fig. 7i and Supplementary Fig. 24). Contrarily, mice that had their initial tumors eradicated showed no signs of secondary tumor growth, and all the animals survived for 100 days (Fig. 7j), showcasing impressive immune memory responses that ensured prolonged protection. To further confirm the immune memory responses induced by mt-NP$^{Bodipy}$ + L, the LNDs of mice were collected on day 76 to assess the population of memory T cells. Using FCM on these LNDs, we observed increased percentages of T$_{EM}$ and central memory T cells (T$_{CM}$) among the CD3$^+$CD8$^+$ T cells in mice that survived to 76 days, compared with those in the naïve mice (Fig. 7k, l). It was also observed that mt-NP$^{Bodipy}$ + L resulted in a 1.6-fold and a 4.4-fold increase in pro-inflammatory cytokines in the circulation, such as TNF-α and IFN-γ (Fig. 7m, n). Collectively, mt-NP$^{Bodipy}$ + L markedly activated antitumor immunity and induced immune memory that prevent tumor recurrence.

In summary, we demonstrated NIR light activation of nanoparticles mt-NP$^{Bodipy}$ for destabilizing the cell membranes for NIR-II fluorescence bioimaging, PDT, and photo-immunotherapy. **In vitro**, mt-NP$^{Bodipy}$ + L could efficiently produce $^1O_2$, thereby inducing rapid dissociation of mt-NP$^{Bodipy}$ and release of P$^{mt}$ to disintegrate the cell membranes. The transcriptome landscape of the CT26 cells treated by mt-NP$^{Bodipy}$ + L displayed influenced expressions of genes in the nucleus, cytoplasm, cytosol, nucleoplasm, and cytoskeleton in Cellular Component, and in the major pathways of cellular responses to DNA damage stimulus, DNA repair, DNA replication, membrane fusion, and double-strand break repair in Biological Process, protein binding, DNA binding, single-stranded DNA binding, double-stranded DNA binding, and damaged DNA binding in Molecular Function. Moreover, mt-NP$^{Bodipy}$ + L significantly influenced the cell cycle, DNA replication, homologous recombination, and various immune and inflammatory-related signaling pathways. These pathways include the MAPK signaling pathway, NF-κB, Toll-like receptor, chemical carcinogenesis-DNA adducts, cytokine-cytokine receptor interaction, IL-17, FoxO signaling pathway, and antigen processing and presentation signaling pathways. Moreover, we found that mt-NP$^{Bodipy}$ + L could effectively eradicate CT26 cells. Mechanistic investigations uncovered that mt-NP$^{Bodipy}$ + L primarily induced swift necrosis of cancer cells by causing membrane lysis, occurring in mere minutes. **In vivo**, mt-NP$^{Bodipy}$ demonstrated outstanding NIR-II fluorescence signals suitable for NIR-II bioimaging. Furthermore, mt-NP$^{Bodipy}$ + L was highly effective in suppressing CT26 tumor growth without noticeable side effects. Additionally, mt-NP$^{Bodipy}$ + L was capable of recruiting DCs, enhancing the presence of antigen-specific CTLs within the TME, and reprogramming the immunosuppressive TME, thereby triggering antitumor immune responses. This work exemplified the strategy of destabilizing the cell membranes for NIR-II fluorescence bioimaging, PDT, and photo-immunotherapy.

## Methods

### Ethical statement

BALB/c mice (SPF Biotechnology Co., Ltd. (Beijing, China), 4–5 weeks old, female) were used. Mice were housed in 12 h light/12 h dark cycle with the temperature maintained between 65-75 °F ( ~18-23 °C), ~50% humidity. Tumor size was measured by a digital caliper at indicated times and calculated according to the formula = 0.5 × length × width$^2$. Mice were euthanized when the tumor size exceeded 1500 mm$^3$ or when the animals became moribund, exhibiting severe weight loss, extreme weakness, or unhealing ulceration. The animals were maintained under pathogen-free conditions and all animal experiments were approved by the Peking University Institutional Animal Care and Use Committee (LA2021316).

### Materials

Unless otherwise noted, all chemicals and reagents were obtained commercially and used without further purification. 4-(Phenylamino) phenol, (2-bromoethoxy)(tert-butyl)dimethylsilane, 4-(dimethylamino)benzaldehyde, 1-(4-bromophenyl)ethanone, acetic acid glacial, diethylamine, nitromethane, ammonium acetate, trifluoride diethyl etherate, tri-tert-butylphosphine, sodium tert-butoxide, and pyridine hydrofluoride were purchased from Energy Chemical. Piperidine was purchased from J&K Scientific Ltd. RPMI 1640 medium and penicillin/streptomycin (P/S) was purchased from Procell Life Science & Technology Co., Ltd. DiO was purchased from Beijing Solarbio Science & Technology Co., Ltd. Fetal bovine serum (FBS), 2-(4-amidinophenyl) −1H-indole-6-carboxamidine (DAPI), Calcein-AM, and propidium iodide (PI) was purchased from Beyotime. MTT was purchased from Aladdin. ELISA kits for the mouse cytokines TNF-α (ab208348) and IFN-γ (ab252363) were purchased from Abcam. Anti-mouse antibodies anti-CD3-PE (ab22268, 1:200 dilution), anti-CD4-APC (ab252152, 1:200 dilution), anti-CD8-FITC (ab237367, 1:200 dilution), anti-CD11c-PE (ab210309, 1:200 dilution), anti-CD80-FITC (ab18279, 1:200 dilution), anti-CD86-APC (ab218757, 1:200 dilution), anti-F4/80-PE (ab105156, 1:200 dilution), anti-CD62L-APC (ab41459, 1:200 dilution), and anti-CD44-PC5.5 (ab234445, 1:200 dilution) were purchased from Abcam. Anti-mouse antibodies anti-CD206-FITC (E-AB-F1135C, 1:200 dilution) were obtained from Elabscience.

### Material characterization

$^1$H NMR spectra were conducted by a Bruker Fourier 300 and Bruker Avance 400 NMR spectrometer (Bruker, USA) at room temperature. Flow cytometric analysis was performed using a flow cytometer (BD Biosciences, USA). TEM images were captured using a JEM-2200FS

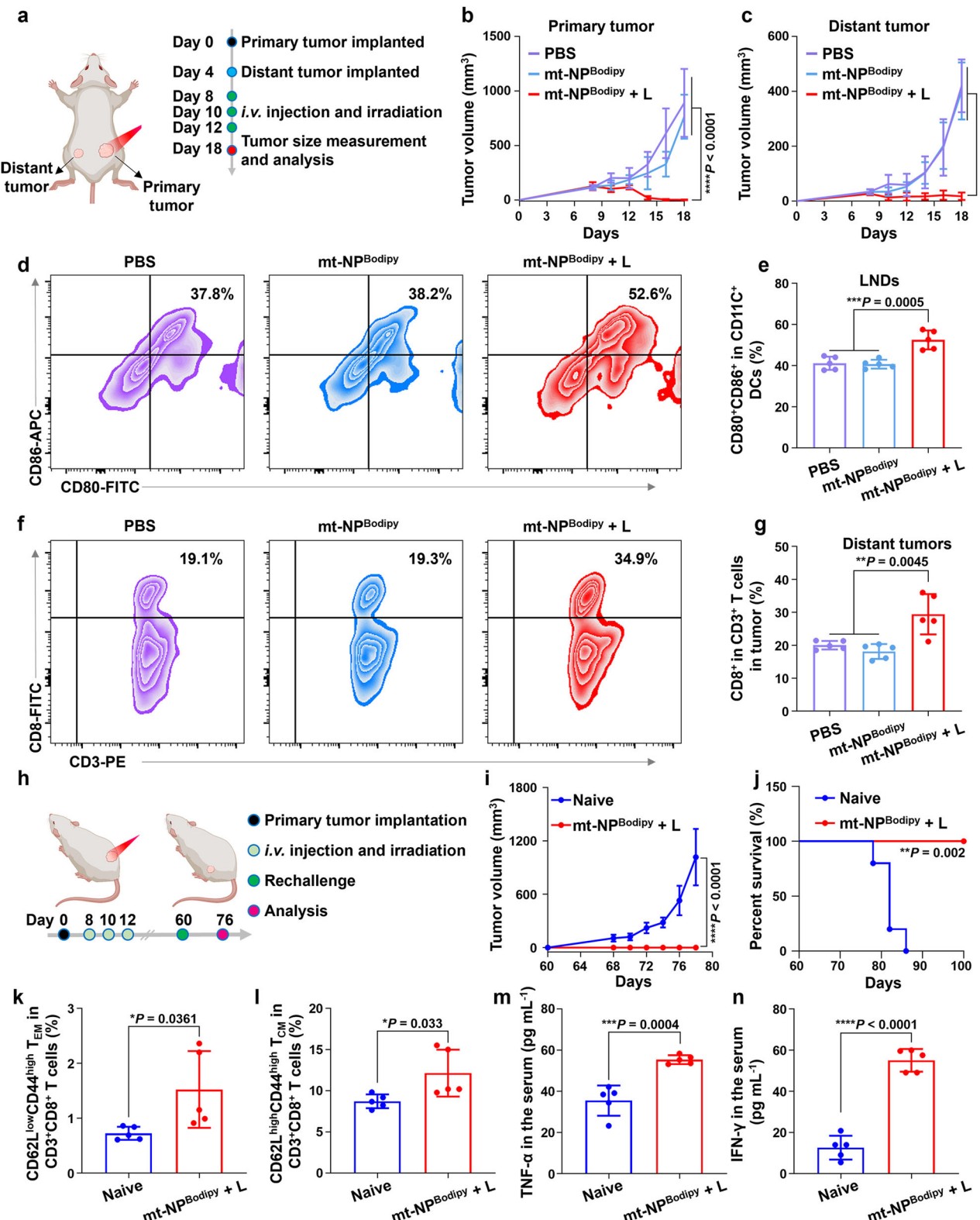

**Fig. 7 | mt-NP^Bodipy + L eliminates distant tumors and induces long-term immune memory effects. a** Schematic treatment schedule of mt-NP^Bodipy + L eliminates distant tumors in a CT26 tumor-bearing BALB/c mice model. Figure created with Biorender.com. **b** Comparison of tumor growth inhibition curves of primary tumors (n = 7). **c** Comparison of tumor growth inhibition curves of distant tumors (n = 7). **d** FCM plot of CD80⁺CD86⁺ dendritic cells gated on CD11c⁺ cells in the lymph nodes. **e** Quantification of (**d**). **f** FCM plot of CD8⁺ gated on CD3⁺ cells in the distant tumors. **g** Quantification of (**f**). **h** Schematic treatment schedule of mt-NP^Bodipy + L induce long-term immune memory effects in a CT26 tumor-bearing BALB/c mice model. Figure created with Biorender.com. **i** Average tumor growth curves of the

treated mice (n = 5). **j** survival curves of the treated mice (n = 5). **k** Relative quantification of effector memory T cells ($T_{em}$, CD62L⁻CD44⁺) subset from CD8⁺ T cells in the lymph nodes. **l** Relative quantification of central memory T cells ($T_{cm}$, CD62L⁺CD44⁺) subset from CD8⁺ T cells in the lymph nodes. **m, n** Cytokine levels of TNF-α (**m**) and IFN-γ (**n**) in the serum after tumor rechallenging. Data were expressed as means ± SD (**e, g, k–n**). n = 5 biologically independent mice for each group (**e, g, k–n**). Data were analyzed by two-way ANOVA with Bonferroni multiple comparisons post-test (**b, c**, and **i**), Log-rank (MantelCox) test) (**j**), and two-tailed unpaired t-test (**e, g, k–n**). *$p < 0.05$, **$p < 0.01$, ***$p < 0.001$, ****$p < 0.0001$. Source data are provided as a Source Data file.

transmission electron microscope (JEOL, Japan). The size, Zeta potential, and polydispersity index of the nanoparticles were measured by dynamic light scattering on an ALV/CGS-3 goniometry system (ALV, Germany). The morphology of cancer cells was observed by a JSM-6700F Scanning Electron Microscopy (JEOL, Japan). The absorbance spectra were recorded with a TU-1901 spectrophotometer (Persee, China). Cell imaging was conducted with an LSM-800 CLSM (Carl Zeiss, Germany). Mice imaging was performed with a VanGogh IGS1000 NIR-II animal fluorescence imaging instrument (Niroptics, China). The mass spectrometry experiments are described in detail below: M5, M7, and Aza-TPA-Bodipy ($1 \times 10^{-4}$ M, n = 1 biologically independent sample) was dissolved in dichloromethane and data (m/z: 500-1500) was obtained with Bruker Autoflex III (Bruker, USA) MALDI-TOF-MS mass spectrometer using FlexControl 3.4 software (Bruker Daltonics). Data analysis was conducted with Prism 8 (GraphPad Software). N,N-dimethyl-chol ($1 \times 10^{-4}$ M, n = 1 biologically independent sample) was dissolved in methanol and data (m/z: 300-800) was obtained with LC/Xevo G2-XS TOFu (Waters, USA) ESI-MS mass spectrometer using Masslynx 4.0 software (Waters). Data analysis was conducted with Prism 8 (GraphPad Software).

## Cells and mice

The murine CT26 was cultured in a complete RPMI-1640 culture medium containing 10% FBS and 1% penicillin-streptomycin in an artificial environment (5% $CO_2$ at 37 °C). The cell experiments were conducted when the CT26 cancer cells were grown to 80% confluence in the culture dish. The CT26 cells lines were purchased from IMMO-CELL (IM-M007, Xiamen, Fujian, China). The cells were detected every two months to exclude mycoplasma.

## Synthesis of M3

A mixture of 4-(dimethylamino)benzaldehyde (7.5 g, 50.2 mmol), 1-(4-bromophenyl)ethanone (5.0 g, 25.1 mmol), acetic acid (500 μL), and piperidine (500 μL) in toluene (100 mL) was stirred at 110 °C for 48 h under a nitrogen atmosphere. The reaction mixture was then diluted with DCM (50 mL), washed with water (100 mL × 3), and saturated NaCl solution (30 mL × 3). The organic layer was dried over anhydrous $Na_2SO_4$, filtered, concentrated under vacuum, and purified by silica gel column chromatography to afford M3 (4.6 g, yield 56%) as a yellow solid. [1]H NMR (DMSO-$d_6$, 300 MHz) δ 8.07 (d, $J$ = 8.52 Hz, 2H), 7.76−7.64 (m, 6H), 6.76 (d, $J$ = 8.55 Hz, 2H), 3.01 (s, 6H).

## Synthesis of M4

A mixture of M3 (1.0 g, 3.0 mmol), diethylamine (5 mL), and nitromethane (20 mL) in EtOH (30 mL) was stirred at 70 °C for 48 h under a nitrogen atmosphere. The reaction mixture was then diluted with $CH_2Cl_2$ (50 mL), washed with water (100 mL × 3) and saturated NaCl solution (30 mL × 3). The organic layer was dried over anhydrous $Na_2SO_4$, filtered, concentrated under vacuum, and purified by silica gel column chromatography to afford M4 (1.0 g, yield 89%) as a light-yellow liquid. 1H NMR (DMSO-$d_6$, 300 MHz,) δ 7.87 (d, $J$ = 8.64 Hz, 2H), 7.74 (d, $J$ = 8.46 Hz, 2H), 7.16 (d, $J$ = 8.67 Hz, 2H), 6.64 (d, $J$ = 8.76 Hz, 2H), 4.94-4.71 (m, 2H), 3.96-3.86 (m, 1H), 3.47 (t, $J$ = 6.69 Hz, 2H), 2.83 (s, 6H).

## Synthesis of M5

A mixture of compound 4 (0.56 g, 1.4 mmol) and ammonium acetate (5.4 g, 70 mmol) in butanol (40 mL) was stirred at 120 °C for 24 h under a nitrogen atmosphere. The mixture was then cool down, and the precipitate was filtered, the solid was washed three times with petroleum ether to afford purple powder. In a 100 mL round bottom flask, the purple powder and diisopropylethylamine (10 mL) were dissolved in 10 mL of $CH_2Cl_2$ in a nitrogen atmosphere. Then trifluoride diethyl etherate (15 mL) was dropwise added, stirring at room temperature for 24 h. Then, the reaction mixture was diluted with $CH_2Cl_2$ (50 mL), washed with water (100 mL × 3), and saturated NaCl solution

(30 mL × 3). The organic layer was dried over anhydrous $Na_2SO_4$, filtered, concentrated under vacuum, and purified by silica gel column chromatography to afford M5 (0.17 g, yield 32%) as a purple powder. [1]H NMR (DMSO-$d_6$, 400 MHz) δ 8.17 (d, $J$ = 9.08 Hz, 4H), 7.95 (d, $J$ = 8.68 Hz, 4H), 7.74 (d, $J$ = 8.68 Hz, 4H), 7.25 (s, 2H), 6.90 (d, $J$ = 9.16 Hz, 4H), 3.09 (s, 12H). MALDI-TOF MS (m/z): calcd for $[C_{36}H_{30}BBr_2F_2N_5]^+$: 741.091; found, 741.376.

## Synthesis of M6

A mixture of 4-(phenylamino)phenol (1.0 g, 5.4 mmol), $K_2CO_3$ (1.5 g, 10.8 mmol), and (2-bromoethoxy)(tert-butyl)dimethylsilane (1.9 g, 8.1 mmol) in acetonitrile (20 mL) was stirred at 80 °C for 24 h under a nitrogen atmosphere. Then, the reaction mixture was diluted with EtOAc (50 mL), washed with water (100 mL × 3), and saturated NaCl solution (30 mL × 3). The organic layer was dried over anhydrous $Na_2SO_4$, filtered, concentrated under vacuum, and purified by silica gel column chromatography to afford M6 (1.3 g, yield 72%) as a white solid. [1]H NMR (DMSO-$d_6$, 400 MHz) δ 7.82 (s, 1H), 7.15 (t, $J$ = 7.68 Hz, 2H), 7.03 (d, $J$ = 8.80 Hz, 2H), 6.92 (d, $J$ = 7.88 Hz, 2H), 6.86 (d, $J$ = 8.80 Hz, 2H), 6.70 (t, $J$ = 7.24 Hz, 2H), 3.97 (t, $J$ = 4.36 Hz, 2H), 3.89 (t, $J$ = 4.64 Hz, 2H), 0.88 (s, 9H), 0.07 (s, 6H).

## Synthesis of M7

A mixture of M5 (50 mg, 0.068 mmol), M6 (68 mg, 0.20 mmol), $Pd_2(dba)_3$ (30 mg), tri-tert-butylphosphine (200 μL), and sodium tert-butoxide (10 mg, 0.10 mmol) in toluene (3 mL) was stirred at 85 °C for 24 h under a nitrogen atmosphere. The reaction mixture was then diluted with $CH_2Cl_2$ (50 mL), washed with water (50 mL × 3), and saturated NaCl solution (30 mL × 3). The organic layer was dried over anhydrous $Na_2SO_4$, filtered, concentrated under vacuum, and purified by silica gel column chromatography to afford M7 (30 mg, yield 35%) as a blue-black solid. [1]H NMR (DMSO-$d_6$, 300 MHz) δ 8.12 (d, $J$ = 8.88 Hz, 4H), 8.03 (d, $J$ = 8.94 Hz, 4H), 7.40 (t, $J$ = 7.92 Hz, 4H), 7.27 (s, 2H), 7.17-7.13 (m, 10H), 7.01 (d, $J$ = 8.91 Hz, 4H), 6.86-6.82 (m, 8H), 4.07 (t, $J$ = 4.11 Hz, 4H), 3.94 (t, $J$ = 4.65 Hz, 4H), 3.05 (s, 12H), 0.87 (s, 18H), 0.07 (s, 12H). MALDI-TOF MS (m/z): calcd for $[C_{76}H_{86}BF_2N_7O_4Si_2 + H]^+$: 1266.642; found, 1266.150.

## Synthesis of Aza-TPA-Bodipy

Pyridine hydrofluoride (200 μL) was slowly added to a mixture of M7 (80 mg, 0.063 mmol) in $CH_2Cl_2$ (10 mL). The mixture was stirred for 4 h at room temperature. Then, the mixture was diluted with $CH_2Cl_2$ (50 mL), and washed with saturated calcium chloride solution (50 mL × 3). The organic layer was dried over anhydrous $Na_2SO_4$, filtered, concentrated under vacuum, and purified by silica gel column chromatography to afford M7 (44 mg, yield 67%) as a blue-black solid. [1]H NMR (DMSO-$d_6$, 400 MHz) δ 8.11 (d, $J$ = 8.48 Hz, 4H), 8.02 (d, $J$ = 8.48 Hz, 4H), 7.40 (t, $J$ = 7.60 Hz, 4H), 7.26 (s, 2H), 7.18-7.14 (m, 10H), 7.02 (d, $J$ = 8.32 Hz, 4H), 6.86 (d, $J$ = 8.88 Hz, 8H), 4.88 (t, $J$ = 5.48 Hz, 2H), 4.02 (t, $J$ = 4.72 Hz, 4H), 3.75-3.72 (m, 4H), 3.30 (s, 12H). MALDI-TOF MS (m/z): calcd for $[C_{64}H_{58}BF_2N_7O_4]^+$: 1037.461; found, 1037.296.

## Synthesis of $P^{Bodipy}$

A mixture of Aza-TPA-Bodipy (40 mg, 0.039 mmol), PSDE (31 mg, 0.16 mmol), and CHD (47 mg, 0.21 mmol) in dry DMF (3 mL) was stirred at 50 °C for 24 h under a nitrogen atmosphere. After magnetic stirring for 24 h, mPEG$_{5000}$-OH (100 mg, 0.02 mmol) was added to the reaction mixture. After magnetic stirring for another 24 h, the mixture was added to 15 mL of deionized water under sonication, followed by dialysis in a dialysis bag (MWCO: 8000-14000 Da). After 72 h, the solution was freeze-dried under reduced pressure to afford 141 mg powder.

## Synthesis of N,N-dimethyl-chol

A mixture of N,N-dimethylethylenediamine (1.3 mL, 11.7 mmol) and $Et_3N$ (1.9 mL, 13.3 mmol) in $CH_2Cl_2$ (40 mL) was slowly added to

cholesterol chloroformate (5.0 g, 11.1 mmol). The solution was stirred for 18 h at room temperature. Then, the reaction mixture was concentrated under a vacuum. The residue was dissolved in 1:1 CH₂Cl₂:EtOAc (40:40 mL) and filtered through a plug of basic alumina, washing with additional CH₂Cl₂ and EtOAc. The filtrate was then washed with water (200 mL × 3) and the organic fraction was separated, dried over Na₂SO₄, filtered, and concentrated under vacuum to afford N,N-dimethyl-chol (4.6 g, 82%) as a white solid. ¹H NMR (300 MHz, CDCl₃) δ 5.38 (d, J = 5.13 Hz, 1H), 5.17 (s, 1H), 4.53-4.45 (m, 1H), 3.27 (d, J = 5.49 Hz, 2H), 2.44 (t, J = 5.94 Hz, 2H), 2.35-2.31 (m, 2H), 2.24 (s, 6H), 2.03-1.98 (m, 2H), 1.94-1.77 (m, 3H), 1.60-1.41 (m, 7H), 1.35-1.24 (m, 4H), 1.17-0.98 (m, 12H), 0.92 (d, J = 6.48 Hz, 3H), 0.88 (dd, J = 1.05 Hz, J = 6.57 Hz, 6H), 0.67 (s, 3H). ESI MS (m/z): calcd for $[C_{32}H_{56}N_2O_2 + H]^+$: 501.44; found, 501.45; $[C_{32}H_{56}N_2O_2 + Na]^+$: 523.42; found, 523.45.

## Synthesis of P^Br

A mixture of EDH (552 mg, 2.3 mmol) and BBrP (560 mg, 2.2 mmol) in dry DMF (3 mL) was stirred at 50 °C for 24 h under a nitrogen atmosphere. After magnetic stirring for 24 h, mPEG₅₀₀₀-OH (1.0 g, 0.2 mmol) was added to the reaction mixture. After magnetic stirring for another 24 h, the mixture was added to 15 mL of deionized water under sonication, followed by dialysis in a dialysis bag (MWCO: 8000-14000 Da). After 72 h, the solution was freeze-dried under reduced pressure to obtain P^Br (992 mg) as a white powder.

## Synthesis of P^mt

A mixture of P^Br (500 mg) and N,N-dimethyl-chol (100 mg) in acetonitrile (10 mL) was stirred at 80 °C for 48 h under a nitrogen atmosphere. Then, the mixture was added to 15 mL of deionized water under sonication, followed by dialysis in a dialysis bag (MWCO: 8000-14000 Da). After 72 h, the solution was freeze-dried under reduced pressure to obtain P^mt (469 mg) as a white powder.

## Computational details

The DFT calculations were performed using a DMOL3 module of Material Studio 2016 (Tables 1–3)[47]. The generalized gradient approximation (GGA) method with Perdew-Burke-Ernzerhof (PBE) function was employed to optimize the structure and calculate the properties as orbitals and optical. The force and energy convergence criterion were set to 0.02 eV Å⁻¹ and 10⁻⁵ eV, respectively. Energy levels of S₁-Sₙ and T₁-Tₙ are calculated by the vertical excitation of the above optimized structures, with the same method of B3LYP/6-31 G(d).

## Fluorescence quantum yield detection

The fluorescence quantum yield of Aza-TPA-Bodipy was determined against the reference fluorophore IR dye 26 with a known quantum yield of 0.5%[48]. The optical absorbance was measured for both an Aza-TPA-Bodipy and an IR dye 26 solution at 808 nm. Then their NIR-II fluorescence emission intensities were measured under the same 808 nm excitation. Using the measured optical density (OD) and spectrally integrated fluorescence intensity (F), one can calculate the quantum yield of Aza-TPA-Bodipy according to the following formula (1):

$$\Phi x(\lambda) = \Phi_{st}(\lambda) \cdot \frac{F_x}{F_{st}} \cdot \frac{A_{st}(\lambda)}{A_x(\lambda)} = \Phi_{st}(\lambda) \cdot \frac{F_x}{F_{st}} \cdot \frac{1-10^{-OD_{st}(\lambda)}}{1-10^{-OD_x(\lambda)}}$$

$$= 0.5\% \cdot \frac{428859208cps}{81393600cps} \cdot \frac{1-10^{-0.2}}{1-10^{-1.07}} = 1.1\%$$

(1)

Therefore, the fluorescence quantum yield for Aza-TPA-Bodipy is 1.1% at 808 nm excitation.

## Singlet oxygen detection

The singlet oxygen generated by Aza-TPA-Bodipy was measured using DPBF. Firstly, the absorbance of DPBF at 415 nm was adjusted to about 1.0 in DMSO. Secondly, the absorbance of Aza-TPA-Bodipy was adjusted to about 0.2. Thirdly, the cuvette was irradiated with 808 nm monochromatic light at various times, and absorption spectra were measured immediately. The quantum yields for ¹O₂ production (Φ_Δ) of Aza-TPA-Bodipy under irradiation in DMSO solution were calculated using ICG as the standard (Φ_Δ = 7.7 in DMSO)[27]. Φ_Δ was calculated by the following Eq. (2):

$$\Phi_{\Delta sam} = \Phi_{\Delta sam} \left( \frac{m_{sam}}{m_{std}} \right) \left( \frac{F_{std}}{F_{sam}} \right)$$

(2)

Where "*sam*" and "*std*" represent the "Aza-TPA-Bodipy" and "ICG", respectively. "*m*" is the slope of absorbance change curve of DPBF at 415 nm, $F = 1-10^{-O.D.}$ (O.D. is the absorbance of the solution at 808 nm).

## Preparation of NP^Bodipy

P^Bodipy (10 mg) was dissolved in 1 mL of DMSO, which was then dropped into 9 mL of water during the period of magnetic stirring. Then, the mixture was dialyzed in a dialysis bag (MWCO: 8000–14000 Da). After 72 h, the mixture was concentrated through ultrafiltration (5000 g), followed by filtration using a 0.22 μm syringe-driven filter to give NP^Bodipy.

## Preparation of mt-NP^Bodipy

P^mt (10 mg) was first dissolved in 1 mL of DMSO, which was then dropped into 9 mL of water during the period of magnetic stirring. Then, the mixture was dialyzed in a dialysis bag (MWCO: 8000–14000 Da) to obtain NP^mt. After 72 h, the mixture was concentrated to 5 mL through ultrafiltration. Next, P^Bodipy (10 mg) dissolved in ethanol was evaporated under a rotatory evaporator to generate a thin film, which was further dried under an ultra-high vacuum for 0.5 h. The film was hydrated with NP^mt for 30 min and then sonicated for 12 min by using a pulse 3/2 s on/off at a power output of 60 W, followed by filtration using a 0.22 μm syringe-driven filter to give mt-NP^Bodipy.

## Preparation of mt-NP^Bodipy@Cy5.5

Cy5.5 NHS (1 mg) was dissolved in 1 mL of DMSO, N,N-dimethylethylenediamine (0.2 mg) was added, and the solution was stirred for 2 h at room temperature. Next, the mixture and P^mt (20 mg) in dry DMF (3 mL) were stirred at 50 °C for 24 h under a nitrogen atmosphere. The mixture was added to 5 mL of deionized water under sonication, followed by dialysis in a dialysis bag (MWCO: 8000-14000 Da). After 72 h, the solution was freeze-dried under reduced pressure to obtain P^mt@Cy5.5. P^mt@Cy5.5 (10 mg) was first dissolved in 1 mL of DMSO, which was then dropped into 9 mL of water during the period of magnetic stirring. Then, the mixture was dialyzed in a dialysis bag (MWCO: 8000-14000 Da) to obtain NP^mt@Cy5.5. After 72 h, the mixture was concentrated to 5 mL through ultrafiltration. Next, P^Bodipy (10 mg) dissolved in ethanol was evaporated under a rotatory evaporator to generate a thin film, which was further dried under an ultra-high vacuum for 0.5 h. The film was hydrated with NP^mt@Cy5.5 for 30 min and then sonicated for 12 min by using a pulse 3/2 s on/off at a power output of 60 W, followed by filtration using a 0.22 μm syringe-driven filter to give mt-NP^Bodipy@Cy5.5.

## Cytotoxicity study

The cytotoxicity of mt-NP^Bodipy + L was evaluated on CT26 cells by an MTT assay. The CT26 cells ($5 \times 10^3$ per well) were seeded into 96-well plates and grew for 24 h to reach the confluence of 80%. CT26 cells were incubated with various concentrations of Aza-TPA-Bodipy (0.025, 0.25, 2.50, 5.00, and 10.00 μg mL⁻¹). Subsequently, the cells were

exposed to 1064 nm laser irradiation (1.0 W cm$^{-2}$) for 5 min. The cell viability was measured by an MTT assay.

## Cell membrane imaging

The CT26 cells were seeded into confocal dishes at a density of 3×10$^5$ cells/dish. mt-NP$^{Bodipy}$@Cy5.5 was firstly irradiated by an 808 nm laser at different time points from 0-10 min, resulting in partially dissociated nanoparticles (mt-NP$^{Bodipy}$@Cy5.5/Dis). mt-NP$^{Bodipy}$@Cy5.5/Dis were then incubation with CT26 cells for other 1 h. After 1 h of incubation, the cell culture medium was aspirated and the cells were washed 2 times with PBS. Add the appropriate volume of cell membrane staining working solution (DiO), gently shake to make the dye cover all the cells evenly, and incubate the cells for 10 min at 37 °C protected from light. Aspirate the cell membrane staining working solution, wash the cells with PBS 3 times, and then add the pre-warmed cell culture solution at 37 °C and then observe by CLSM.

## Live/dead cell staining of 3D tumor spheroids

For live/dead cell staining of 3D tumor spheroids, 1% agarose gel solution (50 μL) was added to each 96-well plate. Afterward, 1600 cells (200 μL complete medium) were added to each well. On the 7th day, the cell spheres were formed. The fresh complete medium containing different formulations (PBS, NP$^{Bodipy}$, mt-NP$^{Bodipy}$, NP$^{Bodipy}$ + L, and mt-NP$^{Bodipy}$ + L) was added after removing the culture medium. After 12 h incubation, the cells of the laser groups were irradiated with an NIR light of 808 nm at the intensity of 1.0 W cm$^{-2}$ for 5 min. After further incubation for another 12 h at 37 °C, the 3D tumor spheroids were washed with PBS three times and then were successively stained with PI based on the manufacturer's instruction of the live/dead cell staining kit, respectively. Last, the cells were imaged by CLSM with Z-stack scanning.

## Morphology changes of cancer cells imaged by SEM

Firstly, 1.2 × 10$^5$/well CT26 cells were seeded on sterilized coverslips in 6-well plates and incubated in DMEM for 24 h. Then the medium was replaced by fresh DMEM containing different formulations (PBS, NP$^{Bodipy}$, mt-NP$^{Bodipy}$, NP$^{Bodipy}$ + L, and mt-NP$^{Bodipy}$ + L). After 12 h incubation, the cells of the laser groups were irradiated with a NIR light of 808 nm at the intensity of 1.0 W cm$^{-2}$ for 5 min, and incubated for 30 min at 37 °C. After the medium was removed, cells were washed with PBS and fixed with 2.5% glutaraldehyde solution overnight. Further, 4% paraformaldehyde was added to fix the CT26 cells for another 20 min. After the sample was dehydrated by 30%, 50%, 70%, 90%, and 100% of ethanol solutions, the cells on coverslips were analyzed by SEM.

## RNA-seq analysis

CT26 cells were treated with PBS, mt-NP$^{Bodipy}$, and mt-NP$^{Bodipy}$ + L for 12 h, the cells of the laser groups were irradiated with a NIR light (808 nm, 1.0 W cm$^{-2}$, 5 min), and incubated for other 12 h at 37 °C. To purify RNA, cells (1 million) were collected from every treatment group (n = 3 experimental replicates). The RNA quality was verified with a NanoDrop 2000/c Spectrophotometer. The sequencing data has been submitted to the National Center for Biotechnology Information (NCBI) Sequence Read Archive (SRA) database under Bioproject ID PRJNA976350 and will be made public upon publication. Sequencing was performed using the BGISEQ-500 platform. Gene transcription levels were quantified using RSEM.

## In vivo biodistribution

CT26 cells were subcutaneously injected into the right buttocks of BALB/c mice (1,/ × 10$^6$ cells per mouse). Six days later, tumor-bearing mice were intravenously injected with 200 μL of mt-NP$^{Bodipy}$ (500 μg mL$^{-1}$). Then mice were imaged by a NIR-II animal fluorescence imaging instrument at 0.5 h, 4 h, 7 h, 12 h, and 24 h. After 24 h, the mice

were euthanized and the tumor and various organs (heart, liver, spleen, lung, and kidney) were excised for fluorescence imaging.

## Optical system for NIR-II fluorescence

An 808 nm diode laser (Artemis, China) was used as the excitation light. A two-dimensional InGaAs camera (Prineton Instruments, U.S.) with 640 pixels × 512 pixels was used for capturing all NIR-II images. A NIR lens (Artemis, China) was used to focus the image onto the photodetector. The emission filter of all NIR-II images was 1000 nm. The exposure time of all NIR-II images was 30 ms. All NIR-II fluorescence images were analyzed by the software (ImageJ).

## In vivo antitumor efficacy

CT26 cells were subcutaneously injected into the right buttocks of BALB/c mice (1 × 10$^6$ cells per mouse). Six days later, tumor-bearing BALB/c mice were randomly divided into five groups (n = 5): (1) PBS alone, (2) NP$^{Bodipy}$, (3) mt-NP$^{Bodipy}$, (4) NP$^{Bodipy}$ + L, (5) mt-NP$^{Bodipy}$ + L. Different reagents were intravenously injected into mice at Aza-TPA-Bodipy equivalent dose of (200 μL, 500 μg mL$^{-1}$) on day 6. Then at 7 h after injection, mice in group NP$^{Bodipy}$ + L and mt-NP$^{Bodipy}$ + L were anesthetized and irradiated with light (5 min) at the tumor site. At the designed time (6, 8, 10, 12, and 14 days), mice were weighed and tumour size measured. Fourteen days later, all mice were euthanized and the tumors were collected for H&E and Ki67 stain.

For CT26 bilateral tumor-bearing BALB/c mice, CT26 cells were subcutaneously injected into the right buttocks (primary tumors) of each mouse (1 × 10$^6$ cells per mouse). Four days later, an equal number of cells were subcutaneously injected into the left buttocks (distant tumors) of the same mouse. Then, the bilateral tumor-bearing mice were randomly divided into three groups (PBS, mt-NP$^{Bodipy}$, and mt-NP$^{Bodipy}$ + L n = 7) after the primary tumors had grown for 8 days. mt-NP$^{Bodipy}$ (200 μL, 500 μg mL$^{-1}$) were intravenously injected on 8, 10, and 12 days post-operation. Then at 7 h after injection, mice in group mt-NP$^{Bodipy}$ + L were anesthetized and irradiated with light (5 min) at the tumor site. At the designed time (8, 10, 12, 14, 16, and 18 days), mice were weighed and tumour size measured. Eighteen days later, the tumors and LNDs were collected for further analysis.

To establish a rechallenge model, CT26 cells were subcutaneously injected into the right buttock of each mouse (1 × 10$^6$ cells per mouse). mt-NP$^{Bodipy}$ (200 μL, 500 μg mL$^{-1}$) were intravenously injected on 8, 10, and 12 days post-operation. Then at 7 h after injection, mice were anesthetized and irradiated with light (5 min) at the tumor site. After 60 days, CT26 cells (1 × 10$^6$ cells) were subcutaneously inoculated into the left buttocks of each mouse. As a control experiment, CT26 cells (1 × 106 cells) were subcutaneously inoculated into the left buttocks of an age-matched naïve mice. At the designed time (68, 70, 72, 74, 76, and 78 days), mice were weighed and tumour size measured. Seventy-eight days later, the LNDs and serum were collected for FCM and Elase analysis, respectively. For survival analysis, mice were continuously monitored for 100 days.

## Flow cytometry analysis of the animal tissue

The obtained tumor tissues, or draining LNDs were used to prepare single-cell suspensions. The single-cell suspensions were further incubated with various antibodies against the immune cells. For the analysis of T cells in the tumor, cells were stained by anti-CD3-PE, anti-CD4-APC, and anti-CD8-FITC. For analyzing DCs in tumors and lymph nodes, cells were stained by anti-CD11c-PE, anti-CD80-FITC, and anti-CD86-APC. For characterizing TAM in tumors, cells were stained by anti-F4/80-PE and anti-CD206-FITC. For analyzing memory T cells in LNDs, cells were stained by anti-CD3-PE, anti-CD8-FITC, anti-CD62L-APC, and anti-CD44-PC5.5. Flow cytometric data acquisition was performed with CytExpert software, and the data were processed using FlowJo software.

## Statistical analysis

All data presented are reported as mean ± SD unless otherwise noted. All the experiments were repeated at least 3 times. Statistical comparisons were made by one-way or two-way ANOVA with Bonferroni multiple comparisons post-test. A log-rank (Mantel-Cox test) was conducted to compare the survival curves. For all graphs, $*p < 0.05$; $**p < 0.01$; $***p < 0.001$; and $****p < 0.0001$. All statistical calculations were carried out with Prism 8 (GraphPad Software).

## Reporting summary

Further information on research design is available in the Nature Portfolio Reporting Summary linked to this article.

## Data availability

The raw sequence data generated in this study have been deposited in the National Center for Biotechnology Information (NCBI) Sequence Read Archive (SRA) database (Bioproject ID PRJNA976350) under accession code https://www.ncbi.nlm.nih.gov/sra/PRJNA976350. The remaining data are available within the Article, Supplementary Information, or Source Data file. Source data are provided with this paper.

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

## Acknowledgements

This work was supported by the Natural Science Foundation of Beijing of China (2202071) and the National Natural Science Foundation of China (52073015). Figures 6a, 7a, h were created with BioRender.com.

## Author contributions

D.T. and H.X. conceived the study. D.T. designed the experiments. D.T. performed the synthesis and quantification. D.T., M.C., and B.W. performed the **in vitro** experiments. D.T., M.C. and B.W. performed the **in vivo** experiments. D.T., G.L., H.Z. and H.X. drafted and finalized the manuscript with input from all other authors.

## Competing interests

The authors declare no competing interests.
