## [Peer Review File · Nature Communications]

REVIEWER COMMENTS

Reviewer #1 (Remarks to the Author):

Review for NCOMMS-23-18572

Nanoparticles Destabilizing the Cell Membranes Triggered by NIR Light for NIR-II Fluorescence Bio-Imaging, Photodynamic Therapy, and Photo-Immunotherapy

The manuscript titled 'Nanoparticles Destabilizing Cell Membranes Triggered by NIR Light for NIR-II Fluorescence Bio-Imaging, Photodynamic Therapy, and Photo-Immunotherapy,' authored by Dongsheng Tang et al., introduces a polymer-based nanoparticle for cancer photodynamic and photoimmunotherapy. While the authors have successfully developed a controllable system using PBodipy with a ROS-cleavable linker to regulate Pmt release, the manuscript lacks organization and the authors' claims lack persuasiveness. Additionally, there are several comments that need to be addressed to improve the clarity of the claims and data. Based on these factors, Unfortunately, this reviewer recommends for author to submit this article in another journal.

Please check the following comments;

[1] According the introduction section, the authors claimed that they developed a membrane-destabilizing nanoparticle shielded by PBodipy in this study. However, in the following results section, the sufficient results regarding on how the shielding with PBodipy effects on the activity or toxicity of NPmt are not provided. Moreover, the toxicity from mt-NPBodipy-induced membrane destabilization was not sufficiently proved.

[2] How can PBodipy and Pmt self-assemble into nanoparticles? How much of each component of Pmt and PBodipy was contained? The results on the characterization of Pmt without shielding are missing.

[3] In the TEM image in Fig. 3j and the DLS result in 3i, it seems that small and compact nanoparticles are fabricated clearly after light irradiation rather than enlargement and dissociation. The authors should give more explanation on this result. How about the number of particles? Does the number of particles reduce? Release of Pmt should be further proved by supportive results. (ex. detection of supernatant)

[4] Cell viability result in Fig. 4e seems that NPBodypy + L and mt-NPBodypy + L showed similar cytotoxicity. The quantitative and clear results to support the membrane destabilization activity of mt-NPBodypy over NPBodypy should be added.

[5] In Fig. 5, the authors need to address more interpretation and discussion on the RNA-Seq sections to help the readers better understand the mechanism of the particles.

[6] In Fig 6b, the mt-NPBodipy can be delivered to the other tissues, such as liver. It seems that V group in Fig. 6k induced the weight loss. Is there any toxicity on the tissues other than tumor? Moreover, the site of tumor and the intensity scale should be presented in Fig. 6b.

[7] In immunological assays, it seems that mt-NPBodipy can induce strong immune response. What is the main component or reason for induction of immune response?

[8] The scheme in Fig. 1 is hard to understand the system. At least, the color of polymer scheme should match with the chemical structure. The names of NPs are confusing. Also, the authors should revise the whole figure number in the manuscript and legend. Ex) the explanation on the figure is not matched with the Figure:

- Fig. 3e, 3f, 3i-k, Fig. 3g in Page 12,

- There is no explanation on Fig. 4c, 4d.

[9] The physical characteristics of NPBodipy and mt-NPBodipy are similar except the zeta potential which is mentioned incorrect for NPBodipy in manuscript . Only Changes in hydrodynamic diameters of mt-NPBodipy after NIR light irradiation (figure 3.i) is shown as advantage over the NPBodipy , but same data has been not produced for it.

[10] Similarly the hydrodynamic radius and PDI of both the nanoparticles are also close but DLS spectrum of NPBodipy does not show the same distribution.

[11] There is no figure 3k in manuscript but mentioned in text.

[12] No comparative studies has been done for vitro evaluation of NPBodipy and mt-NPBodipy . It is not clear at what basis mt-NPBodipy has been chosen over NPBodipy for biomedical application.

Figure 4(c-g) mentioned as Figure 3 (c-g).

[13] In figure 6, the notions are not correct. Figure number 6d has been repeated and scale of x-axis is also wrong.

[14] There are some biological studies in which they have include NPBodipy datas and in some cases they have excluded them without giving any valid reason.

[15] With some presentation errors, this manuscript does not able to justify the reason to synthesize complex organic framework with enhanced properties, instead they are getting similar result with the precursor nanoparticles itself.

[16] In Figure 3j, considering the scale difference from 3c, the NPs seem to aggregate rather than dissociate after NIR irradiation. It is better to image the different areas to investigate the disintegration of NPs. And also, it is better to supplement the explanation of the phenomenon of the small bold spots that came up after the irradiation.

[17] In the investigation of CT26 (Figure 4), the authors insist that ROS generation triggered the dissociation of NPBodypy. However, the ROS generation is not confirmed at the cellular level. It is better to supplement the ROS generation assays for clearance. Moreover, the NIR effect should be considered with the control experimental group of non-dissociative NPs.

[18] It should be supplemented with an explanation of the immunotherapeutic mechanism of the mt-NPBodypy.

Reviewer #2 (Remarks to the Author):

The manuscript by Tang et al. reported the development of a nanoparticle based on a cell-membrane-targeting polymer and a near-infrared-II and ROS responsive polymer for cancer treatment in a spatial-temporal manner upon light-irradiation. Such particle could also enable NIR-II bio-imaging to track the particle biodistribution.

One major problem of the manuscript is the lack of convincing preclinical benefits for such new formulations. The authors should use other immune-cold tumor models (4T1 or B16F10) to demonstrate the advantage of their new particle formulation. One advantage that can be expected is that the photo therapy could enhance the tumor immunogenicity and recruit more immune cells to these immune-cold tumors. CT26 overall is pretty simple to treat using just checkpoint blockade antibodies (Translational Oncology, Volume 20, 2022, 101405). The benefits of such new formulation need to be further tested in more tumor models.

Another problem is that using such new formulation for imaging studies has not been thoroughly studied. Figure 6b-c showed that the particle could mainly accumulate in both liver and tumor. It is understandable that particle formulations often accumulate in the liver. Nevertheless, the authors need to show how long the particle can stay in the tumor tissues. In addition, the tumor penetration profile of the particle should also be revealed. One concern for cationic particles is that they may stay in the tumor

periphery or just near tumor vasculature and cannot get deep into the tumor, presumably because the ionic interaction between cationic particle and negatively charged extracellular matrix and cells close to the vasculature. The authors should consider perform imaging studies to investigate that; especially upon light irradiation, the disassembled polymer should get deeper into tumor tissues to improve efficacy.

In addition, the authors should consider moving some figures to supplementary information as there are 9 figures in the manuscript. Some figures (e.g., representative tissue histology, flow cytometry gating) can be in the supporting figures.

Other problems:

1. Figure 6b-c, need fluorescence intensity scale bar. Also please monitor tumor / liver fluorescence change over long time.
2. Figure 6d. The x axis should be time (hours)? Instead of wavelength?
3. Figure 3, whether ROS could affect the fluorescence light emission of the dye TPA-BODIPY? The ROS is generated upon light irradiation on the dye, and the dye also emits fluorescence. How does the energy transfer occur in such process (Figure 2d only showing the energy from S1 to T1 for photodynamic therapy).
4. Figure 7. The percentage of immune cells in what cell types are not clear. For example, (b-e) the percentage of CD80+ CD86+ is on CD11c+ DCs or CD11c+MHC-II+CD45+ DCs in tumors / lymph nodes? (f-g) is percentage based on CD3+ lymphocytes or CD45+ total tumor infiltration lymphocytes? (h-i) is based on CD45+ total tumor infiltration lymphocytes? The authors also need to provide the total cell population numbers for each treatment, because the cell numbers are also important to tell the outcome of immunotherapy (e.g., whether the total number of DCs and CD8+CD3+ T cells increased in tumor infiltration lymphocytes). Same problems also occur in Figure 8c-f and Figure 9f-h.
5. The n numbers are too low for flow cytometry studies in Figures 7-9, please consider increasing n numbers to 5 for comparison.
6. Figure 9, for memory T cell analysis, the spleen results often could not reflect the memory T cells population changes. Please take on treated tumor to see (i) whether effector memory CD44CD8 T cells increases (Nature volume 520, pages373–377 (2015)). (ii) whether effector and central memory CD8 T cells increase in lymph nodes.
7. The authors did not study other immunosuppressive cells change over the course of treatment. For example, the Treg cells and MDSC cells are not examined. In addition, for CD8CD3 T cells, whether these cells have viability (ki67+) and functionality (IFN-gamma+ or TNF-alpha+) are not studied. The problem that T cells become exhaustion upon treatment is a serious problem for cancer immunotherapy.
8. All NMR spectra in Supplementary figures did not have integration and peak assignments. Please assign all peaks for all NMR spectra.

9. The size exclusion chromatography for all polymers are missing. No molecular weights and molecular weight distribution information for all polymers.

10. For DFT computation, please give the computation methods information (functionals, basis sets, etc.). Such information is critical especially for the excited state calculation. Molecule coordinates are also missing in supplementary information.

Some typos (please proofread the manuscript):

11. In abstract: "In this study, Herein, we report a cell membrane-targeting..." please remove either "in this study" or "herein"

12. Figure 9 legend "Data are represented as mean \pm SD in g, h, l, and j..." should be "Data are represented as mean \pm SD in g, h, i, and j..."

Reviewer #3 (Remarks to the Author):

In this MS, Tang et al. report the cationic polymer nanoparticles that enable photodynamic therapy and immunotherapy of cancer as well as NIR-II bioimaging. However, there are numerous studies on cancer treatment using various nanomaterials with multiple functions (such as PTT, PDT, chemotherapy, immunotherapy, etc.), and the therapeutic effects in this manuscript do not seem to differ from these studies. The design of integrating different functions into nanoparticles for cancer treatment is quite common. In addition, the experimental results did not adequately support the material design that the negatively charged PBodipy can shield the quaternary ammonium cation of Pmt and reduce the toxic side effects of cationic polymer in bio-applications. Therefore, this manuscript is not suitable for publication in Nature Communications.

1) Fig.1 was not cited in the main text.

2) The zeta potential of NPmt, and that of mt-NPBodipy after light irradiation should be investigated and included in Fig.3e.

3) Investigate the stability of mt-NPBodipy as the TME (such as pH and ROS...) may dissociate nanoparticles.

4) There is no evidence to support the authors' assumption that the negatively charged PBodipy can shield the quaternary ammonium cation from Pmt and reduce the toxic side effects of cationic polymer in bio-applications. The NPmt group should be included in relevant experiments both in vitro and in vivo.

- 5) After 808-nm laser induced dissociation, does the released cationic Pmt circulate to other tissues and cause toxic side effects?
- 6) The optical properties of mt-NPBodipy are notably different from those of Aza-TPA-Bodipy. During in vivo imaging, 808-nm laser irradiation can cause mt-NPBodipy dissociation, affecting tumor imaging analysis (Fig.6b-d).
- 7) Has the thermal effect caused by 808-nm laser affected the photodynamic immunotherapy? For example, Nat. Commun. 2021,12:742 and J. Am. Chem. Soc. 2021, 143, 8116–8128.
- 8) The authors need to specify other parameters for bioimaging application, such as power density, filter and exposure time, etc.
- 9) The protocol of preparing mt-NPBodipy@cy5.5 should be provided. After laser-induced dissociation of mt-NPBodipy@cy5.5, is there any free cy5.5 that interferes with cell labeling?
- 10) There are two d in Fig.6 and the abscissa “Wavelength (nm)” is incorrect. Also, Figs.6f-j show the same information as Fig.6e and should be part of the supporting data. Please check all figures to make the manuscript clearer and easier to read.

REVIEWER COMMENTS

Reviewer #1 (Remarks to the Author):

Review for NCOMMS-23-18572

Nanoparticles Destabilizing the Cell Membranes Triggered by NIR Light for NIR-II Fluorescence Bio-Imaging, Photodynamic Therapy, and Photo-Immunotherapy

The manuscript titled 'Nanoparticles Destabilizing Cell Membranes Triggered by NIR Light for NIR-II Fluorescence Bio-Imaging, Photodynamic Therapy, and Photo-Immunotherapy,' authored by Dongsheng Tang et al., introduces a polymer-based nanoparticle for cancer photodynamic and photoimmunotherapy. While the authors have successfully developed a controllable system using PBodipy with a ROS-cleavable linker to regulate Pmt release, the manuscript lacks organization and the authors' claims lack persuasiveness. Additionally, there are several comments that need to be addressed to improve the clarity of the claims and data. Based on these factors, Unfortunately, this reviewer recommends for author to submit this article in another journal.

Please check the following comments;

[1] According the introduction section, the authors claimed that they developed a membrane-destabilizing nanoparticle shielded by PBodipy in this study. However, in the following results section, the sufficient results regarding on how the shielding with PBodipy effects on the activity or toxicity of NPmt are not provided. Moreover, the toxicity from mt-NPBodipy-induced membrane destabilization was not sufficiently proved.

Response: Many thanks to the reviewers for their constructive comments. As shown in Fig. R1, the hemolysis rate caused by mt-NP^{Bodipy} at 50 mg mL⁻¹ is less than 5%. However, after 808 nm light irradiation 5 min, a significant increase in hemolysis rate was observed. Meanwhile, more than 5% hemolysis rate observed in NP^{mt} at 50

mg mL⁻¹. These results indicated that mt-NP^{Bodipy} has good biocompatibility, but after exposure to light, mt-NP^{Bodipy} releases NP^{mt} leading to cell death.

Fig. R1. a Photographs of erythrocytes after incubation with 1× PBS buffer (negative control), 1% Triton X-100 (positive control), and mt-NP^{Bodipy}, mt-NP^{Bodipy} + L 5 min, and NP^{mt} at the different concentration for 2 h, followed by centrifugation. **b** Hemolysis percentages of erythrocytes after incubation with mt-NP^{Bodipy}, mt-NP^{Bodipy} + L 5 min, and NP^{mt} at different concentrations for 2 h (n = 3).

[2] How can PBodipy and Pmt self-assemble into nanoparticles? How much of each component of Pmt and PBodipy was contained? The results on the characterization of Pmt without shielding are missing.

Response: Many thanks to the reviewers for their constructive comments.

(a) How can PBodipy and Pmt self-assemble into nanoparticles?

Response: Based on our previous research (Fig. R2a, *Angew. Chem. Int. Ed.* **2022**, 61, e202201486), P^{Bodipy} and P^{mt} possess both hydrophilic and hydrophobic components, facilitating their self-assembly into nanoparticles *via* interactions between the hydrophilic and hydrophobic units.

(b) How much of each component of Pmt and PBodypy was contained?

Response: By calculation, 340 μg P^{Bodypy} and 220 μg P^{mt} were included in 1 mL mt-NP^{Bodypy}.

(c) The results on the characterization of Pmt without shielding are missing

Response: P^{mt} was self-assembled to NP^{mt}, which was subsequently shielded by P^{Bodypy} to get mt-NP^{Bodypy}. The average particle sizes of NP^{mt} measured by dynamic light scattering (DLS) were around 3.7 nm (Fig. 2b) and Zeta potentials (mV) at 35 mV.

Fig. R2. a Early research on polymer assembly. b DLS profiles of NP^{mt}. c Hydrodynamic diameters and Zeta potentials of NP^{mt}.

[3] In the TEM image in Fig. 3j and the DLS result in 3i, it seems that small and compact nanoparticles are fabricated clearly after light irradiation rather than enlargement and dissociation. The authors should give more explanation on this result. How about the number of particles? Does the number of particles reduce? Release of Pmt should be further proved by supportive results. (ex. detection of supernatant)

Response: Many thanks to the reviewers for their constructive comments. As shown in Fig. 3i, after 808 nm light irradiation for 10 min, the particle sizes were enlarged and a small portion of mt-NP^{Bodipy} was disassociated possibly owing to the breakdown of thioketal bonds. The disturbance of the regular nanostructure may account for this phenomenon, resulting in the simultaneous occurrence of small particle dispersion and significant aggregation. This phenomenon has been observed in other studies (Fig. R3).

Fig. R3. Disassociated nanoparticles in another research.

[4] Cell viability result in Fig. 4e seems that NP^{Bodipy} + L and mt-NP^{Bodipy} + L showed similar cytotoxicity. The quantitative and clear results to support the membrane destabilization activity of mt-NP^{Bodipy} over NP^{Bodipy} should be added.

Response: Many thanks to the reviewers for their constructive comments. We observed that the ROS generation of Aza-TPA-Bodipy was significantly more efficient than that of ICG, with a quantum yield of 16.4% compared to 7.7% for ICG. Therefore, a substantial amount of ROS could be generated by both NP^{Bodipy} + L and mt-NP^{Bodipy} + L to induce cancer cell death. However, a higher level of attachment to the cell membranes through electrostatic interaction was exhibited by mt-NP^{Bodipy}, leading to more severe damage to the membranes and an increase in cell necrosis. Therefore, mt-NP^{Bodipy} + L demonstrated strong anticancer activity with lower IC₅₀ values than mt-NP^{Bodipy} + L (NP^{Bodipy} + L, IC₅₀ = 0.62 ± 0.30 µg mL⁻¹; mt-NP^{Bodipy} + L, IC₅₀ = 0.50 ± 0.20 µg mL⁻¹).

Moreover, in order to study the effect of mt-NP^{Bodipy} + L on cell membranes, morphological changes of CT26 cells were observed by scanning electron microscopy (SEM) (*Adv. Mater.* **2020**, 32, 2001108). Upon treatment of the cells with mt-NP^{Bodipy} + L, a large number of necrotic cells with completely disintegrated cell membranes were observed. This phenomenon indicated that the mt-NP^{Bodipy} + L could successfully release cationic polymer P^{mt} to destroy the cell membranes.

[5] In Fig. 5, the authors need to address more interpretation and discussion on the RNA-Seq sections to help the readers better understand the mechanism of the particles.

Response: Many thanks to the reviewers for their constructive comments. we have added some interpretations and discussions in the revised manuscript, such as “Gene Ontology (GO) enrichment analysis showed that the CT26 cells treated with mt-NP^{Bodipy} + L mainly displayed influenced expressions of genes in the nucleus, cytoplasm, cytosol, nucleoplasm, and cytoskeleton in Cellular Component (Fig. 5d), and in the major pathways of cellular responses to DNA damage stimulus, DNA repair, DNA replication, membrane fusion, and double-strand break repair in

Biological Process (Fig. 5e), protein binding, DNA binding, single-stranded DNA binding, double-stranded DNA binding, and damaged DNA binding in Molecular Function (Fig. 5f). It is well known that ROS can induce base oxidation and single strand breaks (SSBs) in DNA, which can be repaired through base excision/single strand breaks (BER/SSBR). If left unrepaired, damage caused by ROS can impede DNA replication and transcription, ultimately resulting in genomic instability, mutations, and the promotion of tumor development. Therefore, this is consistent with the experimental results that mt-NP^{Bodipy} + L exhibited predominant localization within the cell membranes and yielded a substantial quantity of ROS, capable of inducing DNA damage while also perturbing cell membrane function. What's more, the Kyoto Encyclopedia of Genes and Genomes (KEGG) and Gene set enrichment analysis (GSEA) showed that mt-NP^{Bodipy} + L substantially affected the cell cycle, DNA replication, homologous recombination, as well as a series of immune and inflammatory-related signaling pathways, including mitogen-activated protein kinase (MAPK) signaling pathway, forkhead box O (FoxO) signaling pathway, NF- κ B, Toll-like receptor, chemical carcinogenesis-DNA adducts, cytokine-cytokine receptor interaction, IL-17 and antigen processing and presentation signaling pathways (Fig. 5g, h, and Supplementary Figs. 19)".

[6] In Fig 6b, the mt-NPBodipy can be delivered to the other tissues, such as liver. It seems that V group in Fig. 6k induced the weight loss. Is there any toxicity on the tissues other than tumor? Moreover, the site of tumor and the intensity scale should be presented in Fig. 6b.

Response: Many thanks to the reviewers for their constructive comments.

(a) In Fig 6b, the mt-NPBodipy can be delivered to the other tissues, such as liver. It seems that V group in Fig. 6k induced the weight loss. Is there any toxicity on the tissues other than tumor?

Response: As shown in Figure R4, no noticeable histopathological damage or abnormal blood biochemical indexes were observed in the different treatment groups, indicating minimal side effects of NP^{Bodipy} and mt-NP^{Bodipy}.

Fig. R4. Biosafety evaluation of NP^{Bodipy} and mt-NP^{Bodipy} *in vivo*. **a** H&E staining of major organs (heart, liver, spleen, lung, and kidney) after different treatment. **b** Biochemical analysis of serum: alanine aminotransferase (ALT), aspartate aminotransferase (AST), alkaline phosphatase (ALP), creatinine (CRE), and blood urea nitrogen (BUN). $n = 3$. Data are presented as mean \pm SD.

(b) Moreover, the site of tumor and the intensity scale should be presented in Fig. 6b.

Response: As shown in Figure R5, we have revised the intensity scale the site of tumor and organ in the revised manuscript.

Fig. R5. NIR-II fluorescence bioimaging of mt-NP^{Bodipy} *in vivo*.

[7] In immunological assays, it seems that mt-NP^{Bodipy} can induce strong immune response. What is the main component or reason for induction of immune response?

Response: Many thanks to the reviewers for their constructive comments. Firstly, ROS is produced by mt-NP^{Bodipy} under 808 nm light, which induces ICD in cells and subsequently activates the anti-tumor immune response. Secondly, P^{mt} cationic polymers are released by mt-NP^{Bodipy} under light irradiation, which can induce endosome membrane destruction, cathepsin B release, and K⁺ efflux, playing a key role in the inflammation activation of the host (*ACS Nano* **2019**, 13, 3083.).

[8] The scheme in Fig. 1 is hard to understand the system. At least, the color of polymer scheme should match with the chemical structure. The names of NPs are confusing. Also, the authors should revise the whole figure number in the manuscript and legend. Ex) the explanation on the figure is not matched with the Figure:

- Fig. 3e, 3f, 3i-k, Fig. 3g in Page 12,
- There is no explanation on Fig. 4c, 4d.

Response: Many thanks to the reviewers for their constructive comments.

First, we have revised the color of polymer in the revised manuscript (Fig. R6). P^{Bodipy} represent the amphiphilic polymer with Aza-TPA-Bodipy units and ROS-responsive thioketal bonds (2,2'-(propane-2,2-diy)bis(sulfanediyl))bis(ethan-1-ol), defined as PSDE) on its main chains. P^{Bodipy} was co-assembled into the nanoparticles,

termed as NP^{Bodipy}. P^{mt} represent the cell membrane-targeting cationic polymer with antineoplastic activity. P^{mt} was self-assembled to NP^{mt}, which was subsequently shielded by P^{Bodipy} to get mt-NP^{Bodipy}.

Fig. R6 Schematic illustration of the chemical structure of P^{Bodipy} and P^{mt}. P^{Bodipy} with NIR-II fluorescent Bodipy units and ROS-responsive thioketal linkages can be excited by NIR light to induce ROS, resulting in subsequent breakage of the thioketal bonds and the degradation of the polymer. P^{mt} contains numerous cholesterol molecules for cell membrane targeting and quaternary ammonium salts for cell membrane destabilization.

Second, we have revised the figure number and legend in the revised manuscript, such as “The results indicated that NP^{Bodipy} and mt-NP^{Bodipy} had spherical morphology and a uniform size distribution (Fig. 3c and Supplementary Fig. 17a). The average particle sizes of NP^{Bodipy} and mt-NP^{Bodipy} measured by dynamic light scattering (DLS) were around 84 nm and 92 nm (Fig. 3d, e and Supplementary Fig. 17b-d) with polydispersity indexes (PDI) at 0.20 and 0.24, and Zeta potentials (mV) at -20 mV and -11 mV, respectively.”

“To further study the optical properties of NP^{Bodipy} and mt-NP^{Bodipy}, the absorption spectra of NP^{Bodipy} and mt-NP^{Bodipy} were measured to show that they had similar UV-Vis properties (Fig. 3f and Supplementary Fig. 17e). Notably, compared with Aza-TPA-Bodipy, the maximum absorption peak (λ_{\max}) of NP^{Bodipy} and mt-NP^{Bodipy} decreased. For example, the λ_{\max} of NP^{Bodipy} and mt-NP^{Bodipy} decreased from

833 nm to 789 nm and 792 nm, respectively. Furthermore, the fluorescence (FL) emission spectra of NP^{Bodipy} and mt-NP^{Bodipy} demonstrated that they had maximum emission peaks at 926 and 914 nm, respectively, with a long emission tail extended to 1000-1200 nm (Fig. 3g and Supplementary Fig. 17f).”

“Subsequently, the generation of ¹O₂ of mt-NP^{Bodipy} was detected by electron spin resonance (ESR) using 2,2,6,6-tetramethylpiperide (TEMP) as a trapping agent (Fig. 3h).”

“Further, the morphological and size changes of the mt-NP^{Bodipy} + L were studied. After 808 nm light irradiation for 10 min, the particle sizes were enlarged and a small portion of mt-NP^{Bodipy} was disassociated possibly owing to the breakdown of thioketal bonds (Fig. 3i). Notably, the TEM results displayed that the spheric nanoparticles almost disappeared after light irradiation for 10 min, indicating the complete disassociation of mt-NP^{Bodipy} (Fig. 3j).”

“The cytotoxicity of the mt-NP^{Bodipy} + L was evaluated against mouse colon cancer cell line CT26 by a 3-(4,5-dimethylthiazol-2-yl)-2,5-diphenyltetra-zolium bromide (MTT) assay (Fig. 4e).”

“The anticancer activity of mt-NP^{Bodipy} + L was further visualized by 3D tumor cell spheres by CLSM (Fig. 4f).”

“As shown in Fig. 4g, compared to PBS, NP^{Bodipy}, and mt-NP^{Bodipy} treated cells with intact membranes, the cells shrank and the morphology of the membranes became irregular after treatment with NP^{Bodipy} + L.”

Third, we have added the description about Fig. 4c and 4d in the revised manuscript, such as “The quantification of the fluorescence signal indicated that mt-NP^{Bodipy}@Cy5.5 irradiated for 10 min demonstrated strong red fluorescence (Fig. 4c, d).”

[9] The physical characteristics of NP^{Bodipy} and mt-NP^{Bodipy} are similar except the zeta potential which is mentioned incorrect for NP^{Bodipy} in manuscript. Only Changes in hydrodynamic diameters of mt-NP^{Bodipy} after NIR light irradiation

(figure 3.i) is shown as advantage over the NP^{Bodipy}, but same data has been not produced for it

Response: Many thanks to the reviewers for their constructive comments. Firstly, we have revised physical characteristics of NP^{Bodipy} and mt-NP^{Bodipy} in the revised manuscript. Moreover, the changes in hydrodynamic diameters, PDI, and Zeta potential of NP^{Bodipy} after NIR light irradiation was evaluated (Fig. R7). After light irradiation 5 min (Fig. R7a, b), the particle sizes were enlarged (~170 nm) owing to the scission of the ROS-activatable thioketal bonds. Importantly, after light irradiation 10 min (Fig. R7a, b), the larger particle sizes (~200 nm) and PDI (~0.4) were observed. However, the Zeta potential of NP^{Bodipy} after NIR light irradiation both 5 and 10 min were maintained at -20 mV.

Fig. R7. Physical parameters of NP^{Bodipy} after light irradiation. **a** DLS profiles of NP^{Bodipy} after light irradiation 5 min. **b** Hydrodynamic diameters and PDI of NP^{Bodipy} after light irradiation 5 min. **c** Zeta potentials of mt- NP^{Bodipy} after light irradiation 5 min. **d** DLS profiles of NP^{Bodipy} after light irradiation 10 min. **e** Hydrodynamic

diameters and PDI of NP^{Bodipy} after light irradiation 10 min. f Zeta potentials of mt-NP^{Bodipy} after light irradiation 10 min.

[10] Similarly the hydrodynamic radius and PDI of both the nanoparticles are also close but DLS spectrum of NP^{Bodipy} does not show the same distribution.

Response: Many thanks to the reviewers for their constructive comments. DLS experiments were performed in three independent replicates, data from only one of which are presented in the manuscript (Fig. R8).

Fig. R8. DLS experiments were performed in three independent replicates.

[11] There is no figure 3k in manuscript but mentioned in text.

Response: Many thanks to the reviewers for their constructive comments. we have checked and revised the figure number and text in the revised manuscript, such as “Notably, the TEM results displayed that the spheric nanoparticles almost disappeared after light irradiation for 10 min, indicating the complete disassociation of mt-NP^{Bodipy} (Fig. 3j).”

[12] No comparative studies has been done for vitro evaluation of NP^{Bodipy} and mt-NP^{Bodipy}. It is not clear at what basis mt-NP^{Bodipy} has been chosen over NP^{Bodipy} for biomedical application. Figure 4(c-g) mentioned as Figure 3 (c-g).

Response: Many thanks to the reviewers for their constructive comments.

(a) No comparative studies has been done for vitro evaluation of NP^{Bodipy} and mt-NP^{Bodipy}.

Response: As depicted in Fig. R9, both NP^{Bodipy} and mt-NP^{Bodipy} exhibited spherical morphology and a uniform size distribution. The average particle sizes of NP^{Bodipy} and mt-NP^{Bodipy} measured by dynamic light scattering (DLS) were around 84 nm and 92 nm (Fig. 3d, e and Supplementary Fig. 17b-d) with polydispersity indexes (PDI) at 0.20 and 0.24, and Zeta potentials (mV) at -20 mV and -11 mV, respectively. The absorption spectra of NP^{Bodipy} and mt-NP^{Bodipy} were measured to show that they had similar UV-Vis properties. Furthermore, the fluorescence (FL) emission spectra of NP^{Bodipy} and mt-NP^{Bodipy} demonstrated that they had maximum emission peaks at 926 and 914 nm, respectively, with a long emission tail extended to 1000-1200 nm. Notably, NP^{Bodipy} and mt-NP^{Bodipy} had a large portion of the emission spectrum in the NIR-II region (>1000 nm), suggesting that the emitted photons could be used for *in vivo* NIR-II fluorescence bioimaging.

Fig. R9. Characterization of mt-NP^{Bodipy} and NP^{Bodipy}. **a** The representative TEM images of mt-NP^{Bodipy}. **b** Hydrodynamic diameters and PDI of mt-NP^{Bodipy}. **c** Zeta potentials of mt-NP^{Bodipy}. **d** Absorption spectra of mt-NP^{Bodipy}. **e** PL spectra of mt-NP^{Bodipy}. **f** The representative TEM images of NP^{Bodipy}. **g** Hydrodynamic diameters and PDI of NP^{Bodipy}. **h** Zeta potentials of NP^{Bodipy}. **i** Absorption spectra of NP^{Bodipy}. **j** PL spectra of NP^{Bodipy}.

(b) It is not clear at what basis mt-NP^{Bodipy} has been chosen over NP^{Bodipy} for biomedical application.

Response: it was reported that positively charged polymers containing primary, secondary, or tertiary amines could serve as adjuvants possibly because the protonated amines could cause endo/lysosomal membrane disruption and pro-inflammation factor release or a stimulator of interferon genes (STING)-dependent pathway. It was reported that particles could cause NLRP3 inflammasome activation and IL-1 β secretion after cellular internalization and escape from endosome. This process mainly induces endosome membrane destruction, cathepsin B release, and K⁺ efflux, which plays a key role in inflammation activation of the host. Li et al. reported mesoporous silica microrods with adsorbed polyethylenimine (PEI) as adjuvant could serve as a cancer vaccine after neoantigen encapsulation (*ACS Nano* **2019**, 13, 3083). In this work, mt-NP^{Bodipy} was obtained by coating NP^{mt} with P^{Bodipy}. The quaternary ammonium cations in mt-NP^{Bodipy} were thus shielded by the negatively charged P^{Bodipy}. This unique design ensured that the quaternary ammonium cations could directly interact with the cell membranes on the one hand, causing their destabilization. In this process, it is expected that immune-enhancing factors will be released by the fragmented cells to enhance the effectiveness of photodynamic immunotherapy.

(c) Figure 4 (c-g) mentioned as Figure 3 (c-g).

Response: we have checked and revised the figure number and text in the revised manuscript, such as “The quantification of the fluorescence signal indicated that mt-NP^{Bodipy}@Cy5.5 irradiated for 10 min demonstrated strong red fluorescence (Fig. 4c, d). These results clearly demonstrated that NIR light could induce ROS generation and trigger the dissociation of mt-NP^{Bodipy} to release P^{mt} to target and destroy the cell membranes.” “The cytotoxicity of the mt-NP^{Bodipy} + L was evaluated against mouse colon cancer cell line CT26 by a 3-(4,5-dimethylthiazol-2-yl)-2,5-diphenyltetrazolium bromide (MTT) assay (Fig. 4e)” “The anticancer activity of mt-NP^{Bodipy} + L was further visualized by 3D tumor cell spheres by CLSM (Fig. 4f).” “As shown in Fig. 4g, compared to PBS, NP^{Bodipy}, and mt-NP^{Bodipy} treated cells with intact

membranes, the cells shrank and the morphology of the membranes became irregular after treatment with NP^{Bodipy} + L.”

[13] In figure 6, the notions are not correct. Figure number 6d has been repeated and scale of x-axis is also wrong.

Response: Many thanks to the reviewers for their constructive comments. we have checked and revised the figure number and the scale of x-axis of fig. 6d in the revised manuscript (Fig. R10). Meanwhile, we have moved some data to supporting information to make the manuscript clearer and revised the notions in Fig. 6.

Fig. R10. (Revised Fig. 6) Imaging and photo-immunotherapy properties of mt-NP^{Bodipy} + L were evaluated in a CT26 tumor-bearing BALB/c mice model. a Schedule diagram in a CT26 tumor-bearing BALB/c mice model treated with mt-NP^{Bodipy} + L. Figure created with Biorender.com. **b** NIR-II fluorescence bioimaging of mice injected with mt-NP^{Bodipy} after various time points *in vivo*. **c** NIR-II fluorescence imaging of major tissues and organs (S, spleen; H, heart; Lu, lung; Ki, kidney; L, liver; T, tumor) after sacrificing the mice at 24 h. **d** Semi-quantitative NIR-II fluorescence analysis in the tumor sites at different times. **e** Semi-quantitative NIR-II fluorescence analysis of organs after 24 h (S, spleen; H, heart; Lu, lung; Ki, kidney; L, liver; T, tumor). **f** Comparison of tumor growth inhibition curves. **g** Monitoring of the weight change of the animal model. **h** FCM plot of CD80⁺CD86⁺ dendritic cells gated on CD11c⁺ cells in the tumorous tissues. **i** Quantification of (h). **j** FCM plot of CD80⁺CD86⁺ dendritic cells gated on CD11c⁺ cells in the lymph nodes. **k** Quantification of (j). **l** FCM plot of CD8⁺ gated on CD3⁺ cells in the tumorous tissues. **m** Quantification of (l). **n** FCM plot of M2 macrophages (CD11b⁺F4/80⁺CD206⁺) in the tumorous tissues. **o** Quantification of (n). Data were expressed as means ± SD. n = 5 biologically independent mice for each group. Data were analyzed by one-way ANOVA with Bonferroni multiple comparisons post-test (i, k, m, and o), and two-way ANOVA with Bonferroni multiple comparisons post-test (f). *p < 0.05, **p < 0.01, ***p < 0.001, ****p < 0.0001.

[14] There are some biological studies in which they have include NP^{Bodipy} datas and in some cases they have excluded them without giving any valid reason.

Response: Many thanks to the reviewers for their constructive comments. We studied the anti-tumor effects of PBS, NP^{Bodipy}, mt-NP^{Bodipy}, NP^{Bodipy} + L, and mt-NP^{Bodipy} + L *in vitro* and *in vivo*. The results showed that mt-NP^{Bodipy} + L had the best effect both *in vivo* and *in vitro*. Therefore, mt-NP^{Bodipy} + L eliminates distant tumors and induces long-term immune memory effects were further researched.

[15] With some presentation errors, this manuscript does not able to justify the reason to synthesize complex organic framework with enhanced properties, instead they are getting similar result with the precursor nanoparticles itself.

Response: Many thanks to the reviewers for their constructive comments. We apologize for the misunderstanding caused by our incorrect statement. We have checked and revised in the revised manuscript.

[16] In Figure 3j, considering the scale difference from 3c, the NPs seem to aggregate rather than dissociate after NIR irradiation. It is better to image the different areas to investigate the disintegration of NPs. And also, it is better to supplement the explanation of the phenomenon of the small bold spots that came up after the irradiation.

Response: Many thanks to the reviewers for their constructive comments. Indeed, we observed nanoparticle dissociation in various regions through TEM (Fig. R11a). Our results were consistent across all observations. The disturbance of the regular nanostructure may account for this phenomenon, resulting in the simultaneous occurrence of small particle dispersion and significant aggregation. This phenomenon has been observed in other studies (Fig. R11b).

Nature Communications **2022**, 13, 3468

Fig. R11. **a** Representative TEM images of different regions. **b** Reference (*Nature Communications* **2022**, 13, 3468) for dissociated nanoparticles.

[17] In the investigation of CT26 (Figure 4), the authors insist that ROS generation triggered the dissociation of NPBodypy. However, the ROS generation is not confirmed at the cellular level. It is better to supplement the ROS generation assays

for clearance. Moreover, the NIR effect should be considered with the control experimental group of non-dissociative NPs.

Response: Many thanks to the reviewers for their constructive comments. To assess the ROS-producing potential of mt-NP^{Bodipy} + L in CT-26 cells, we employed 2',7'-dichlorodihydrofluorescein diacetate (DCFH-DA) as an indicator. DCFH-DA reacts with ROS to produce 2',7'-dichlorofluorescein (DCF), which emits a characteristic green fluorescence (Ex/Em = 488/525 nm). As shown in Fig. R12, CT-26 cells treated with PBS, NP^{Bodipy}, and mt-NP^{Bodipy} exhibited negligible green fluorescence, while those exposed to NP^{Bodipy} + L and mt-NP^{Bodipy} + L displayed strong green fluorescence. This heightened green fluorescence indicates the efficient generation of ROS by NP^{Bodipy} + L and mt-NP^{Bodipy} + L.

Fig. R12. CLSM images showing ROS generation in CT-26 cells *via* a DCFH-DA probe. The green color indicates ROS; the blue color indicates cell nucleus stained by DAPI; scale bar = 5 μ m.

[18] It should be supplemented with an explanation of the immunotherapeutic mechanism of the mt-NP^{Bodipy}.

Response: Many thanks to the reviewers for their constructive comments. As shown in Fig. R13, mt-NP^{Bodipy} + L could produce large amounts of ROS upon continuous light irradiation, resulting in lipid peroxidation and further destabilization of cell membranes, and inducing photodynamic immunotherapy. Specifically, ROS generated by mt-NP^{Bodipy} + L induces ICD in mice can recruit the DCs, promote antigens specific CTLs to the TME, and reprogram the immunosuppressive TME in tumors, to activate the antitumor immune responses.

Fig. R13. Schematic diagram of mt-NP^{Bodipy} + L induced immunotherapy mechanism.

Reviewer #2 (Remarks to the Author):

The manuscript by Tang et al. reported the development of a nanoparticle based on a cell-membrane-targeting polymer and a near-infrared-II and ROS responsive polymer for cancer treatment in a spatial-temporal manner upon light-irradiation. Such particle could also enable NIR-II bio-imaging to track the particle biodistribution.

One major problem of the manuscript is the lack of convincing preclinical benefits for such new formulations. The authors should use other immune-cold tumor models (4T1 or B16F10) to demonstrate the advantage of their new particle formulation. One advantage that can be expected is that the photo therapy could enhance the tumor immunogenicity and recruit more immune cells to these immune-cold tumors. CT26 overall is pretty simple to treat using just checkpoint blockade antibodies (Translational Oncology, Volume 20, 2022, 101405). The benefits of such new formulation need to be further tested in more tumor models.

Another problem is that using such new formulation for imaging studies has not been thoroughly studied. Figure 6b-c showed that the particle could mainly accumulate in both liver and tumor. It is understandable that particle formulations often accumulate in the liver. Nevertheless, the authors need to show how long the particle can stay in the tumor tissues. In addition, the tumor penetration profile of the particle should also be revealed. One concern for cationic particles is that they may stay in the tumor

periphery or just near tumor vasculature and cannot get deep into the tumor, presumably because the ionic interaction between cationic particle and negatively charged extracellular matrix and cells close to the vasculature. The authors should consider perform imaging studies to investigate that; especially upon light irradiation, the disassembled polymer should get deeper into tumor tissues to improve efficacy.

In addition, the authors should consider moving some figures to supplementary information as there are 9 figures in the manuscript. Some figures (e.g., representative tissue histology, flow cytometry gating) can be in the supporting figures.

Response: Many thanks to the reviewers for their constructive comments.

(a) One major problem of the manuscript is the lack of convincing preclinical benefits for such new formulations. The authors should use other immune-cold tumor models (4T1 or B16F10) to demonstrate the advantage of their new particle formulation. One advantage that can be expected is that the photo therapy could enhance the tumor immunogenicity and recruit more immune cells to these immune-cold tumors. CT26 overall is pretty simple to treat using just checkpoint blockade antibodies (Translational Oncology, Volume 20, 2022, 101405). The benefits of such new formulation need to be further tested in more tumor models.

Response: to evaluate the immunogenicity in immune-cold tumor models, a bilateral mice model bearing 4T1 tumors was established. The primary tumor was inoculated on the right flank, followed by the implantation of a second tumor on the left flank six days later, thus simulating distant tumors (Fig. R14a). The results demonstrated that both PBS and mt-NP^{Bodipy} alone did not exhibit noticeable effects on the inhibition of the primary and distant tumors (Fig. R14b, c). However, not only were the primary 4T1 tumors eradicated by mt-NP^{Bodipy} + L, but the growth of distant tumors was also significantly suppressed. These results suggest that mt-NP^{Bodipy} + L possesses the potential to enhance tumor immunogenicity and facilitate the recruitment of more immune cells to these immune-cold tumors.

Fig. R14. mt-NP^{Bodipy} + L eliminates distant tumors in a 4T1 tumor-bearing BALB/c mice model. **a** Schematic treatment schedule of mt-NP^{Bodipy} + L eliminates distant tumors in a 4T1 tumor-bearing BALB/c mice model. Figure created with Biorender.com. **b** Comparison of tumor growth inhibition curves of primary tumors. **c** Comparison of tumor growth inhibition curves of distant tumors.

(b) Another problem is that using such new formulation for imaging studies has not been thoroughly studied. Figure 6b-c showed that the particle could mainly accumulate in both liver and tumor. It is understandable that particle formulations often accumulate in the liver. Nevertheless, the authors need to show how long the particle can stay in the tumor tissues.

Response: In order to examine the retention of mt-NP^{Bodipy} in tumors, mice were intravenously administered with mt-NP^{Bodipy} for a duration of 60 hours for bioimaging. As shown in Fig. R15, the results indicated that the fluorescence intensities in tumors were sustained following the injection of mt-NP^{Bodipy} for the entire 60-hour period.

Fig. 15. a NIR-II fluorescence bioimaging of mice injected with mt-NP^{Bodipy} after various time points *in vivo*. **b** Semi-quantitative NIR-II fluorescence analysis in the tumor sites at different times.

(c) In addition, the tumor penetration profile of the particle should also be revealed. One concern for cationic particles is that they may stay in the tumor periphery or just near tumor vasculature and cannot get deep into the tumor, presumably because the ionic interaction between cationic particle and negatively charged extracellular matrix and cells close to the vasculature. The authors should consider perform imaging studies to investigate that; especially upon light irradiation, the disassembled polymer should get deeper into tumor tissues to improve efficacy.

Response: To unravel whether mt-NP^{Bodipy} + L can extravasate and penetrate deeply into tumours following tumour delivery, we stained platelet endothelial cell adhesion molecule 1 (PECAM1) to visualize the tumour vasculature (Fig. R16). The widespread green immunofluorescence revealed an extensive distribution of blood vessels in CT26 tumours. Free Cy5.5-injected mice exhibited no observable signal in the tumours after 24 h post *i.v.* administration. In stark contrast, mt-NP^{Bodipy}@Cy5.5 were distributed throughout the tumour section after 24 h post *i.v.* administration. Interestingly, most red fluorescence signals from mt-NP^{Bodipy}@Cy5.5 + L were

observed throughout the tumour section, suggestive of a deep tumour penetration capability, which is crucial for anti-tumour efficacy.

Fig. R16. a Schematic of tumour penetration by mt-NP^{Bodipy}@Cy5.5 + L. **b** Investigation of the ability of mt-NP^{Bodipy}@Cy5.5 + L to extravasate and penetrate the tumour after *i.v.* administration into mice with subcutaneous CT26 tumours. 24 h after *i.v.* injection of mt-NP^{Bodipy}@Cy5.5 (red), confocal laser scanning microscopy of sections of CT26 tumours was performed. Blood vessels were marked with PECAM1 antibody followed by Alexa Fluor 488 secondary antibody staining (green). Cell nuclei were stained by DAPI (blue). Scale bars: 25 μ m.

(d) In addition, the authors should consider moving some figures to supplementary information as there are 9 figures in the manuscript. Some figures (e.g.,

representative tissue histology, flow cytometry gating) can be in the supporting figures.

Response: we have moved some data to supporting information to make the manuscript clearer and easier to read. The revised manuscript is shown in Fig. R17 and Fig. R18.

Fig. R17. Imaging and photo-immunotherapy properties of mt-NP^{Bodipy} + L were evaluated in a CT26 tumor-bearing BALB/c mice model. a Schedule diagram in a CT26 tumor-bearing BALB/c mice model treated with mt-NP^{Bodipy} + L. Figure

created with Biorender.com. **b** NIR-II fluorescence bioimaging of mice injected with mt-NP^{Bodipy} after various time points *in vivo*. **c** NIR-II fluorescence imaging of major tissues and organs (S, spleen; H, heart; Lu, lung; Ki, kidney; L, liver; T, tumor) after sacrificing the mice at 24 h. **d** Semi-quantitative NIR-II fluorescence analysis in the tumor sites at different times. **e** Semi-quantitative NIR-II fluorescence analysis of organs after 24 h (S, spleen; H, heart; Lu, lung; Ki, kidney; L, liver; T, tumor). **f** Comparison of tumor growth inhibition curves. **g** Monitoring of the weight change of the animal model. **h** FCM plot of CD80⁺CD86⁺ dendritic cells gated on CD11c⁺ cells in the tumorous tissues. **i** Quantification of (**h**). **j** FCM plot of CD80⁺CD86⁺ dendritic cells gated on CD11c⁺ cells in the lymph nodes. **k** Quantification of (**j**). **l** FCM plot of CD8⁺ gated on CD3⁺ cells in the tumorous tissues. **m** Quantification of (**l**). **n** FCM plot of M2 macrophages (CD11b⁺F4/80⁺CD206⁺) in the tumorous tissues. **o** Quantification of (**n**). Data were expressed as means \pm SD. **n** = 5 biologically independent mice for each group. Data were analyzed by one-way ANOVA with Bonferroni multiple comparisons post-test (**i**, **k**, **m**, and **o**), and two-way ANOVA with Bonferroni multiple comparisons post-test (**f**). * p < 0.05, ** p < 0.01, *** p < 0.001, **** p < 0.0001.

Fig. R18. mt-NP^{Bodipy} + L eliminates distant tumors and induces long-term immune memory effects. **a** Schematic treatment schedule of mt-NP^{Bodipy} + L eliminates distant tumors in a CT26 tumor-bearing BALB/c mice model. Figure created with Biorender.com. **b** Comparison of tumor growth inhibition curves of primary tumors. **c** Comparison of tumor growth inhibition curves of distant tumors. **d** FCM plot of CD80⁺CD86⁺ dendritic cells gated on CD11c⁺ cells in the lymph nodes.

e Quantification of **(d)**. **f** FCM plot of CD8⁺ gated on CD3⁺ cells in the distant tumors. **g** Quantification of **(f)**. **h** Schematic treatment schedule of mt-NP^{Bodipy} + L induce long-term immune memory effects in a CT26 tumor-bearing BALB/c mice model. **i** Average tumor growth curves of the treated mice. **j** survival curves of the treated mice. **k** Relative quantification of effector memory T cells (T_{em}, CD62L⁻CD44⁺) subset from CD8⁺ T cells in the spleen. **l** Relative quantification of central memory T cells (T_{cm}, CD62L⁺CD44⁺) subset from CD8⁺ T cells in the spleen. **m, n** Cytokine levels of TNF- α (**m**) and IFN- γ (**n**) in the serum after tumor rechallenging. Data were expressed as means \pm SD. n = 5 biologically independent mice for each group. Data were analyzed by two-way ANOVA with Bonferroni multiple comparisons post-test (**b, c, and i**), Log-rank (MantelCox) test) (**j**), and one-way ANOVA with Bonferroni multiple comparisons post-test (**e, g, k, l, m, and n**). * $p < 0.05$, ** $p < 0.01$, *** $p < 0.001$, **** $p < 0.0001$.

Other problems:

[1] Figure 6b-c, need fluorescence intensity scale bar. Also please monitor tumor / liver fluorescence change over long time.

Response: Many thanks to the reviewers for their constructive comments.

(a) Figure 6b-c, need fluorescence intensity scale bar.

Response: As shown in Fig. R19, we have checked and revised in the revised manuscript.

Fig. R19. b NIR-II fluorescence bioimaging of mice injected with mt-NP^{Bodipy} after various time points *in vivo*. **c** NIR-II fluorescence imaging of major tissues and organs

(S, spleen; H, heart; Lu, lung; Ki, kidney; L, liver; T, tumor) after sacrificing the mice at 24 h.

(b) Also please monitor tumor / liver fluorescence change over long time.

Response: As depicted in Fig. R20, mice were intravenously administered with mt-NP^{Bodipy} for bioimaging purposes lasting until 60 hours. The results demonstrated that the fluorescence intensities in tumors were sustained throughout the 60-h period. After *in vivo* bioimaging, the mice were subsequently sacrificed, and the tumors and major organs (including heart, liver, lung, kidneys, and spleen) were visualized *ex vivo* for biodistribution. The results showed that the fluorescence intensity inside the tumors was 15.6, 3.3, 3.1, and 3.5 times stronger than those in the heart, spleen, lung, and kidneys respectively, indicating that mt-NP^{Bodipy} had a good tumor-targeting effect.

Fig. R20. **a** NIR-II fluorescence bioimaging of mice injected with mt-NPBodipy after various time points *in vivo*. **b** Semi-quantitative NIR-II fluorescence analysis in the tumor sites at different times. **c** NIR-II fluorescence imaging of major tissues and organs (S, spleen; H, heart; Lu, lung; Ki, kidney; L, liver; T, tumor) after sacrificing the mice at 60 h. **d** Semi-quantitative NIR-II fluorescence analysis of organs after 24 h (S, spleen; H, heart; Lu, lung; Ki, kidney; L, liver; T, tumor).

[2] Figure 6d. The x axis should be time (hours)? Instead of wavelength?

Response: As shown in Fig. R21, we have checked and revised in the revised manuscript.

Fig. R21. Semi-quantitative NIR-II fluorescence analysis in the tumor sites at different times.

[3] Figure 3, whether ROS could affect the fluorescence light emission of the dye TPA-BODIPY? The ROS is generated upon light irradiation on the dye, and the dye also emits fluorescence. How does the energy transfer occur in such process (Figure 2d only showing the energy from S1 to T1 for photodynamic therapy).

Response: Many thanks to the reviewers for their constructive comments. As shown in Fig. R22, there is a competitive relationship between ROS, fluorescence, and non-radiative relaxation. This manifestation is commonly present in photosensitizers (Fig. R23). In fact, the ROS generation of Aza-TPA-Bodipy was even more efficient (16.4% quantum yield) than ICG (7.7% quantum yield). Meanwhile, the fluorescence quantum yield of Aza-TPA-Bodipy was even more efficient (1.1% quantum yield) than IR26 (0.5% quantum yield, commercial fluorescent dyes).

Fluorescence quantum yield detection: The fluorescence quantum yield of Aza-TPA-Bodipy was determined against the reference fluorophore IR dye 26 with a known quantum yield of 0.5%. The optical absorbance was measured for both an Aza-TPA-Bodipy and an IR dye 26 solution at 808 nm. Then their NIR-II fluorescence emission intensities were measured under the same 808 nm excitation. Using the measured optical density (OD) and spectrally integrated fluorescence

intensity (F), one can calculate the quantum yield of Aza-TPA-Bodipy according to the following formula,

$$\begin{aligned} \Phi_X(\lambda) &= \Phi_{st}(\lambda) \cdot \frac{F_x}{F_{st}} \cdot \frac{A_{st}(\lambda)}{A_x(\lambda)} \\ &= \Phi_{st}(\lambda) \cdot \frac{F_x}{F_{st}} \cdot \frac{1 - 10^{-OD_{st}(\lambda)}}{1 - 10^{-OD_x(\lambda)}} \\ &= 0.5\% \cdot \frac{428859208 \text{ cps}}{81393600 \text{ cps}} \cdot \frac{1 - 10^{-0.2}}{1 - 10^{-1.07}} \\ &= 1.1\% \end{aligned}$$

Fig. R22. Illustration of photochemical mechanisms.

Adv. Mater. **2021**, 2103748.

Fig. R23. Schematic diagram of ROS and fluorescence generated by photosensitizers.

[4] Figure 7. The percentage of immune cells in what cell types are not clear. For example, (b-e) the percentage of CD80+ CD86+ is on CD11c+ DCs or CD11c+MHC-II+CD45+ DCs in tumors / lymph nodes? (f-g) is percentage based on CD3+ lymphocytes or CD45+ total tumor infiltration lymphocytes? (h-i) is based on CD45+ total tumor infiltration lymphocytes? The authors also need to provide the total cell population numbers for each treatment, because the cell numbers are also important to tell the outcome of immunotherapy (e.g., whether the total number of DCs and CD8+CD3+ T cells increased in tumor infiltration lymphocytes). Same problems also occur in Figure 8c-f and Figure 9f-h.

Response: Many thanks to the reviewers for their constructive comments. As

shown in Fig. R24, we have checked and revised the flow analysis results in the revised manuscript. Meanwhile, flow cytometry analysis revealed that the intratumoural density of CD80⁺CD86⁺ cells in mice LNDs treated with mt-NP^{Bodipy} + L in bilateral CT26 tumor-bearing mice model was around 2.5-fold and 2.2-fold of the densities in mice treated with mt-NP^{Bodipy} and PBS, respectively (Fig. R25a). A similar trend was observed in the percentage of CD8⁺ T cells among CD3⁺ T cells. Specifically, the intratumoural density of CD8⁺ cells in mice tumors treated with mt-NP^{Bodipy} + L in bilateral CT26 tumor-bearing mice model was around 2.0-fold and 1.7-fold of the densities in mice treated with mt-NP^{Bodipy} and PBS, respectively (Fig. R25b)

Fig. R24. The revised the flow analysis results.

Fig. R25. a Intratumoural densities of CD80⁺CD86⁺ cells. **b** Intratumoural densities of CD8⁺ T lymphocytes.

[5] The n numbers are too low for flow cytometry studies in Figures 7-9, please consider increasing n numbers to 5 for comparison.

Response: Many thanks to the reviewers for their constructive comments. Based on the suggestions provided by the reviewer and relevant immunotherapy literature, the flow cytometry study was repeated, and the number of mice was increased to 5.

[6] Figure 9, for memory T cell analysis, the spleen results often could not reflect the memory T cells population changes. Please take on treated tumor to see (i) whether effector memory CD44CD8 T cells increases (Nature volume 520, pages373–377 (2015)). (ii) whether effector and central memory CD8 T cells increase in lymph nodes.

Response: Many thanks to the reviewers for their constructive comments. Based on the suggestions provided by the reviewer and relevant immunotherapy literature (Nature volume 520, pages373–377 (2015)), the flow cytometry study was repeated, and research was conducted on the effector and central memory CD8 T cells in lymph nodes (Fig. R26). The description of memory T cell analysis has been revised in the revised manuscript. Such as “To further verify the immunological memory responses induced by mt-NP^{Bodipy} + L, the lymph nodes of mice were collected at 76 days to determine changes in memory T cells. Using FCM on these LNDs, we observed an

increased percentage of T_{EM} and central memory T cells (T_{CM}) in the $CD3^+CD8^+$ T cells for the mice that survived to 76 days compared with that in the naïve mice (Fig. 9f-h).

Fig. R26. a Relative quantification of effector memory T cells (T_{em} , $CD62L^{\text{low}}CD44^{\text{hi}}$) subset from $CD8^+$ T cells in the lymph nodes. **b** Relative quantification of central memory T cells (T_{cm} , $CD62L^{\text{hi}}CD44^{\text{low}}$) subset from $CD8^+$ T cells in the lymph nodes.

[7] The authors did not study other immunosuppressive cells change over the course of treatment. For example, the Treg cells and MDSC cells are not examine. In addition, for $CD8CD3$ T cells, whether these cells have viability ($ki67^+$) and functionality ($IFN\text{-}\gamma^+$ or $TNF\text{-}\alpha^+$) are not studied. The problem that T cells become exhaustion upon treatment is a serious problem for cancer immunotherapy.

Response: Many thanks to the reviewers for their constructive comments. In this work, $mt\text{-}NP^{\text{Bodipy}} + L$ not only eradicated the primary CT26 tumors but also remarkably suppressed the distant tumor growth. These results indicated that $mt\text{-}NP^{\text{Bodipy}} + L$ further activated the T cells and enhanced the $CD8^+$ T cell infiltration in the tumor. In other words, $mt\text{-}NP^{\text{Bodipy}} + L$ could reprogram the immunosuppressive microenvironment, thereby increasing the infiltration of CTLs and reducing the proportion of negative regulatory immune cells. Therefore, the MDSC (myeloid-derived suppressor cells) at the distant CT26 tumor tissues were analyzed by flow cytometry. The results showed that MDSC in the distant tumor of mice treated with $mt\text{-}NP^{\text{Bodipy}} + L$ as compared to PBS decreased from 68.2% to 53.3%, which indicated that $mt\text{-}NP^{\text{Bodipy}} + L$ could reprogram the immunosuppressive microenvironment (Fig. R27a, b). Moreover, Moreover, the frequency of tumor-infiltrating $IFN\text{-}\gamma^+$ $CD3^+$

CD8⁺ T cells in distant CT26 tumor tissues treated with mt-NP^{Bodipy} + L was 7-fold higher than that in tissues treated with PBS (Fig. R27c, d). These results demonstrated that mt-NP^{Bodipy} + L elicited potent antitumor immunity to eliminate established tumors.

Fig. R27. **a** Representative flow cytometric plots of MDSC (CD11b⁺ Gr-1⁺) cells in CD45⁺ cells at the distant CT26 tumor tissues. **b** Percentages of MDSC (CD11b⁺ Gr-1⁺) cells in CD45⁺ cells at the distant CT26 tumor tissues. **c** Representative flow cytometric plots of IFN-γ⁺ cells in CD3⁺ CD8⁺ T cells at the distant CT26 tumor tissues. **d** Percentages of IFN-γ⁺ cells in CD3⁺ CD8⁺ T cells at the distant CT26 tumor tissues. Data are represented as mean ± SD (**p* < 0.05, ***p* < 0.01, ****p* < 0.001, *n* = 5, The data were analyzed by one-way ANOVA with Bonferroni multiple comparisons post-test).

[8] All NMR spectra in Supplementary figures did not have integration and peak assignments. Please assign all peaks for all NMR spectra.

Response: Many thanks to the reviewers for their constructive comments. As shown in Fig. R28-37, we have checked and revised the NMR spectra in the revised manuscript.

Fig. R28. ¹H NMR spectrum of M3 in DMSO-*d*₆.

Fig. R29. ¹H NMR spectrum of M4 in DMSO-*d*₆.

Fig. R30. ¹H NMR spectrum of M5 in DMSO-*d*₆.

Fig. R31. ¹H NMR spectrum of M6 in DMSO-*d*₆.

Fig. R32. ¹H NMR spectrum of M7 in DMSO-*d*₆.

Fig. R33. ^1H NMR spectrum of Aza-TPA-Bodipy in $\text{DMSO-}d_6$.

Fig. R34. ^1H NMR spectrum of P^{Bodipy} in DMSO- d_6 .

Fig. R35. ^1H NMR spectrum of N,N-dimethyl-cholesterol in CDCl_3 .

Fig. R36. ^1H NMR spectrum of P^{Br} in $\text{DMSO-}d_6$.

Fig. R37. 1H NMR spectrum of P^{mt} in $DMSO-d_6$.

[9] The size exclusion chromatography for all polymers are missing. No molecular weights and molecular weight distribution information for all polymers.

Response: Many thanks to the reviewers for their constructive comments. As shown in Fig. R38 and R39, we have added the information of molecular weights and molecular weight distribution information for P^{Bodipy} and P^{mt} .

Fig. R38. Gel permeation chromatography (GPC) curve of P^{Bodipy} in DMF.

Fig. R39. Gel permeation chromatography (GPC) curve of P^{mt} in DMF.

[10] For DFT computation, please give the computation methods information (functionals, basis sets, etc.). Such information is critical especially for the excited state calculation. Molecule coordinates are also missing in supplementary information.

Some typos (please proofread the manuscript):

Response: Many thanks to the reviewers for their constructive comments. We have checked and revised the information of DFT computation in the revised manuscript. Such as “

Computational Details. The density functional theory (DFT) calculations were performed using a DMOL3 module of Material Studio 2016.⁴¹ The generalized gradient approximation (GGA) method with Perdew-Burke-Ernzerhof (PBE) function was employed to optimize the structure and calculate the properties as orbitals and optical. The force and energy convergence criterion were set to 0.02 eV Å⁻¹ and 10⁻⁵ eV, respectively. Energy levels of S₁-S_n and T₁-T_n are calculated by the vertical excitation of the above optimized structures, with the same method of B3LYP/6-31G(d).

Table 1. Cartesian coordinates of optimized ground state calculated by the DFT, B3LYP/6-31G(d), Gaussian 16 program.

Atom	x	y	z
C	-4.10633	7.77056	0.1619
N	-4.02095	6.38343	0.1711
B	-2.82209	5.57168	-0.39307
N	-1.61888	6.55118	-0.41886
C	-1.74762	7.93703	-0.48838
C	-3.00965	8.52266	-0.29444
C	-5.3758	8.1332	0.74783
C	-6.01248	6.94753	1.04981
C	-5.17941	5.86673	0.66278
C	-0.30797	6.2145	-0.55714
C	0.43966	7.41617	-0.67644
C	-0.42183	8.49046	-0.6381
C	-5.92848	9.47965	1.1081
C	-5.44224	4.46292	0.68855
C	0.04813	9.91416	-0.62322
C	0.18504	4.86821	-0.54215

F	-2.53732	4.49533	0.46471
F	-3.10484	5.0952	-1.67157
C	-6.58197	3.89111	1.147
C	-6.8552	2.46923	1.10429
C	1.46913	4.56746	-0.85809
C	2.13266	3.28348	-0.80216
C	-7.86679	1.90298	1.90103
C	-8.09567	0.53634	1.92892
C	-7.32781	-0.32584	1.13103
C	-6.35858	0.23739	0.27885
C	-6.12441	1.59503	0.27532
C	3.46662	3.19857	-1.24356
C	4.1888	2.02421	-1.17255
C	3.61282	0.863	-0.62849
C	2.27037	0.92345	-0.20511
C	1.55244	2.10293	-0.29494
C	0.74241	-6.43475	-3.76498
C	0.46781	-5.11178	-4.10301
C	-0.5752	-4.43274	-3.48149
C	-1.35273	-5.05512	-2.49393
C	-1.07034	-6.3898	-2.17157
C	-0.03769	-7.0737	-2.80389
C	-2.49435	-4.33783	-1.87448
C	-3.3159	-3.52016	-2.81057
C	-2.78084	-4.40277	-0.54732
C	-4.04571	-3.85174	-0.0019
C	-1.82795	-4.9127	0.47361
C	-3.59675	-2.1768	-2.52867
C	-4.36152	-1.41064	-3.40065
C	-4.86138	-1.97492	-4.57321
C	-4.57365	-3.30463	-4.87406
C	-3.7974	-4.0664	-4.00609
C	-2.24121	-5.84523	1.43402
C	-1.37994	-6.25136	2.4467
C	-0.1024	-5.7028	2.53946
C	0.31268	-4.7598	1.60338
C	-0.53712	-4.37903	0.56961
C	-5.2857	-4.12557	-0.59415
C	-6.4392	-3.47613	-0.17854
C	-6.3744	-2.53074	0.84786
C	-5.16246	-2.32799	1.51632
C	-4.01645	-2.98568	1.10005
N	-7.49437	-1.71924	1.17262
C	-8.76776	-2.31976	1.34154
C	-8.89043	-3.49129	2.10097
C	-10.12105	-4.10502	2.2563
C	-11.26742	-3.55397	1.67295
C	-11.15575	-2.38397	0.91934
C	-9.91015	-1.78364	0.74792

O	-12.43365	-4.22474	1.89476
C	-13.62783	-3.7126	1.32046
C	-14.74841	-4.64551	1.72993
O	-15.93415	-4.11984	1.15038
O	0.9291	-7.81311	0.13409
C	2.27973	-7.84228	-0.27868
C	3.02316	-6.57355	0.08332
O	2.43722	-5.52955	-0.69167
C	2.82453	-1.99195	-1.38716
C	2.36736	-3.30014	-1.41435
C	2.91815	-4.26234	-0.55977
C	3.89991	-3.88221	0.35671
C	4.33493	-2.56036	0.39683
C	3.82463	-1.60574	-0.48392
N	4.38216	-0.29862	-0.49659
C	8.01638	-0.86761	-1.02396
C	6.64032	-0.93188	-1.19036
C	5.79588	-0.19667	-0.35414
C	6.35363	0.61645	0.63606
C	7.72953	0.67105	0.79977
C	10.5034	1.1104	-1.97831
C	11.24892	1.30093	-3.13697
C	12.35708	0.49706	-3.39587
C	12.71047	-0.50015	-2.4885
C	11.97154	-0.68215	-1.32555
C	8.96923	-1.76702	2.52253
C	8.22712	-2.169	3.62693
C	8.32361	-1.47022	4.82916
C	9.18301	-0.37746	4.92145
C	9.9346	0.01631	3.81901
C	10.86387	0.1288	-1.04556
C	8.58654	-0.08626	-0.00988
C	10.05831	-0.06345	0.18889
C	9.82344	-0.65879	2.5967
C	10.61876	-0.21972	1.41839
C	14.15773	-0.64869	2.69528
C	12.80782	-0.87221	2.44593
C	12.05926	0.03339	1.68202
C	12.69363	1.18904	1.20737
C	14.03971	1.41887	1.46579
C	14.77987	0.49734	2.20436
C	-3.21433	9.97432	-0.65592
F	-4.47172	10.22441	-1.0678
F	-2.42149	10.36068	-1.6739
F	-2.96481	10.8178	0.3816
H	-6.97979	6.85704	1.52179
H	1.51634	7.48286	-0.71928
H	-6.67096	9.36559	1.90018
H	-5.15899	10.16273	1.46826

H	-6.41713	9.96276	0.26026
H	-4.64827	3.83022	0.3187
H	-0.49534	10.52543	0.09738
H	1.10579	9.94199	-0.3546
H	-0.06108	10.39194	-1.59814
H	-0.51327	4.09678	-0.24719
H	-7.35122	4.51522	1.59648
H	2.10917	5.38072	-1.19219
H	-8.45849	2.54916	2.54264
H	-8.85694	0.12537	2.58084
H	-5.79501	-0.40714	-0.38338
H	-5.38329	1.995	-0.40673
H	3.94109	4.08293	-1.65811
H	5.21103	1.99655	-1.52726
H	1.80626	0.04274	0.22173
H	0.5303	2.11856	0.06702
H	1.55119	-6.96703	-4.25399
H	1.06531	-4.60685	-4.85505
H	-0.79688	-3.40903	-3.76187
H	-1.68142	-6.89863	-1.4355
H	0.15505	-8.10626	-2.53725
H	-3.21484	-1.7379	-1.61433
H	-4.56425	-0.37039	-3.16774
H	-5.4623	-1.37906	-5.25193
H	-4.95222	-3.74913	-5.78874
H	-3.56732	-5.09863	-4.24746
H	-3.24668	-6.24899	1.38027
H	-1.70673	-6.99286	3.16784
H	0.56609	-6.01252	3.33545
H	1.2983	-4.3138	1.67329
H	-0.2081	-3.64762	-0.15881
H	-5.33358	-4.81484	-1.4284
H	-7.38377	-3.6608	-0.67689
H	-5.12219	-1.61515	2.33178
H	-3.07573	-2.79943	1.60578
H	-8.00867	-3.91784	2.56482
H	-10.22335	-5.01184	2.84108
H	-12.02176	-1.93879	0.44746
H	-9.82871	-0.88607	0.14652
H	-13.83463	-2.70049	1.68589
H	-13.55298	-3.67851	0.22786
H	-14.52753	-5.65749	1.36628
H	-14.80757	-4.68033	2.82557
H	-16.67261	-4.68898	1.3879
H	0.53998	-6.98683	-0.18334
H	2.7444	-8.69237	0.22648
H	2.36591	-7.99973	-1.3629
H	2.90516	-6.36237	1.15169
H	4.09168	-6.66177	-0.15091

H	2.41855	-1.26489	-2.08086
H	1.60668	-3.61526	-2.11754
H	4.33395	-4.59908	1.04116
H	5.10347	-2.27255	1.10432
H	8.66361	-1.43688	-1.68196
H	6.20764	-1.55779	-1.96233
H	5.70114	1.19698	1.27787
H	8.1499	1.2885	1.58423
H	9.63576	1.7315	-1.78406
H	10.96248	2.07585	-3.84052
H	12.93618	0.64098	-4.30197
H	13.56415	-1.13948	-2.68794
H	12.25296	-1.4551	-0.62049
H	8.88729	-2.30917	1.58799
H	7.57451	-3.03286	3.5509
H	7.74072	-1.78062	5.68992
H	9.27047	0.16818	5.85526
H	10.60987	0.86136	3.89698
H	14.72414	-1.36855	3.2771
H	12.32417	-1.75991	2.83855
H	12.12344	1.906	0.62891
H	14.51214	2.32088	1.09099
H	15.83145	0.67551	2.40322

Table 2. Cartesian coordinates of optimized S_1 excited state calculated by the DFT, B3LYP/6-31G(d), Gaussian 16 program.

Atom	x	y	z
C	3.73724	7.91101	-0.10824
N	3.72322	6.53768	-0.10384
B	2.58921	5.6727	0.50922
N	1.33447	6.58271	0.52584
C	1.3866	7.96072	0.58873
C	2.61771	8.61794	0.39582
C	4.95203	8.34822	-0.736
C	5.6493	7.19964	-1.0552
C	4.89446	6.08128	-0.64443
C	0.03423	6.18049	0.66998
C	-0.76535	7.34203	0.7965
C	0.04188	8.45434	0.74543
C	5.39522	9.72708	-1.10846
C	5.2286	4.69655	-0.6818
C	-0.48088	9.85493	0.7265
C	-0.4024	4.82193	0.63713
F	2.35535	4.56367	-0.3096
F	2.92209	5.25371	1.78732
C	6.38476	4.17105	-1.18071
C	6.72648	2.77169	-1.12819

C	-1.68098	4.46249	0.95762
C	-2.30435	3.16717	0.86771
C	7.77553	2.25365	-1.91936
C	8.09953	0.91039	-1.91451
C	7.39456	0.0153	-1.0884
C	6.37704	0.52558	-0.2592
C	6.05171	1.86185	-0.2848
C	-3.63212	3.03689	1.32519
C	-4.33183	1.85502	1.22595
C	-3.74424	0.72182	0.63539
C	-2.40722	0.81937	0.19836
C	-1.7087	2.0101	0.31653
C	-0.08923	-7.50311	3.13107
C	0.03124	-6.23191	3.6875
C	1.00675	-5.35609	3.22643
C	1.87265	-5.7219	2.18319
C	1.74611	-7.01032	1.63989
C	0.77754	-7.89163	2.10983
C	2.92728	-4.78564	1.72506
C	3.61488	-4.03798	2.81103
C	3.24085	-4.60218	0.40231
C	4.43442	-3.84459	-0.0304
C	2.3823	-5.07708	-0.71598
C	3.7385	-2.64192	2.7598
C	4.35973	-1.94325	3.78883
C	4.87354	-2.62673	4.88931
C	4.74555	-4.01225	4.96003
C	4.11119	-4.70964	3.93757
C	2.94156	-5.77445	-1.79589
C	2.1559	-6.17065	-2.87313
C	0.80137	-5.85092	-2.90601
C	0.23682	-5.13801	-1.85017
C	1.01706	-4.76107	-0.76109
C	5.69817	-4.02925	0.55952
C	6.78325	-3.24899	0.19589
C	6.62968	-2.24981	-0.77584
C	5.39479	-2.10657	-1.42567
C	4.32247	-2.89751	-1.06346
N	7.6753	-1.35518	-1.08395
C	8.98616	-1.8466	-1.24284
C	9.20541	-3.02363	-1.97605
C	10.48233	-3.52791	-2.12644
C	11.5784	-2.86355	-1.56071
C	11.37062	-1.6892	-0.83207
C	10.08233	-1.19433	-0.66965
O	12.7865	-3.43126	-1.77322
C	13.92593	-2.80244	-1.22094
C	15.1244	-3.63961	-1.61364
O	16.24868	-2.99811	-1.05666

O	-0.95892	-7.86892	-0.53873
C	-2.28565	-7.92065	-0.0856
C	-3.04653	-6.6409	-0.36038
O	-2.445	-5.62774	0.43049
C	-2.8887	-2.13128	1.26989
C	-2.39944	-3.4273	1.23466
C	-2.95265	-4.37188	0.35931
C	-3.97758	-3.98536	-0.50591
C	-4.44775	-2.67734	-0.48236
C	-3.92992	-1.73885	0.41427
N	-4.49935	-0.4461	0.48001
C	-8.11781	-1.07555	1.00399
C	-6.74038	-1.14512	1.14989
C	-5.90761	-0.35498	0.35012
C	-6.48275	0.5092	-0.58823
C	-7.86079	0.566	-0.73106
C	-10.52079	0.79537	2.18674
C	-11.20557	0.8771	3.3947
C	-12.30856	0.06106	3.62971
C	-12.71925	-0.83745	2.6483
C	-12.04063	-0.91109	1.43718
C	-9.26043	-1.71959	-2.62116
C	-8.60192	-2.06534	-3.796
C	-8.73927	-1.27686	-4.9355
C	-9.55383	-0.14852	-4.89482
C	-10.22252	0.18952	-3.72303
C	-10.93699	-0.08792	1.1821
C	-8.70702	-0.2361	0.04742
C	-10.18482	-0.1778	-0.0996
C	-10.07115	-0.57929	-2.56183
C	-10.79676	-0.21248	-1.31454
C	-14.38685	-0.39528	-2.51149
C	-13.0424	-0.69414	-2.31835
C	-12.2371	0.11027	-1.50106
C	-12.81166	1.24295	-0.91088
C	-14.15265	1.54856	-1.11308
C	-14.9476	0.7273	-1.90857
C	2.77086	10.05304	0.78773
F	4.02459	10.3356	1.17646
F	1.98478	10.37922	1.82678
F	2.47301	10.92244	-0.20955
H	6.60782	7.1655	-1.55754
H	-1.84643	7.35902	0.84714
H	6.11135	9.66734	-1.934
H	4.56015	10.35545	-1.43134
H	5.88546	10.24568	-0.27784
H	4.47754	4.02656	-0.27708
H	0.04212	10.48162	-0.00132
H	-1.54222	9.84063	0.45977

H	-0.38722	10.34623	1.70028
H	0.32824	4.08215	0.32613
H	7.10923	4.82539	-1.66356
H	-2.3453	5.24499	1.32282
H	8.32055	2.92648	-2.57733
H	8.88585	0.53685	-2.56356
H	5.85766	-0.14255	0.42044
H	5.28162	2.22534	0.38851
H	-4.117	3.89915	1.77738
H	-5.34906	1.79629	1.59938
H	-1.93221	-0.03991	-0.2659
H	-0.69047	2.05873	-0.06011
H	-0.84638	-8.191	3.49739
H	-0.63446	-5.92155	4.48876
H	1.10492	-4.371	3.67529
H	2.42325	-7.32383	0.85085
H	0.70121	-8.88306	1.6718
H	3.33606	-2.10641	1.90427
H	4.43608	-0.86017	3.73581
H	5.36167	-2.08071	5.69217
H	5.13649	-4.55238	5.81853
H	4.00083	-5.78915	4.00309
H	4.00316	-6.00888	-1.78198
H	2.60377	-6.72766	-3.69183
H	0.18942	-6.15242	-3.75154
H	-0.81528	-4.86391	-1.87596
H	0.57108	-4.20574	0.05954
H	5.81541	-4.77482	1.33997
H	7.74507	-3.38238	0.68251
H	5.28607	-1.35072	-2.19811
H	3.36911	-2.77159	-1.56892
H	8.36086	-3.53463	-2.42866
H	10.66211	-4.43528	-2.69456
H	12.1998	-1.16158	-0.37357
H	9.92354	-0.29207	-0.08683
H	14.04038	-1.78281	-1.6125
H	13.85287	-2.74808	-0.12653
H	14.99256	-4.66109	-1.22665
H	15.17845	-3.69827	-2.711
H	17.03185	-3.51109	-1.28986
H	-0.5562	-7.07827	-0.14771
H	-2.77136	-8.74188	-0.62336
H	-2.34226	-8.14056	0.99264
H	-2.97137	-6.38656	-1.42658
H	-4.1076	-6.75017	-0.09194
H	-2.47798	-1.4168	1.97746
H	-1.60796	-3.74652	1.9073
H	-4.41804	-4.69075	-1.20221
H	-5.24637	-2.38384	-1.15709

H	-8.75505	-1.68135	1.64283
H	-6.29701	-1.81028	1.88523
H	-5.84041	1.13288	-1.20307
H	-8.29419	1.2328	-1.47042
H	-9.65404	1.4276	2.01134
H	-10.87385	1.57852	4.15585
H	-12.84102	0.11972	4.57523
H	-13.57111	-1.48823	2.82733
H	-12.36503	-1.61324	0.67462
H	-9.15288	-2.33816	-1.73463
H	-7.98425	-2.95951	-3.82386
H	-8.22195	-1.54532	-5.85268
H	-9.67333	0.46971	-5.7807
H	-10.86807	1.06373	-3.69933
H	-14.9971	-1.03945	-3.13922
H	-12.60566	-1.5645	-2.80131
H	-12.19573	1.88714	-0.29018
H	-14.57732	2.43533	-0.64995
H	-15.99638	0.96567	-2.06429

Table 3. Cartesian coordinates of optimized T_1 excited state calculated by the DFT, B3LYP/6-31G(d), Gaussian 16 program.

Atom	x	y	z
C	4.07202	7.72591	-0.19961
N	3.98943	6.36487	-0.18438
B	2.83328	5.56693	0.47663
N	1.63012	6.54095	0.51914
C	1.75783	7.90346	0.55664
C	3.02181	8.51128	0.34678
C	5.29576	8.09974	-0.8976
C	5.92071	6.9236	-1.23039
C	5.12945	5.83097	-0.77589
C	0.2933	6.20133	0.69331
C	-0.43522	7.41945	0.81819
C	0.43083	8.47902	0.72447
C	5.76126	9.46569	-1.29336
C	5.3867	4.45291	-0.80516
C	0.02002	9.91729	0.67531
C	-0.20038	4.88188	0.68656
F	2.51428	4.44698	-0.31654
F	3.18788	5.14279	1.75678
C	6.50435	3.85411	-1.33691
C	6.76721	2.44399	-1.27767
C	-1.5075	4.57597	0.98218
C	-2.15568	3.29712	0.91107
C	7.78356	1.86154	-2.06677
C	8.01741	0.49876	-2.06971

C	7.25147	-0.35502	-1.25595
C	6.27504	0.22055	-0.41649
C	6.03569	1.57446	-0.43531
C	-3.5059	3.20223	1.31512
C	-4.21598	2.02466	1.22762
C	-3.61585	0.86246	0.70733
C	-2.26129	0.93007	0.32208
C	-1.55361	2.11283	0.42578
C	-0.66482	-6.2942	4.02864
C	-0.38839	-4.95695	4.30315
C	0.63335	-4.30001	3.62517
C	1.38729	-4.96023	2.64385
C	1.10375	-6.30868	2.3859
C	0.09235	-6.9698	3.07429
C	2.50875	-4.26551	1.96527
C	3.34321	-3.39564	2.84119
C	2.76788	-4.39186	0.63686
C	4.01817	-3.85845	0.0422
C	1.79827	-4.95197	-0.34092
C	3.59839	-2.06372	2.48935
C	4.375	-1.24779	3.3039
C	4.91276	-1.75032	4.48771
C	4.65091	-3.06786	4.85801
C	3.86263	-3.87918	4.04771
C	2.19675	-5.92721	-1.26445
C	1.31901	-6.38033	-2.24244
C	0.03922	-5.8376	-2.33837
C	-0.3615	-4.85335	-1.4393
C	0.50485	-4.42488	-0.43847
C	5.27066	-4.10064	0.62228
C	6.41133	-3.4605	0.16022
C	6.31904	-2.55693	-0.90073
C	5.09535	-2.39031	-1.5567
C	3.96224	-3.0402	-1.09462
N	7.42587	-1.74438	-1.27008
C	8.69642	-2.34454	-1.46219
C	8.80195	-3.52772	-2.20587
C	10.02961	-4.14112	-2.38207
C	11.18871	-3.57924	-1.83443
C	11.09306	-2.39834	-1.09511
C	9.85117	-1.79722	-0.90352
O	12.34951	-4.25102	-2.07375
C	13.55735	-3.72895	-1.5366
C	14.66896	-4.66648	-1.95975
O	15.86686	-4.13032	-1.4165
O	-0.92735	-7.83826	0.19298
C	-2.27313	-7.84951	0.62256
C	-3.01949	-6.59595	0.21647
O	-2.42467	-5.5218	0.94252

C	-2.82175	-1.9639	1.51853
C	-2.36294	-3.26961	1.59121
C	-2.90713	-4.26075	0.76612
C	-3.88359	-3.91242	-0.16859
C	-4.3194	-2.59339	-0.25619
C	-3.81662	-1.60885	0.59638
N	-4.37498	-0.30355	0.56129
C	-8.02696	-0.85277	0.97912
C	-6.6575	-0.9199	1.19144
C	-5.78336	-0.20378	0.36926
C	-6.30411	0.59291	-0.65367
C	-7.67365	0.64977	-0.86342
C	-10.53113	1.15399	1.81695
C	-11.31168	1.36925	2.94788
C	-12.43184	0.57568	3.18609
C	-12.76212	-0.436	2.28605
C	-11.98805	-0.64275	1.15042
C	-8.87384	-1.81366	-2.58074
C	-8.09903	-2.23944	-3.65334
C	-8.15129	-1.56067	-4.86967
C	-8.99944	-0.46402	-5.00854
C	-9.78372	-0.04646	-3.93813
C	-10.8675	0.15734	0.89118
C	-8.56003	-0.08842	-0.06781
C	-10.02461	-0.06145	-0.31357
C	-9.71709	-0.70118	-2.70157
C	-10.54672	-0.23646	-1.55734
C	-14.04567	-0.66704	-2.9394
C	-12.70585	-0.89407	-2.64317
C	-11.97644	0.02023	-1.87116
C	-12.61821	1.18757	-1.43679
C	-13.95379	1.42078	-1.74225
C	-14.67578	0.49094	-2.48833
C	3.27887	9.90614	0.79094
F	4.56657	10.09374	1.16521
F	2.52959	10.24778	1.86547
F	3.02669	10.86679	-0.15607
H	6.85017	6.83605	-1.77445
H	-1.50972	7.49711	0.89543
H	6.47138	9.38879	-2.11908
H	4.93295	10.09807	-1.61784
H	6.25481	9.98791	-0.47179
H	4.61725	3.82507	-0.37881
H	0.56296	10.46928	-0.09377
H	-1.04642	9.98759	0.45217
H	0.20087	10.4312	1.62094
H	0.49807	4.10023	0.41856
H	7.24137	4.46527	-1.85201
H	-2.15322	5.38955	1.30231

H	8.37306	2.49894	-2.71885
H	8.7799	0.07959	-2.71478
H	5.71363	-0.41385	0.25698
H	5.2911	1.97895	0.23958
H	-3.99692	4.08493	1.71319
H	-5.24768	1.99066	1.55337
H	-1.78004	0.05017	-0.08729
H	-0.52203	2.1327	0.09246
H	-1.45674	-6.80899	4.56191
H	-0.96758	-4.42359	5.04987
H	0.85652	-3.26438	3.85599
H	1.6982	-6.8459	1.65642
H	-0.1019	-8.01351	2.85672
H	3.18609	-1.67302	1.56633
H	4.55686	-0.2169	3.01818
H	5.5225	-1.11563	5.12183
H	5.0589	-3.46388	5.78224
H	3.65262	-4.90158	4.34323
H	3.20365	-6.32731	-1.20954
H	1.63464	-7.15389	-2.93424
H	-0.64206	-6.18376	-3.10807
H	-1.34872	-4.41153	-1.51278
H	0.18668	-3.66169	0.26177
H	5.33803	-4.75478	1.48295
H	7.36611	-3.61649	0.64882
H	5.03546	-1.71001	-2.39834
H	3.01113	-2.88168	-1.59013
H	7.90987	-3.96267	-2.64122
H	10.11995	-5.05649	-2.95526
H	11.96916	-1.94577	-0.64978
H	9.78166	-0.89187	-0.31241
H	13.75307	-2.72224	-1.92216
H	13.50904	-3.67904	-0.44322
H	14.45863	-5.67332	-1.57617
H	14.70167	-4.71711	-3.05587
H	16.60104	-4.70093	-1.66359
H	-0.53517	-6.99944	0.47124
H	-2.74397	-8.71939	0.15857
H	-2.34646	-7.96173	1.71334
H	-2.91142	-6.42968	-0.8607
H	-4.08572	-6.67208	0.46464
H	-2.42192	-1.21488	2.19192
H	-1.60534	-3.55965	2.30828
H	-4.3128	-4.65218	-0.8314
H	-5.08221	-2.33075	-0.9793
H	-8.69814	-1.40597	1.62669
H	-6.25311	-1.53213	1.98927
H	-5.62801	1.15874	-1.28403
H	-8.0654	1.25436	-1.67226

H	-9.65466	1.7675	1.63893
H	-11.04337	2.1555	3.64595
H	-13.03839	0.73896	4.07067
H	-13.62529	-1.06722	2.46986
H	-12.25163	-1.42673	0.45067
H	-8.82679	-2.34053	-1.63509
H	-7.45555	-3.10626	-3.5416
H	-7.54293	-1.8896	-5.70555
H	-9.05251	0.06606	-5.95382
H	-10.44999	0.80159	-4.0524
H	-14.59779	-1.39348	-3.52674
H	-12.21556	-1.79118	-3.00495
H	-12.06225	1.9111	-0.8527
H	-14.43217	2.33191	-1.39835
H	-15.71928	0.67181	-2.72387

[11] In abstract: “In this study, Herein, we report a cell membrane-targeting...” please remove either “in this study” or “herein”

Response: Many thanks to the reviewers for their constructive comments. We have checked and revised in the revised manuscript. Such as “Herein, we report a cell membrane-targeting cationic polymer with antineoplastic activity (P^{mt}) and a second near-infrared (NIR-II) fluorescent biodegradable polymer with photosensitizer Bodipy units and reactive oxygen species (ROS) responsive thioketal bonds (P^{Bodipy}).”

[12] Figure 9 legend “Data are represented as mean \pm SD in g, h, I, and j...” should be “Data are represented as mean \pm SD in g, h, i, and j...”

Response: Many thanks to the reviewers for their constructive comments. We have checked and revised in the revised manuscript.

Reviewer #3 (Remarks to the Author):

In this MS, Tang et al. report the cationic polymer nanoparticles that enable photodynamic therapy and immunotherapy of cancer as well as NIR-II bioimaging. However, there are numerous studies on cancer treatment using various nanomaterials with multiple functions (such as PTT, PDT, chemotherapy, immunotherapy, etc.), and the therapeutic effects in this manuscript do not seem to differ from these studies. The design of integrating different functions into nanoparticles for cancer treatment is

quite common. In addition, the experimental results did not adequately support the material design that the negatively charged PBodipy can shield the quaternary ammonium cation of Pmt and reduce the toxic side effects of cationic polymer in bio-applications. Therefore, this manuscript is not suitable for publication in Nature Communications.

[1] Fig.1 was not cited in the main text.

Response: Many thanks to the reviewers for their constructive comments. We have checked and revised the figure number and text in the revised manuscript, such as “Herein, we developed cationic antineoplastic anticancer nanoparticles (denoted as mt-NP^{Bodipy}) that can specifically anchor and destabilize cell membranes upon NIR light irradiation for PDT, photo-immunotherapy, and NIR-II fluorescence bio-imaging (Fig.1).” Meanwhile, we also have checked and revised the figure caption in the revised manuscript, such as “**Fig. 1 Schematic illustration of nanoparticles destabilizing the cell membranes triggered by NIR light.** P^{Bodipy} with NIR-II fluorescent Bodipy units and ROS-responsive thioketal linkages can be excited by NIR light to induce ROS, resulting in subsequent breakage of the thioketal bonds and the degradation of the polymer. P^{mt} contains numerous cholesterol molecules for cell membrane targeting and quaternary ammonium salts for cell membrane destabilization. Firstly, P^{mt} was self-assembled to NP^{mt}, which was subsequently shielded by P^{Bodipy} into mt-NP^{Bodipy}. Under excitation with a laser at 808 nm, mt-NP^{Bodipy} was dissociated and the cationic polymers P^{mt} were released. Through electrostatic interactions, P^{mt} can target and insert cell membranes (i, ii) and then destabilize and lyse the cell membranes (iii, iv), causing tumor cell death.”

[2] The zeta potential of NP^{mt}, and that of mt-NP^{Bodipy} after light irradiation should be investigated and included in Fig.3e.

Response: Many thanks to the reviewers for their constructive comments. As shown in Fig. R40 a, the Zeta potentials (mV) of NP^{mt} measured by dynamic light scattering (DLS) were approximately 35 mV. Furthermore, the zeta potential of mt-NP^{Bodipy} after 5 and 10 minutes of light irradiation (808 nm, 1 W cm⁻²), measured by DLS, were approximately -2.6 mV and -0.1 mV, respectively (Fig. R40 b). These

results suggest that $\text{mt-NP}^{\text{Bodipy}} + \text{L}$ causes the dissociation of nanoparticles, leading to the release of NP^{mt} .

Fig. R40. **a** Zeta potential of NP^{mt} . **b** The zeta potential of $\text{mt-NP}^{\text{Bodipy}}$ after 5 and 10 minutes of light irradiation (808 nm , 1 W cm^{-2}).

[3] Investigate the stability of $\text{mt-NP}^{\text{Bodipy}}$ as the TME (such as pH and ROS...) may dissociate nanoparticles.

Response: Many thanks to the reviewers for their constructive comments. The stability of $\text{mt-NP}^{\text{Bodipy}}$ in simulating tumor microenvironment was investigated. As shown in Fig. R41a, b, the result indicated $\text{mt-NP}^{\text{Bodipy}}$ in pH 5.0 solution 24 h exhibited a uniform size distribution according to DLS. However, $\text{mt-NP}^{\text{Bodipy}}$ in 10 mM H_2O_2 solution 24 h displaying a bimodal distribution and a large PDI (~ 0.7) (Fig. R41c, d).

Fig. R41. **a, b** Hydrodynamic diameters and PDI of $\text{mt-NP}^{\text{Bodipy}}$ in pH 5.0 solution 24 h. **c, d** Hydrodynamic diameters and PDI of $\text{mt-NP}^{\text{Bodipy}}$ in 10 mM H_2O_2 solution 24 h.

[4] There is no evidence to support the authors' assumption that the negatively charged PBodipy can shield the quaternary ammonium cation from Pmt and reduce the toxic side effects of cationic polymer in bio-applications. The NPmt group should be included in relevant experiments both in vitro and in vivo.

Response: Many thanks to the reviewers for their constructive comments. As shown in Fig. R42, than 5% hemolysis rate observed in NP^{mt} at 50 mg ml⁻¹. These results indicated that NP^{mt} is difficult to administer directly through the tail vein, which can cause significant toxic side effects..

Fig. R42. **a** Photographs of erythrocytes after incubation with 1× PBS buffer (negative control), 1% Triton X-100 (positive control), and mt-NP^{Bodipy}, mt-NP^{Bodipy} + L 5 min, and NP^{mt} at the different concentration for 2 h, followed by centrifugation. **b** Hemolysis percentages of erythrocytes after incubation with mt-NP^{Bodipy}, mt-NP^{Bodipy} + L 5 min, and NP^{mt} at different concentrations for 2 h (n = 3).

[5] After 808-nm laser induced dissociation, does the released cationic Pmt circulate to other tissues and cause toxic side effects?

Response: Many thanks to the reviewers for their constructive comments. In order to study the safety of released cationic P^{mt}, the H&E staining and blood

biochemical indexes of mice treated with NP^{Bodipy} + L and mt-NP^{Bodipy} + L were observed. As shown in Fig. R43, no noticeable histopathological damage or abnormal blood biochemical indexes were observed in the different treatment groups, indicating minimal side effects of NP^{Bodipy} + L and mt-NP^{Bodipy} + L.

Fig. R43. Biosafety evaluation of NP^{Bodipy} + L and mt-NP^{Bodipy} + L *in vivo*. **a** H&E staining of major organs (heart, liver, spleen, lung, and kidney) after different treatment. **b** Biochemical analysis of serum: alanine aminotransferase (ALT), aspartate aminotransferase (AST), alkaline phosphatase (ALP), creatinine (CRE), and blood urea nitrogen (BUN). n = 3. Data are presented as mean ± SD.

[6] The optical properties of mt-NPBodipy are notably different from those of Aza-TPA-Bodipy. During *in vivo* imaging, 808-nm laser irradiation can cause mt-NPBodipy dissociation, affecting tumor imaging analysis (Fig.6b-d).

Response: Many thanks to the reviewers for their constructive comments. A difference in the optical properties between mt-NP^{Bodipy} and Aza-TPA-Bodipy does exist. However, both mt-NP^{Bodipy} and Aza-TPA-Bodipy exhibit a large portion of their emission spectrum in the NIR-II region (>1000 nm). This suggests that the emitted photons can be utilized for *in vivo* NIR-II fluorescence bioimaging. To investigate the impact of 808 nm laser irradiation on mt-NP^{Bodipy}, NP@Aza-TPA-Bodipy was intravenously administered to mice for bioimaging at various time points (Fig. R44). The results demonstrated a continuous increase in fluorescence intensities within tumors following the injection of NP@Aza-TPA-Bodipy, with the highest fluorescence observed at 24 h. Subsequently, the mice were sacrificed, and *ex vivo* visualization was performed on the tumors and major organs (including the heart, liver, lung, kidneys, and spleen) to assess biodistribution. The findings revealed that the fluorescence intensity within the tumors was 9.5, 3.7, 2.8, and 2.8 times stronger compared to the heart, spleen, lung, and kidneys, respectively. This indicates a favorable tumor-targeting effect of NP@Aza-TPA-Bodipy. Despite slight variations in the fluorescence spectra of mt-NP^{Bodipy} and Aza-TPA-Bodipy, they exhibit similar tumor enrichment and are effective for NIR II imaging of tumors. The above results also indicate that the imaging of mt-NP^{Bodipy} is not affected by the dissociation of nanoparticles caused by 808 nm light, due to the inability of NP@Aza-TPA-Bodipy to dissociate.

Fig. R44. *In vivo* distribution of NP@Aza-TPA-Bodipy. **a** Schematic illustration of NP@Aza-TPA-Bodipy. **b** NIR-II fluorescence bioimaging of mice injected with NP@Aza-TPA-Bodipy after various time points *in vivo*. **c** Semi-quantitative NIR-II fluorescence analysis in the tumor sites at different times **d**. NIR-II fluorescence imaging of major tissues and organs (S, spleen; H, heart; Lu, lung; Ki, kidney; L, liver; T, tumor) after sacrificing the mice at 24 h. **e** Semi-quantitative NIR-II fluorescence analysis of organs after 24 h (S, spleen; H, heart; Lu, lung; Ki, kidney; L, liver; T, tumor).

[7] Has the thermal effect caused by 808-nm laser affected the photodynamic immunotherapy? For example, Nat. Commun. 2021,12:742 and J. Am. Chem. Soc. 2021, 143, 8116–8128.

Response: Many thanks to the reviewers for their constructive comments. To validate the photothermal of mt-NP^{Bodipy} *in vivo*, mt-NP^{Bodipy} was intravenous injected into the mice bearing CT26 tumor and then laser irradiation was conducted. After 808 nm laser photoirradiation, the tumor surface temperatures of mice treated with PBS

and mt-NP^{Bodipy} were below 37 °C. (Fig. R45). These results indicated that mt-NP^{Bodipy} has no thermal effect under illumination *in vivo*.

Nat. Commun. **2021**, *12*, 742

Fig. R45. Maximum surface tumor temperature of CT26 tumor-bearing mice during 808 nm photoirradiation. **a** Photothermal heating curve from CT26 tumor-bearing mice treated with PBS. **b** Photothermal heating curve from CT26 tumor-bearing mice treated with mt-NP^{Bodipy}. **c** Photothermal heating curve from related references.

[8] The authors need to specify other parameters for bioimaging application, such as power density, filter and exposure time, etc.

Response: Many thanks to the reviewers for their constructive comments. Accordingly, we have added the text in the revised manuscript, such as:

“Optical System for NIR-II Fluorescence. An 808 nm diode laser (Artemis, China) was used as the excitation light. A two-dimensional InGaAs camera (Princeton Instruments, U.S.) with 640 pixels × 512 pixels was used for capturing all NIR-II images. A NIR lens (Artemis, China) was used to focus the image onto the photodetector. The emission filter of all NIR-II images was 1000 nm. The exposure time of all NIR-II images was 50 ms. All NIR- II fluorescence images were analyzed by the software (ImageJ).”

[9] The protocol of preparing mt-NP^{Bodipy}@cy5.5 should be provided. After laser-induced dissociation of mt-NP^{Bodipy}@cy5.5, is there any free cy5.5 that interferes with cell labeling?

Response: Many thanks to the reviewers for their constructive comments. Accordingly, we have added the protocol of preparing mt-NP^{Bodipy}@cy5.5 in the revised manuscript (Fig. R46), such as “Preparation of mt-NP^{Bodipy}@Cy5.5. Cy5.5 NHS (1 mg) was dissolved in 1 mL of DMSO, N,N-dimethylethylenediamine (0.2 mg) was added, and the solution was stirred for 2 h at room temperature. Next, the mixture and P^{mt} (20 mg) in dry DMF (3 mL) were stirred at 50 °C for 24 h under a nitrogen atmosphere. The mixture was added to 5 mL of deionized water under sonication, followed by dialysis in a dialysis bag (MWCO: 8000-14000 Da). After 72 h, the solution was freeze-dried under reduced pressure to obtain P^{mt}@Cy5.5. P^{mt}@Cy5.5 (10 mg) was first dissolved in 1 mL of DMSO, which was then dropped into 9 mL of water during the period of magnetic stirring. Then, the mixture was dialyzed in a dialysis bag (MWCO: 8000-14000 Da) to obtain NP^{mt}@Cy5.5. After 72 h, the mixture was concentrated to 5 mL through ultrafiltration. Next, P^{Bodipy} dissolved in ethanol was evaporated under a rotatory evaporator to generate a thin film, which was further dried under an ultra-high vacuum for 0.5 h. The film was hydrated with NP^{mt}@Cy5.5 for 30 min and then sonicated for 12 min by using a pulse 3/2 s on/off at a power output of 60 W, followed by filtration using a 0.22 μm syringe-driven filter to give mt-NP^{Bodipy}@Cy5.5.”

Fig. R46. Synthetic route of $P^{mt}@Cy5.5$.

[10] There are two d in Fig.6 and the abscissa “Wavelength (nm)” is incorrect. Also, Figs.6f-j show the same information as Fig.6e and should be part of the supporting data. Please check all figures to make the manuscript clearer and easier to read.

Response: Many thanks to the reviewers for their constructive comments. As shown in Fig. R47, we have checked and revised the Fig. 6e in the revised manuscript. Meanwhile, considering the suggestions of all reviewers, we have moved some data to supporting information to make the manuscript clearer and easier to read (Fig. R48).

Fig. R47 (Fig. 6 in revised manuscript) Imaging and photo-immunotherapy properties of mt-NP^{Bodipy} + L were evaluated in a CT26 tumor-bearing BALB/c mice model. **a** Schedule diagram in a CT26 tumor-bearing BALB/c mice model treated with mt-NP^{Bodipy} + L. Figure created with Biorender.com. **b** NIR-II fluorescence bioimaging of mice injected with mt-NP^{Bodipy} after various time points *in vivo*. **c** NIR-II fluorescence imaging of major tissues and organs (S, spleen; H, heart; Lu, lung; Ki, kidney; L, liver; T, tumor) after sacrificing the mice at 24 h. **d** Semi-quantitative NIR-II fluorescence analysis in the tumor sites at different times. **e** Semi-quantitative NIR-II fluorescence analysis of organs after 24 h (S, spleen; H, heart; Lu, lung; Ki, kidney; T, tumor). **f** Tumor volume (mm³) over 16 days for five groups (I-V). **g** Body weight (% change) over 14 days for five groups (I-V). **h, j, l, n** Flow cytometry plots for CD86-APC vs CD80-FITC, CD86-APC vs CD80-FITC, CD8-FITC vs CD3-PE, and CD206-APC vs F4/80-PE, respectively, for groups I-V. **i, k, m, o** Bar graphs quantifying the percentage of CD86⁺CD80⁻ in CD11C⁺ Dcs, CD86⁺CD80⁻ in CD11C⁺ Dcs, CD8⁺ in CD3⁺ T cells in tumor, and CD206⁺ in F4/80⁺ cells in tumor, respectively, for groups I-V.

L, liver; T, tumor). **f** Comparison of tumor growth inhibition curves. **g** Monitoring of the weight change of the animal model. **h** FCM plot of CD80⁺CD86⁺ dendritic cells gated on CD11c⁺ cells in the tumorous tissues. **i** Quantification of (**h**). **j** FCM plot of CD80⁺CD86⁺ dendritic cells gated on CD11c⁺ cells in the lymph nodes. **k** Quantification of (**j**). **l** FCM plot of CD8⁺ gated on CD3⁺ cells in the tumorous tissues. **m** Quantification of (**l**). **n** FCM plot of M2 macrophages (CD11b⁺F4/80⁺CD206⁺) in the tumorous tissues. **o** Quantification of (**n**). Data were expressed as means ± SD. n = 5 biologically independent mice for each group. Data were analyzed by one-way ANOVA with Bonferroni multiple comparisons post-test (**i**, **k**, **m**, and **o**), and two-way ANOVA with Bonferroni multiple comparisons post-test (**f**). * $p < 0.05$, ** $p < 0.01$, *** $p < 0.001$, **** $p < 0.0001$.

Fig. R48 (Fig. 7 in revised manuscript) mt-NP^{Bodipy} + L eliminates distant tumors and induces long-term immune memory effects. **a** Schematic treatment schedule of mt-NP^{Bodipy} + L eliminates distant tumors in a CT26 tumor-bearing BALB/c mice model. Figure created with Biorender.com. **b** Comparison of tumor growth inhibition curves of primary tumors. **c** Comparison of tumor growth inhibition curves of distant tumors. **d** FCM plot of CD80⁺CD86⁺ dendritic cells gated on CD11c⁺ cells in the

lymph nodes. **e** Quantification of **(d)**. **f** FCM plot of CD8⁺ gated on CD3⁺ cells in the distant tumors. **g** Quantification of **(f)**. **h** Schematic treatment schedule of mt-NP^{Bodipy} + L induce long-term immune memory effects in a CT26 tumor-bearing BALB/c mice model. **i** Average tumor growth curves of the treated mice. **j** survival curves of the treated mice. **k** Relative quantification of effector memory T cells (T_{em}, CD62L⁻CD44⁺) subset from CD8⁺ T cells in the spleen. **l** Relative quantification of central memory T cells (T_{cm}, CD62L⁺CD44⁺) subset from CD8⁺ T cells in the spleen. **m, n** Cytokine levels of TNF- α (**m**) and IFN- γ (**n**) in the serum after tumor rechallenging. Data were expressed as means \pm SD. n = 5 biologically independent mice for each group. Data were analyzed by two-way ANOVA with Bonferroni multiple comparisons post-test (**b**, **c**, and **i**), Log-rank (MantelCox) test (**j**), and one-way ANOVA with Bonferroni multiple comparisons post-test (**e**, **g**, **k**, **l**, **m**, and **n**). * p < 0.05, ** p < 0.01, *** p < 0.001, **** p < 0.0001.

REVIEWER COMMENTS

Reviewer #1 (Remarks to the Author):

In this revision, although the authors have developed a controllable system using PBodipy with a ROS-cleavable linker to regulate Pmt release, the manuscript still need to be improved to be published such a high profile journal such as nature communication. Additionally, this referee still raise several questions to be convinced about the results. Unfortunately, this reviewer recommends for author to submit this article in other journal.

Comment 1: The treatment of normal cells with mt-NPBodipy nanoparticles, when activated by NIR light to induce ROS, raises important questions about potential cytotoxic effects on healthy tissues and the need for precision in targeting cancer cells. (reference: Nature communications 10.1 (2019): 3349). And

Comment 2: The investigation into the long-term effects of mt-NPBodipy nanoparticles in vivo, including studies on potential chronic toxicity and biocompatibility, is crucial to ensure their safety and effectiveness. (reference: Nature Communications 13.1 (2022): 7149.)

Comment 3: How does the effective penetration depth of NIR light in tissue impact the applicability of this therapy, particularly for deep-seated tumors ?

Comment 4: In Figure 4 (e), the ability to reduce cell viability of NPBodipy + L and mt-NPBodipy + L appears similar. Although the authors insist that the IC50 value was lower for mt-NPBodipy + L than for NPBodipy + L, Author need analyze the significancy of the data.

Comment 5: In figure 4f, the 3D tumor spheroids of CT26 cells treated with mt-NPBodipy + L appear torn; could this be attributed to the action of Pmt, and if so, additional images should be included in the supplementary data to further elucidate this observation..

Comment 6: In Figure 6 (g), the body weight of the rats in the mt-NPBodipy + L group appears to be lower than that of the other groups. Are there any side effects caused by mt-NPBodipy + L?

Comment 7: In Figure 6 (h)-(n), there is no significant difference in the quantitative results like the proportion of mature DC cells, CD8+ T cells, and M2 macrophages between the mt-NPBodipy + L group and the NPBodipy + L group. Authors need to statistically analyze the tumor volume to insist significantly difference in the mt-NPBodipy + L group in Figure 6 (f)?

Comment 8: is there any evident to indicate that mt-NPBodipy has an effect on membrane lysis in RNA-sequencing data?

Comment 9: Author also need to explain in detail why the NPmt were dissociated to Pmt in after NIR light irradiation figure 1. Should the mt-NPBodipy be turned NPmt not Pmt after irradiation?

Reviewer #2 (Remarks to the Author):

The manuscript has addressed the reviewer's concerns. I suggest that the manuscript should be accepted.

Reviewer #3 (Remarks to the Author):

The technical content of this revised version has been largely improved. Most of my concerns are addressed. I now recommend its publication.

Response for Nature Communications manuscript NCOMMS-23-18572A

Reviewer #1 (Remarks to the Author):

In this revision, although the authors have developed a controllable system using PBodipy with a ROS-cleavable linker to regulate Pmt release, the manuscript still need to be improved to be published such a high profile journal such as nature communication. Additionally, this referee still raise several questions to be convinced about the results. Unfortunately, this reviewer recommends for author to submit this article in other journal.

Comment 1: The treatment of normal cells with mt-NPBodipy nanoparticles, when activated by NIR light to induce ROS, raises important questions about potential cytotoxic effects on healthy tissues and the need for precision in targeting cancer cells. (reference: Nature communications 10.1 (2019): 3349).

Response: Many thanks to the reviewers for their constructive comments. Photodynamic therapy (PDT) is a spatiotemporally controllable antitumor approach that can produce ROS to damage the structure of proteins, DNA, and membranes (*Adv. Mater.*, **2022**, 34, 2207174.). PDT has unique capabilities compared with those of other ablative modalities used in the treatment of cancer (such as surgery, cryoablation, microwave ablation, radiofrequency ablation and brachytherapy). The use of photosensitizers that accumulate in cancerous tissues provides a degree of additional treatment selectivity. The control of light placement further minimizes off-target toxicity to surrounding tissues (*Nat. Rev. Clin. Oncol.*, **2020**, 17, 657-674).

In this work, we found that mt-NP^{Bodipy} can efficiently accumulate at the tumor site. After tail vein administration of mt-NP^{Bodipy}, continuous monitoring of mouse status revealed the animals behaved normally without any signs of pain or stress and did not lose or gain any weight. Subsequently, an 808 nm laser was used to only irradiate the tumor site, causing tumor cell death without causing damage to other organs. The results showed that under 808 nm laser irradiation, the weight loss of mice treated with mt-NP^{Bodipy} + L did not exceed 6%, and a significant improvement in mouse status was observed. On the 14th day, the animals behaved normally without any signs of pain or stress and did not lose or gain any weight.

Fig. R1. **a** NIR-II fluorescence bioimaging of mice injected with mt-NP^{Bodipy} after various time points *in vivo*. **b** NIR-II fluorescence imaging of major tissues and organs (S, spleen; H, heart; Lu, lung; Ki, kidney; L, liver; T, tumor) after sacrificing the mice at 24 h. **c** Semi-quantitative NIR-II fluorescence analysis of organs after 24 h. **d** Monitoring of the weight change of the animal model.

Comment 2: The investigation into the long-term effects of mt-NP^{Bodipy} nanoparticles *in vivo*, including studies on potential chronic toxicity and biocompatibility, is crucial to ensure their safety and effectiveness. (reference: Nature Communications 13.1 (2022): 7149.)

Response: Many thanks to the reviewers for their constructive comments. The biodegradability of nanocarriers holds significant importance in the field of cancer diagnosis and therapy. Certain functional materials lack degradability, which can result in the accumulation of toxicity within the organism when multiple injections are administered. This factor, to some extent, hinders the future clinical applicability. However, recent years have witnessed remarkable advancements in stimulus-responsive polymers, enabling targeted drug delivery and site-specific release of therapeutic agents. These polymers respond to external stimuli such as light, magnetic fields, or ultrasound, as well as internal factors like pH value, temperature, enzyme activity, or redox potential. In the case of traditional stimulus-responsive

polymers, a stimulus interacts with sensitive bonds, consequently resulting in polymer degradation. This not only enhances biocompatibility but also reduces toxicity (*Biomaterials*, **2022**, 289, 121795. *Nat. Commun.*, **2020**, 11, 5828).

In this work, we synthesized a NIR-II fluorescent biodegradable polymer (P^{Bodipy}) containing NIR-II fluorescent Bodipy units with PDT effect and ROS-responsive thioketal linkages in the polymer main chain and pendant negative charged pair-wised carboxylic acids. Under excitation with an 808 nm laser, P^{Bodipy} can generate ROS, leading to the cleavage of thioketal bonds and polymer degradation. As a result, our nanomedicine holds the potential to enhance biocompatibility and safety. After tail vein administration of mt- NP^{Bodipy} , continuous monitoring of mouse status revealed the animals behaved normally without any signs of pain or stress and did not lose or gain any weight. To studies on potential chronic toxicity and biocompatibility of mt- NP^{Bodipy} , H&E staining and blood biochemical indicators were detected on the 14th day. As shown in Figure R2, no noticeable histopathological damage or abnormal blood biochemical indexes were observed in the different treatment groups, indicating minimal side effects of NP^{Bodipy} and mt- NP^{Bodipy} .

Fig. R2. Biosafety evaluation of NP^{Bodipy} and mt-NP^{Bodipy} *in vivo*. **a** H&E staining of major organs (heart, liver, spleen, lung, and kidney) after different treatment. **b** Biochemical analysis of serum: alanine aminotransferase (ALT), aspartate aminotransferase (AST), alkaline phosphatase (ALP), creatinine (CRE), and blood urea nitrogen (BUN). n = 3. Data are presented as mean \pm SD.

Comment 3: How does the effective penetration depth of NIR light in tissue impact the applicability of this therapy, particularly for deep-seated tumors ?

Response: Many thanks to the reviewers for their constructive comments. To date, several photosensitizers of PDT, such as protoporphyrin IX, chlorin e6, and methylene blue, have been clinically approved for the treatment of skin, esophageal, and lung tumors. However, the main absorption spectra of these photosensitizers are still located in the visible region <700 nm, which largely attenuates the therapeutic effect in deep tissues because of the intense light absorption and scattering within biological tissues. Near-infrared (NIR) light in the biological transparency window of 700-1000 nm displays deeper body penetration and minimal tissue absorption, and thus much attention has been recently focused on developing NIR photosensitizers for deep PDT treatment, including boron dipyrromethene (BODIPY) molecules, cyanine derivatives, metal-organic complexes, and aggregation-induced emission (AIE) compounds (*Small* **2022**, 18, 2204851).

Fig. R3. NIR activated photodynamic.

Comment 4: In Figure 4 (e), the ability to reduce cell viability of NPBodypy + L and mt-NPBodypy + L appears similar. Although the authors insist that the IC50 value was lower for mt-NPBodypy + L than for NPBodypy + L, Author need analyze the significance of the data.

Response: Many thanks to the reviewers for their constructive comments. As shown in Fig. R4, we had analyzed the significance of the cell viability of NP^{Bodypy} + L and mt-NP^{Bodypy} + L. Data were analyzed by two-way ANOVA with Bonferroni multiple comparisons post-test. * $p < 0.05$, ** $p < 0.01$, *** $p < 0.001$, **** $p < 0.0001$.

Fig. R4. Drug-response curves upon treatment of CT26 cells with mt-NP^{Bodipy} + L. Data were expressed as means \pm SD. $n = 4$ biologically independent mice for each group. Data were analyzed by two-way ANOVA with Bonferroni multiple comparisons post-test. $*p < 0.05$, $**p < 0.01$, $***p < 0.001$, $****p < 0.0001$.

Comment 5: In figure 4f, the 3D tumor spheroids of CT26 cells treated with mt-NPBodipy + L appear torn; could this be attributed to the action of Pmt, and if so, additional images should be included in the supplementary data to further elucidate this observation.

Response: Many thanks to the reviewers for their constructive comments. In Figure 4f, the three-dimensional tumor spheroids from CT26 cells treated with mt-NPBodipy + L appear disrupted, potentially due to irregularities in the three-dimensional cell culture. Actually, the anticancer activity of mt-NP^{Bodipy} + L was further visualized by 3D tumor cell spheres by CLSM. The live cells labeled with Calcein AM (green) and Dead cells marked by probium iodide (red). The results showed that 3D tumor cell spheres treated with NP^{Bodipy} + L and mt-NP^{Bodipy} + L had strong red fluorescence (cells dead), indicating NP^{Bodipy} + L and mt-NP^{Bodipy} + L could cause significant cell death.

Comment 6: In Figure 6 (g), the body weight of the rats in the mt-NPBodipy + L group appears to be lower than that of the other groups. Are there any side effects caused by mt-NPBodipy + L?

Response: Many thanks to the reviewers for their constructive comments. Under excitation with a laser at 808 nm, mt-NP^{Bodipy} was dissociated and the cationic polymers P^{mt} were released. Through electrostatic interactions, P^{mt} can target and insert cell membranes and

then destabilize and lyse the cell membranes, causing tumor cell death. We hypothesize that this process will initially cause a temporary decrease in mouse weight. However, due to the rapid release of P^{mt} under light, the resulting weight loss was quickly reversed (Figure 6g). Simultaneously, we performed H&E analysis on the main organs of mice at 14 days, including the heart, liver, spleen, lungs, and kidneys. As shown in Fig. R5, no noticeable histopathological damage was observed in the various treatment groups, indicating minimal side effects of mt-NP^{Bodipy} + L.

Fig. R5. Drug-response curves upon treatment of CT26 cells with mt-NP^{Bodipy} + L. Data were expressed as means \pm SD. $n = 4$ biologically independent mice for each group. Data were analyzed by two-way ANOVA with Bonferroni multiple comparisons post-test. $*p < 0.05$, $**p < 0.01$, $***p < 0.001$, $****p < 0.0001$.

Comment 7: In Figure 6 (h)-(n), there is no significant difference in the quantitative results like the proportion of mature DC cells, CD8⁺ T cells, and M2 macrophages between the mt-NP^{Bodipy} + L

group and the NP^{Bodipy} + L group. Authors need to statistically analyze the tumor volume to insist significantly difference in the mt-NP^{Bodipy} + L group in Figure 6 (f)?

Response: Many thanks to the reviewers for their constructive comments. In this work, we developed cationic antineoplastic anticancer nanoparticles (denoted as mt-NP^{Bodipy}) that can specifically anchor and destabilize cell membranes upon NIR light irradiation for PDT, photo-immunotherapy, and NIR-II fluorescence bio-imaging. This work provided a paradigm for the use of external spatiotemporally controllable stimuli to expose cationic charge-shielded nanoparticles with cholesterol to specifically target and anchor cell membranes, thereby generating large amounts of ROS, resulting in cell membrane destabilization for cancer therapy. Actually, we observed that both mt-NP^{Bodipy} + L and NP^{Bodipy} + L treatments induced robust immune responses, with no significant difference between the two. However, the light-induced release of P^{mt} in mt-NP^{Bodipy} led to notable tumor regression. These findings suggest that cationic polymers also demonstrate antineoplastic activity by disrupting cell membranes *in vivo*. Meanwhile, we have added a discussion in the revised manuscript, such as “Moreover, a partial tumor inhibitory effect was observed in mice treated with NP^{Bodipy} + L. The tumor inhibition rate of NP^{Bodipy} + L with a tumor inhibition rate at 68%. Notably, the tumor inhibition rate of mt-NP^{Bodipy} + L exceeded 90%, indicating that the tumor growth in mice treated with mt-NP^{Bodipy} + L was effectively inhibited. These findings suggest that cationic polymers also demonstrate antineoplastic activity by disrupting cell membranes *in vivo*”. Finally, as depicted in Figure 6f, we conducted a statistical analysis of the tumor volume in mice treated with mt-NP^{Bodipy} + L.

Comment 8: is there any evident to indicate that mt-NP^{Bodipy} has an effect on membrane lysis in RNA-sequencing data?

Response: Many thanks to the reviewers for their constructive comments. As shown in Fig. R6, Gene Ontology (GO) enrichment analysis showed that the CT26 cells treated with mt-NP^{Bodipy} + L mainly displayed influenced expressions of genes in the major pathways of cellular responses to DNA damage stimulus, DNA repair, DNA replication, **membrane fusion**, and double-strand break repair in Biological Process. We hypothesize that this may be attributed to the successful release of cationic polymer P^{mt} by mt-NP^{Bodipy} + L, leading to the destruction of cell membranes.

Fig. R6. GO categorization of Biological Process after mt-NP^{Bodipy} + L treatment.

Comment 9: Author also need to explain in detail why the NPmt were dissociated to Pmt in after NIR light irradiation figure 1. Should the mt-NPBodipy be turned NPmt not Pmt after irradiation?

Response: Many thanks to the reviewers for their constructive comments. As shown in Fig. R7, the results of mt-NP^{Bodipy} + L are more similar to NP^{mt} compared to P^{mt}. Therefore, we speculate that mt-NP^{Bodipy} be turned NP^{mt} after irradiation.

Fig. R7. DLS of NP^{mt}, P^{mt}, mt-NP^{Bodipy}, and mt-NP^{Bodipy} + L.

Reviewer #2 (Remarks to the Author):

The manuscript has addressed the reviewer's concerns. I suggest that the manuscript should be accepted.

Reviewer #3 (Remarks to the Author):

The technical content of this revised version has been largely improved. Most of my concerns are addressed. I now recommend its publication.

REVIEWERS' COMMENTS

Reviewer #1 (Remarks to the Author):

The technical content of this revised version has been largely improved. All of my concerns are addressed. This reviewer now recommends its publication.